# Soma-localized Rab39 inhibits synaptic autophagy by controlling trafficking of Atg9 vesicles

Ayse Kilic [ID] [1,2], Gokhan Ozturan [ID] [1,2], Dirk Vandekerkhove[1,2], Sabine Kuenen [ID] [1,2], Jef Swerts [ID] [1,2], Esther Muñoz Pedrazo [ID] [1,2], Carles Calatayud Aristoy[1,2], Abril Escamilla Ayala [ID] [3], Nikky Corthout [ID] [3], Pablo Hernández Varas [ID] [3], Stéphane Plaisance [ID] [4], Valerie Uytterhoeven [ID] [1,2✉], Eliana Nachman [ID] [1,2✉] & Patrik Verstreken [ID] [1,2✉]

## Abstract

**Presynaptic terminals can be located far from the neuronal cell body and are thought to independently regulate protein and organelle turnover. Autophagy is a critical process for maintaining proteostasis, and its synaptic dysregulation is associated with neurodegenerative diseases. In this work, we report a soma-centered mechanism that regulates autophagy-controlled protein turnover at distant presynaptic terminals in *Drosophila*. We show that a central component of this system is Rab39, whose human homolog RAB39B is mutated in Parkinson's disease. Although Rab39 is localized in the soma, its loss of function or a human pathogenic mutation causes increased autophagy at presynaptic terminals, resulting in faster synaptic protein turnover and dopaminergic synapse degeneration. Using a large-scale unbiased genetic modifier screen, we identified genes encoding cytoskeletal and axonal organizing proteins, including Shortstop (Shot), as suppressors of synaptic autophagy. We demonstrate that active Rab39 selectively controls Shot- and Unc104/KIF1A-mediated delivery of autophagy-related Atg9-positive vesicles to synapses. Our findings suggest that Rab39-mediated trafficking in the soma orchestrates a cross-compartmental mechanism that regulates the levels of autophagy at synapses.**

**Keywords** Synaptic Autophagy; Rab39; *Drosophila*; Parkinson's Disease; Genetic Suppressor Screen
**Subject Categories** Autophagy & Cell Death; Membranes & Trafficking; Neuroscience

See also: S Huang and SJ Sigrist

## Introduction

Synaptic autophagy is essential for maintaining neuronal health by ensuring the clearance and recycling of dysfunctional synaptic components (Soukup et al, 2016; Bademosi et al, 2023; Vijayan and Verstreken, 2017; Birdsall and Waites, 2019). This process is likely crucial for coping with the high metabolic activity and is required for the functional integrity of synapses and effective neuronal communication. Given that synapses are often located far from the cell body, they appear to independently regulate this process. It remains unclear whether there is also cell body oversight in the regulation of synaptic autophagy.

Several proteins mutated in Parkinson's disease (PD) play key roles in regulating autophagy at synapses (Grosso Jasutkar and Yamamoto, 2023; Nachman and Verstreken, 2022). Leucine-rich repeat kinase 2 (LRRK2)/Lrrk-dependent phosphorylation of the synaptic protein EndophilinA (EndoA) (Matta et al, 2012) drives the process at *Drosophila* neuromuscular junctions (NMJ) in a starvation and activity-dependent manner, and PD pathogenic mutations in EndoA disrupt autophagy at presynaptic terminals in flies and human dopaminergic induced neurons (Bademosi et al, 2023; Soukup et al, 2016). Similarly, EndoA binds to several PD-linked proteins, including Parkin (Cao et al, 2014) and Synaptojanin-1 (Watanabe et al, 2018), which are also both causally mutated in PD. Synaptojanin-1, like EndoA, regulates vesicle trafficking and autophagy specifically at synapses and not in neuronal cell bodies in flies and nematodes (Vanhauwaert et al, 2017; Yang et al, 2022). Interestingly, in PD, neurodegeneration is believed to begin with subtle synaptic defects before leading to overt neuronal death (Schirinzi et al, 2016) and deregulation of synaptic autophagy, including the pathogenic mutation in EndoA, causes neurodegeneration in vivo (Bademosi et al, 2023; Soukup et al, 2016; Grosso Jasutkar and Yamamoto, 2023). Interestingly, both excessive and insufficient synaptic autophagy have the same detrimental outcome, highlighting the importance of maintaining a finely tuned balance of this process to ensure neuronal health.

Human genetic evidence links vesicle trafficking defects to PD. Rab GTPases are crucial regulators of cellular trafficking events (Schimmöller et al, 1998) and Genome-Wide Association Studies

[1]VIB-KU Leuven Center for Brain & Disease Research, Leuven 3000, Belgium. [2]KU Leuven, Department of Neurosciences, Leuven Brain Institute, Leuven 3000, Belgium. [3]VIB BioImaging Core, VIB-Center for Brain and Disease Research, Leuven 3000, Belgium. [4]VIB Nucleomics Core, Leuven 3000, Belgium. ✉E-mail: valerie.uytterhoeven@kuleuven.be; eliana.nachman@kuleuven.be; patrik.verstreken@kuleuven.be

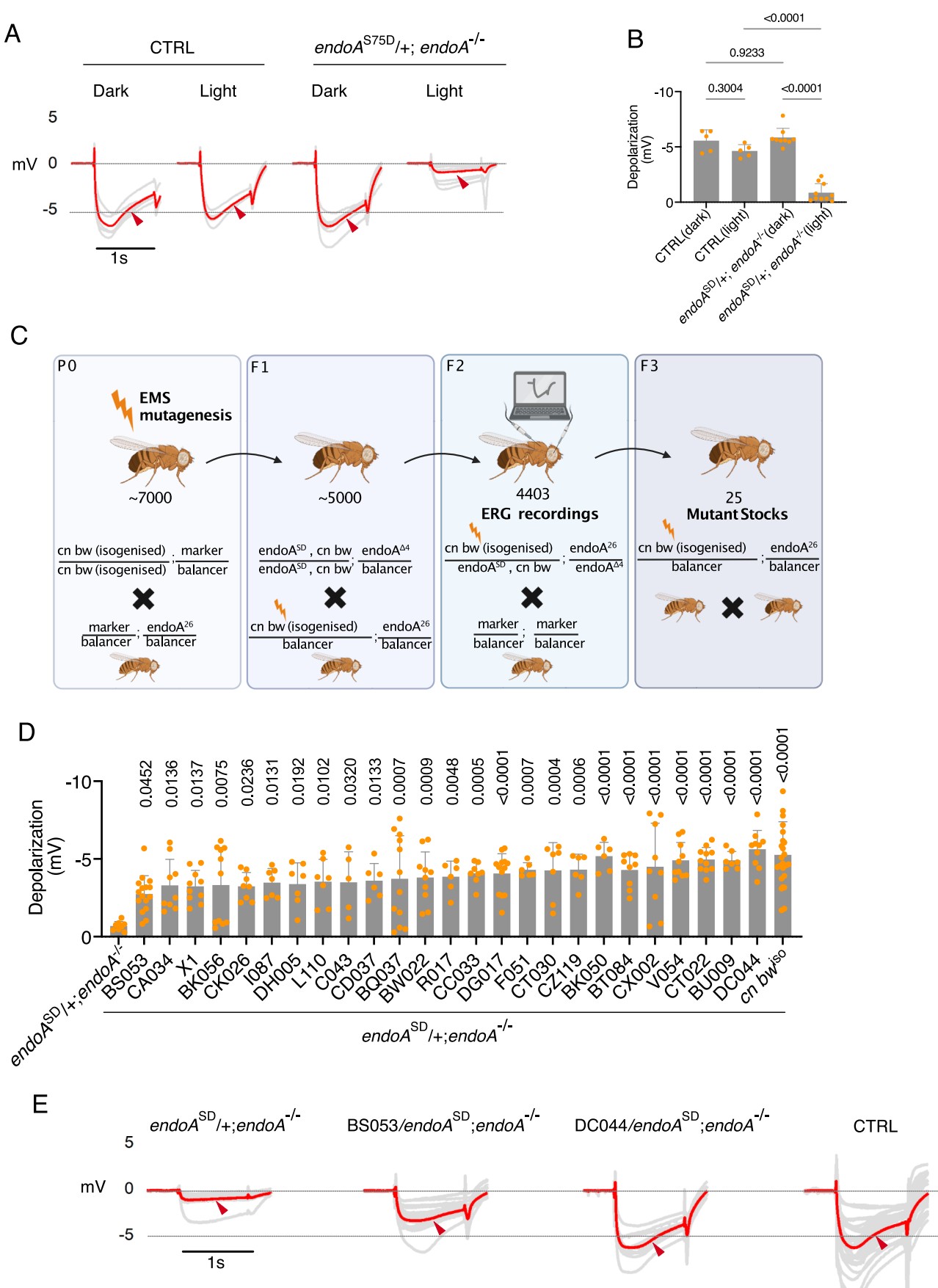

◄ **Figure 1. A forward genetic screen identified modifiers of EndoA-mediated neuronal dysfunction.**

(A) ERG traces of *cn bw* flies (control, CTRL) and *endoA* mutant flies expressing *endoA*[S75D] in the *endoA*[−/−] background (referred as *endoA*[SD] mutant) at endogenous levels. Flies were kept in darkness or exposed to 3 days of constant light prior to recording. Gray traces represent individual animals; red trace indicates the mean. Red arrowhead marks depolarization. (B) Quantification of ERG depolarization amplitudes shown in (A) CTRL (light)- *endoA*[SD]/+; *endoA*[−/−] (light): exact $P = 5.204944 \times 10^{-12}$, *endoA*[SD]/+; *endoA*[−/−] (light)- *endoA*[SD]/+; *endoA*[−/−] (dark): exact $P = 4.753198 \times 10^{-12}$. Statistical test: one-way ANOVA with Tukey's multiple comparison test; $n = 5$–10; error bars: mean ± SD. (C) Schematic representation of the forward genetic screen on the second chromosome using EMS mutagenesis to identify modifiers of the *endoA*[S75D]- light-induced phenotype (see "Methods"). (D) Quantification of ERG depolarization amplitudes from 25 mutant lines that rescued the *endoA*[SD] phenotype. *endoA*[SD]/+; *endoA*[−/−] - [DG017] exact $P = 6.604688682 \times 10^{-5}$; [BK050] exact $P = 1.300037296 \times 10^{-5}$; [BT084] exact $P = 8.404176049 \times 10^{-5}$; [CX002] exact $P = 2.032532121 \times 10^{-5}$; [V054] exact $P = 6.627729696 \times 10^{-7}$; [CT022] exact $P = 1.732450590 \times 10^{-7}$; [BU009] exact $P = 1.552056966 \times 10^{-5}$; [DC044] exact $P = 1.561985208 \times 10^{-9}$. Statistical test: one-way ANOVA with Dunnett's post hoc test; $n = 5$–20; error bars: mean ± SD. (E) Representative ERG traces from modifier lines BS053 and DC044 that suppressed the depolarization defect of *endoA*[SD] animals. Mean traces from 5 to 10 flies exposed to 3 days of constant light are shown. Genotypes: *cn bw* (control, CTRL), *endoA*[SD]/+; *endoA*[−/−] (*endoA*[SD]), BS053/*endoA*[SD]; *endoA*[−/−] (BS053), DC044/*endoA*[SD]; *endoA*[−/−] (DC044). Red arrowhead indicates depolarization. Source data are available online for this figure.

(GWAS) have identified variants at the RAB29 locus that increase the risk of PD (Kia et al, 2021; Nalls et al, 2019; Tan et al, 2024). In addition, a rare mutation in another Rab-GTPase, RAB32, also increases the risk for PD (Gustavsson et al, 2024; Hop et al, 2024), and familial mutations implicate the loss of function of RAB39B, as causative to the disease (Gao et al, 2020b; Jacobson et al, 2024). Moreover, LRRK2 has been shown to mediate the phosphorylation of various Rabs, including Rab3, Rab8, Rab10, Rab12, and possibly Rab29 (Steger et al, 2016; Fujimoto et al, 2018; Liu et al, 2018; Purlyte et al, 2018; Dou et al, 2024). While the connection of PD to Rab GTPases is evident, their potential relation to synaptic EndoA-dependent autophagy remains less well understood.

Here, we demonstrate that similar to a *rab39* knockout, PD-associated mutations in RAB39B result in increased synaptic autophagy, akin to the effects observed in phosphomimetic EndoA[S75D] mutants (Soukup et al, 2016). To investigate the regulation of this process, we utilized the *endoA*[S75D] mutant in a large-scale unbiased genetic modifier screen in *Drosophila*. This approach uncovered key regulators of synaptic autophagy, indicating that the loss of function in genes involved in cytoskeletal organization and axonal transport suppresses the increased autophagy in both *endoA*[S75D] and *rab39* mutants. While EndoA localizes directly at synapses, Rab39 is primarily restricted to the neuronal cell body (Chan et al, 2011; Gao et al, 2020c). Strikingly, we found that Rab39 within the soma plays a critical role in inhibiting autophagy at the synapse by controlling the trafficking of Atg9-positive vesicles. Disruption of this Rab39-dependent trafficking pathway results in increased Atg9 transport, and this facilitates synaptic autophagy, ultimately leading to neurodegeneration, including age-dependent dopaminergic neuron synapse loss. Our work indicates that a fine balance of autophagy at synapses is crucial for neuronal and synapse health, and we reveal a Rab39-mediated trafficking mechanism in the neuronal cell body that regulates the abundance of Atg9 vesicles, controlling synaptic autophagy. This cell body-centric process has significant implications for proteostasis at synapses and provides novel insights into the pathophysiology of PD.

## Results

### A forward genetic screen identifies modifiers of autophagy-induced neuronal dysfunction

To identify regulators of synaptic autophagy, we conducted a forward genetic screen in *Drosophila* with *endoA*[S75D] mutant flies which exhibit increased synaptic autophagy and light-induced

neurodegeneration (LIND) (Soukup et al, 2016; Bademosi et al, 2023). In this assay, flies are exposed to constant light for several days, which stresses the neurons of the visual system and thereby unmasks defects in neuronal maintenance and proteostasis pathways (Xiong and Bellen, 2013; Shieh, 2011). Degeneration of the neurons in the visual circuit results in a reduction of the electroretinogram (ERG) depolarization amplitude. ERGs are an easy electrophysiological assay that measures the voltage differences in the fly compound eye in response to a 1 s light flash (Fig. 1A). Thus, a lower ERG amplitude after light exposure indicates neuronal degeneration (Hotta and Benzer, 1969). As we have shown previously, when *endoA*[S75D] mutant flies are kept in constant light for 3 days, a decrease in the depolarization amplitude of the ERG response can be observed, indicating LIND (Fig. 1A,B) (Soukup et al, 2016). We leveraged this light-induced ERG phenotype as a readout to screen for genetic modifiers of synaptic autophagy. We generated thousands of EMS-induced lethal mutations on the second chromosome of *Drosophila* (Fig. 1C, labeled P0). This collection was derived from approximately 7000 male flies, each carrying an isogenized second chromosome with *cn bw* mutations (leading to white eyes, which facilitates LIND) (Escobedo et al, 2022; Tearle, 1991). In the subsequent F2 generation, ERGs were recorded from 4403 fly lines harboring a second chromosome with a transgene expressing the *endoA*[S75D] mutation and a random EMS-mutated second chromosome (Fig. 1C, labeled F2). The presence of one of 25 EMS-mutated second chromosomes, hereafter called "modifier lines", rescued the *endoA*[S75D]–induced ERG phenotype (Fig. 1C, labeled F3, Fig. 1D,E).

### Modifiers of *endoA*-induced neuronal dysfunction also rescue several PD mutants

*EndoA* interacts with several PD genes, including *Lrrk2* and Synaptojanin-1 *(Synj1)* (Cao et al, 2014; Watanabe et al, 2018; Soukup et al, 2016). Notably, all three genes, *EndoA*, *Synj1*, and *Lrrk2*, have been directly implicated in regulating synaptic autophagy in *Drosophila* (Vanhauwaert et al, 2017; Soukup et al, 2016). The PD gene RAB39B is also thought to play a role in autophagy (Corbier and Sellier, 2017; Niu et al, 2020), but its role in synaptic autophagy has not been investigated. When knockout (KO) mutant flies of these genes (Kaempf et al, 2024b) were tested in the LIND assay, ERG recordings of *lrrk*[KO-WS], *synj*[KO-WS], and *rab39*[KO-WS] (hereafter denoted as "KO") showed a consistent decrease in the amplitude of the depolarization (Fig. 2A),

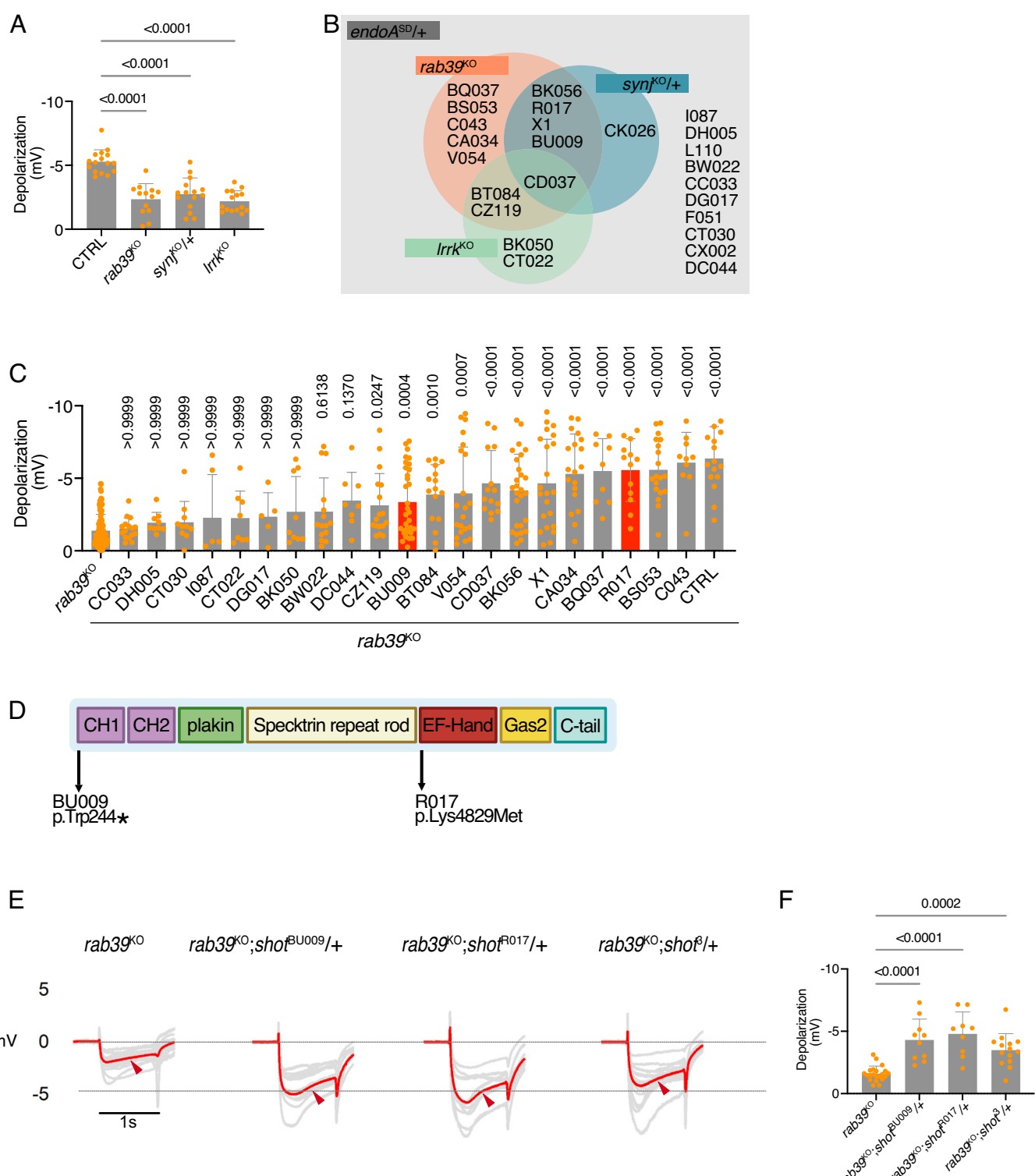

a phenotype like that observed in *endoA*^S75D mutants under the same conditions. We next assessed whether our modifier chromosomes also rescued the ERG phenotypes associated with these other PD genes. Interestingly, we found that 12 of the 25 *endoA*^S75D modifiers also rescued the *rab39*^KO ERG phenotype (Fig. 2B,C), 6

modifiers rescued the *synj*^KO, and 5 rescued the *lrrk*^KO phenotype (Fig. 2B). Hence, several *endoA*^S75D modifiers are shared across *lrrk*, *synj*, and *rab39*.

Given that *endoA*^S75D and *rab39*^KO mutants showed the largest overlap in modifiers (Fig. 2B), we identified the lesions on the

**Figure 2. Modifiers of EndoA-induced neuronal dysfunction are shared among PD mutants.**

(A) Quantification of ERG depolarization amplitudes in flies reared under 7 days of constant light with the following genotypes: * cn bw (control, CTRL) * rab39^KO;cn bw * synj^KO,cn bw/+ * cn bw;lrrk^KO *. Statistical test: one-way ANOVA with Dunnett's multiple comparison test; n = 12–16; error bars: mean ± SD. (B) Venn diagram summarizing EMS modifier line effects on the ERG phenotype in endoA^SD,cn bw;endoA^−/−, rab39^KO;cn bw, synj^KO,cn bw/+, cn bw;lrrk^KO backgrounds. Listed lines rescued the depolarization defect in the corresponding genotypes. (C) Quantification of ERG depolarization amplitudes in rab39^KO animals carrying EMS modifier lines. Lines BU009 and R017, both harboring mutations in the shortstop (shot) gene, are indicated in red. Statistical test: Kruskal–Wallis test with Dunn's multiple comparison test; n = 5–30; error bars: mean ± SD. (D) Schematic of the Shot protein showing EMS-induced mutations: an amino acid substitution (Lys4829Met) in R017 and a premature stop codon (Trp244*) in BU009. (E) Representative ERG traces from flies of the indicated genotypes reared under 7 days of constant light. Gray traces indicate individual flies; red trace indicates the mean. Arrowheads mark depolarization. (F) Quantification of ERG depolarization amplitudes from the experiment in (E). Statistical test: one-way ANOVA with Dunnett's multiple comparison test; n = 9–20; error bars: mean ± SD. Source data are available online for this figure.

12 second chromosomes that could rescue both mutants using whole-genome sequencing. After filtering the sequence variants (Fig. EV1A,B), we retained the mutations that were predicted to have a moderate to high impact on protein function (i.e., mutations affecting splice sites, start and stop codons, and non-synonymous variations) and that occurred more than once independently in the different modifier lines but not in the original isogenized control (Fig. EV1A,B). This approach yielded 23 genes. GO term analysis using FlyBase biological process annotations indicated a predominant representation of "cytoskeleton organization" (Table EV1) (Jenkins et al, 2022). This included a hit in short stop (shot) that we recovered independently twice in our screen: shot^BU009 and shot^R017. Sequencing identified a nonsense mutation at position 244 in shot^BU009, resulting in a premature stop codon (p.Trp244*), and a missense mutation at position 4829 in shot^R017 (p.Lys4829Met) (Fig. 2D). We independently confirmed the causal role for loss of shot in rescuing the rab39^KO ERG phenotype by using the previously generated shot^3 null allele (Kolodziej et al, 1995) that was also effective at rescuing the rab39^KO ERG defect (Fig. 2E,F). Hence, removing one allele of shot rescues endo^S75D and rab39^KO-induced neuronal dysfunction. Notably, a variation in the mammalian shot ortholog, MACF1, has been associated with susceptibility to PD (Wang et al, 2017).

### rab39^KO increases synaptic autophagy

To investigate if Rab39, like EndoA, plays a role in synaptic autophagy, we performed live confocal imaging of Atg8-mCherry expressed under endogenous atg8-promoter control, at rab39^KO and control Drosophila third-instar larval NMJs under basal conditions. Atg8 (autophagy-related gene 8) is a cytosolic protein that is conjugated to autophagosome membranes upon initiation of autophagy and thus serves as a marker for autophagosome formation and the presence of autophagosomes (Nakatogawa et al, 2007). We observed a significant increase in Atg8-labeled puncta in synaptic boutons of rab39^KO larvae compared to controls (Fig. 3A,E), and this defect was rescued by re-expression of wild-type Drosophila rab39 (nSyb-Gal4) (Fig. 3B,C,F). Conversely, we did not observe an increase in Atg8-labeled puncta in neuronal cell bodies of rab39^KO mutants, indicating the effect we observed was synapse-specific (Appendix Fig. S1). The results indicate that loss of Rab39 causes elevated synaptic autophagosome abundance under basal (non-starved, non-stimulated) conditions.

We verified that the puncta we observed were indeed autophagosomes by (1) genetics and by (2) ultrastructural analysis: (1) The Atg8-mCherry labeled puncta at synapses were nearly abolished upon downregulation of the essential autophagy protein

Atg3 using the expression of Atg3-RNAi in control and rab39^KO larvae (Fig. 3D,G). (2) We employed correlative light and electron microscopy (CLEM) to visualize the ultrastructural morphology of the mCherry puncta at rab39^KO synaptic boutons (Fig. EV2). Alignment of the mCherry fluorescent image with the TEM micrograph showed overlap with structures reminiscent of degradative organelles such as autophagosomes and autolysosomes (Eskelinen et al, 2003) (Fig. EV2).

Finally, we asked if the observed increase in Atg8-positive puncta in rab39^KO synapses resulted from enhanced autophagosome biogenesis, a blockage in autophagic flux, or impaired trafficking in axons away from the synapse. To distinguish between these possibilities, we performed a series of complementary experiments.

First, we examined retrograde trafficking of autophagosomes from the synapse in rab39^KO axons. We performed live imaging of Atg8-positive vesicles in axons directly adjacent to type Ib boutons and quantified their movement (Fig. 3H–K). We found both an elevated number of Atg8-positive vesicles in axons of rab39^KO animals (Fig. 3I) as well as an increase in the fraction of retrogradely moving autophagosomes (Fig. 3J) compared to controls. The transport speed of autophagosomes in rab39^KO was comparable to controls (Fig. 3K). These results suggest that autophagosomes can form at synaptic terminals in rab39^KO animals, they remain motile and are efficiently transported away from presynaptic NMJ terminals.

To then assess whether autophagosome formation flux was altered, we expressed the dual-tagged autophagic flux reporter Atg8-mCherry-GFP in NMJs (Fig. 3L) (Chang and Neufeld, 2009). Atg8-positive autophagosomes contain thus both red and green fluorescent proteins, but following the fusion with the acidic lysosome, the GFP signal is quenched by the low pH, leaving only mCherry red signal (Fig. 3L) (Shinoda et al, 2018). At rab39^KO synapses, we observed a higher red over green fluorescence intensity ratio compared to controls (Fig. 3M), suggesting that autophagosomes formed and that their acidification by lysosomal fusion was not impaired in rab39^KO. Hence, our findings argue for increased autophagosome formation in rab39^KO animals or for reduced degradation downstream of lysosomal fusion.

To distinguish between these possibilities, we assessed the turnover of the synaptic protein Synaptobrevin (nSyb) that is degraded by synaptic autophagy (Kallergi et al, 2023; Goldsmith et al, 2022). We monitored nSyb turnover by expressing an nSyb-fluorescent timer (FT) fusion (FT::nSyb) (Uytterhoeven et al, 2011; Fernandes et al, 2014). The FT slowly converts from blue (younger protein) to red (older protein) fluorescence over time (Fig. 3N) (Subach et al, 2009). In rab39^KO we detected an increased blue-to-

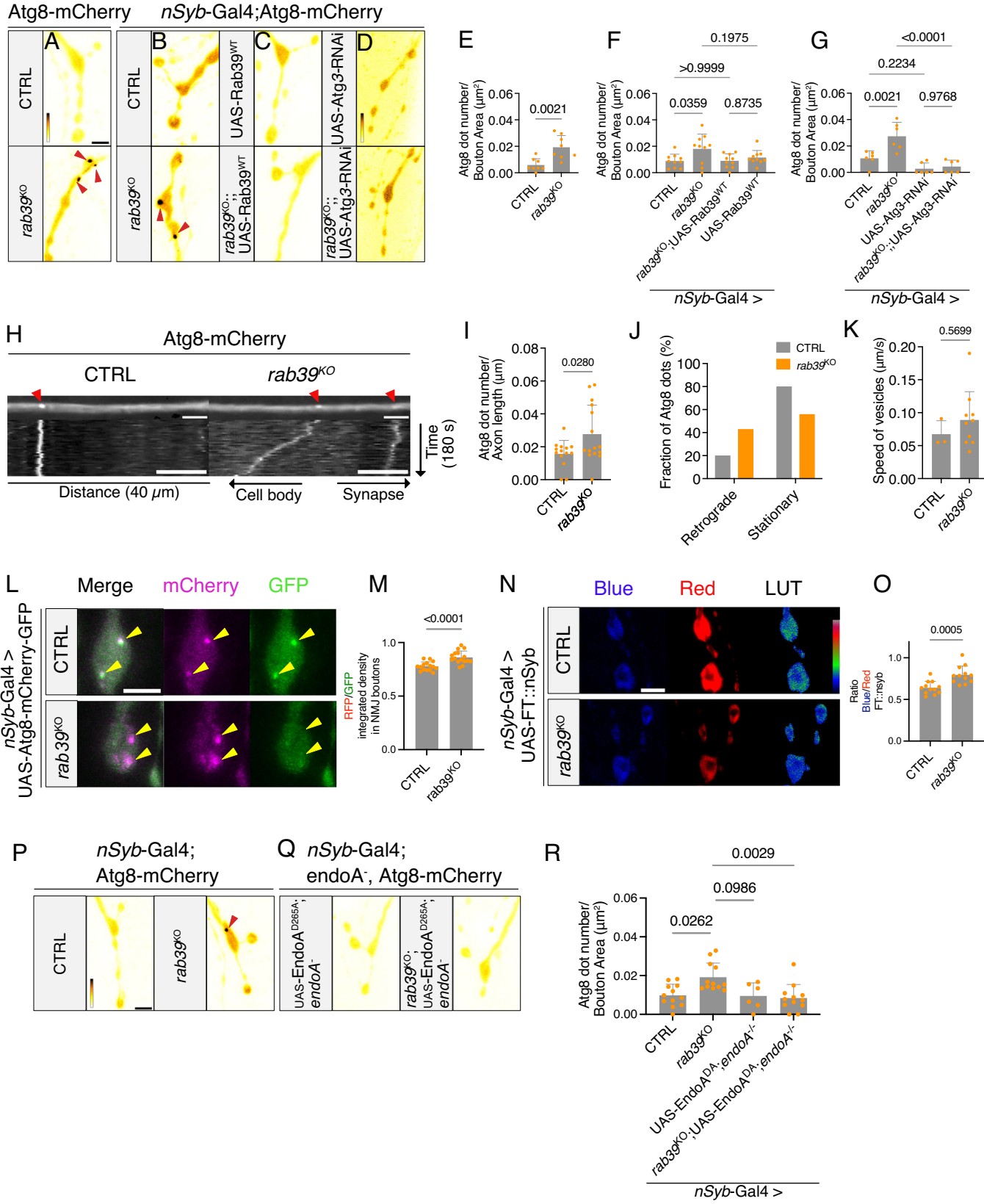

**Figure 3.  rab39^{KO} increases synaptic autophagy at *Drosophila* NMJs.**

(A–D) Live imaging of genomically expressed Atg8-mCherry at NMJ boutons of (A) control ($w^{1118}w^+$) and *rab39*^{KO} animals, and of (B) control and *rab39*^{KO} animals with the *nSyb*-Gal4 driver expressing either (C) UAS-Rab39^{WT}, (D) UAS-Atg3-RNAi. Gray value scaling is shown in (A) for panels (A–C) (range 255–2813) and in (D) (range 138–1359). Arrowheads indicate Atg8-mCherry puncta. Scale bar: 5 µm. (E–G) Quantification of Atg8-mCherry puncta per NMJ area from (A–D) *rab39*^{KO}-*rab39*^{KO};; UAS-Atg3-RNAi: exact $P = 0.0000594$. Statistical tests: unpaired *t* test ((E), $n = 8$); one-way ANOVA with Tukey's multiple comparison test ((F, G), $n = 6$–10); error bars: mean ± SD. (H) Representative images of Atg8-mCherry puncta in axons from third-instar control ($w^{1118}w^+$) and *rab39*^{KO} larvae. Arrowheads indicate Atg8-mCherry-positive puncta. Bottom panels: corresponding kymographs from time-lapse imaging (180 s) of the same axonal region. Diagonal lines represent motile vesicles; vertical lines represent stationary vesicles. Scale bars: 5 µm (axon images), 10 µm (kymographs, *x* axis). (I) Quantification of Atg8-mCherry puncta per axon length from the experiment in (H). Statistical test: Welch's *t* test; $n = 14$–15; error bars: mean ± SD. (J) Fractional distribution of vesicle directionality (retrograde, stationary) pooled across axons from control and Rab39^{KO} animals. Statistical test: Fisher's Exact Test; *P* value $= 0.1755$. Exact vesicle counts are listed in Appendix Table S1. (K) Quantification of the speed (µm/s) of Atg8-mCherry-positive vesicles from (H). Statistical test: Mann–Whitney test; $n = 3$–10; error bars: mean ± SD. (L) Live imaging of control and Rab39^{KO} larvae expressing UAS-Atg8-mCherry-GFP under *nSyb*-Gal4. GFP signal in green, mCherry in magenta. Yellow arrowheads mark Atg8-positive puncta. Scale bar: 5 µm. (M) Quantification of the mCherry/GFP integrated density ratio at NMJ boutons from (L) exact $P = 8.871 \times 10^{-5}$. Statistical test: unpaired *t* test; $n = 14$–15; error bars: mean ± SD. (N) Live imaging of control and *rab39*^{KO} animals expressing the FT::nSyb construct in NMJ boutons using *nSyb*-Gal4. Scale bar: 5 µm. (O) Quantification of blue (young) over red (old) fluorescence intensity ratios at synaptic boutons from (N). Statistical test: unpaired *t* test; $n = 12$; error bars: mean ± SD. (P, Q) Live imaging of Atg8-mCherry at NMJ boutons in control (CTRL) and *rab39*^{KO} animals (P), and in *endoA*^{−/−} and *rab39*^{KO}; *endoA*^{−/−} animals expressing UAS-EndoA^{D265A} (Q), driven by *nSyb*-Gal4. Fluorescence intensity scaling (365–4095) shown in (P), (CTRL). Arrowheads indicate Atg8-mCherry puncta. Scale bar: 5 µm. (R) Quantification of Atg8-mCherry puncta per bouton area from (P, Q). Statistical test: Kruskal–Wallis test with Dunn's multiple comparisons test; $n = 6$–12; error bars: mean ± SD. Source data are available online for this figure.

red ratio compared to controls, indicating relatively less older and more younger FT::nSyb, consistent with increased FT::nSyb turnover (Fig. 3N,O). We also assessed the absolute levels of other key synaptic proteins, such as BRP (Bruchpilot), CSP (Cysteine String Protein), and Syntaxin-1A (Wagh et al, 2006; Umbach et al, 1995; Bennett et al, 1992), that have also been detected as synaptic autophagosome cargo (Kallergi et al, 2023; Goldsmith et al, 2022). However, we found no significant differences in protein abundance between control and *rab39*^{KO} animals (Appendix Fig S2). This suggests that while steady-state protein levels can be maintained, tools like FTs or other labeling techniques (e.g., Stable isotope labeling by amino acids in cell culture (SILAC) (Ong et al, 2002)) are required to assess turnover defects in the context of synaptic autophagy. Taken together, our findings suggest *rab39*^{KO} causes increased synaptic autophagic flux, including retrograde transport, and increased synaptic protein (FT::nSyb) turnover.

### *endoA* is epistatic to *rab39*

Our data show that *endoA* and *rab39* both regulate synaptic autophagy. We therefore assessed their epistatic relationship. We used the *endoA*^{D265A} mutant (in an *endoA* null mutant background) that blocks synaptic autophagy (Bademosi et al, 2023) and combined it with *rab39*^{KO} that increases synaptic autophagy (*rab39*^{KO}; UAS-EndoA^{D265A}/nSyb-Gal4; *endoA*^{Δ4}/*endoA*^{26}). Interestingly, in these animals, we recorded low synaptic autophagy levels, similar to those in *EndoA*^{D265A} mutants (UAS-EndoA^{D265A}/nSyb-Gal4; *endoA*^{Δ4}/*endoA*^{26}) (Fig. 3P–R). This indicates that the loss-of-function phenotype of *endoA*^{D265A} overrides the effect of *Rab39* deletion. In genetic terms, this suggests that *endoA*^{D265A} is epistatic to *rab39*.

### Disrupted *rab39* expression impairs synaptic transmission at the larval NMJ

To investigate if the loss of Rab39 affects synaptic function, we performed two-electrode voltage clamp recordings at third-instar larval NMJs. Evoked junctional currents (EJCs) were recorded from muscle 6 in the presence of 1 mM extracellular Ca²⁺ using 1 Hz nerve stimulation. *rab39*^{KO} larvae exhibited a significant reduction

in EJC amplitudes compared to controls (Fig. 4A,B), indicating impaired synaptic transmission. This phenotype was rescued by neuronal expression of wild-type *rab39* in *rab39*^{KO} (Fig. 4A,B). Interestingly, overexpression of wild-type *rab39* in a control ($w^{1118}$ $w^+$) background also led to a reduction in EJC amplitudes to levels comparable to those in *rab39*^{KO} animals (Fig. 4A,B). These findings suggest that both loss and overexpression of Rab39 disrupt synaptic function, highlighting the importance of precise Rab39 dosage for maintaining neurotransmission.

We also recorded spontaneous miniature EJCs (mEJC). The cumulative distribution of mEJC amplitudes revealed no significant changes between genotypes (Fig. 4C–E), indicating that quantal size is unaffected. Quantal content (EJC/mEJC) was thus significantly reduced in *rab39*^{KO} and upon Rab39 overexpression and was fully restored by Rab39 overexpression in *rab39*^{KO} (Fig. 4E). While the exact cause of this defect is not resolved, the results align with previous reports suggesting that excessive autophagy in neurons contributes to reducing neurotransmitter release by depleting synaptic vesicles (Hernandez et al, 2012) and disrupting calcium homeostasis (Kuijpers et al, 2021; Decet and Verstreken, 2021). Together, these data support a model in which Rab39 regulates synaptic function in a dosage-sensitive manner, potentially by controlling presynaptic autophagy and protein homeostasis.

### The pathogenic mutant RAB39B^{T168K} causes increased synaptic autophagy

Pathogenic mutations in the *RAB39B* gene cause PD (Wilson et al, 2014). We therefore investigated if the pathogenic human RAB39B^{T168K} mutant affects synaptic autophagy in vivo. We used a knock-in strategy where the first exon of the wild-type fly *rab39* gene was replaced by either the entire wild-type human cDNA (*RAB39B*^{WT}) or the pathogenic T168K mutant variant (*RAB39B*^{T168K}) (Fig. EV3A,B). Deletion of the first exon of *Drosophila rab39* results in a complete loss-of-function allele as this exon contains the start codon and essential coding sequence, abolishing endogenous fly *rab39* expression (this is the *rab39*^{KO-WS} aka *rab39*^{KO} allele; Fig. EV3C). We then inserted the human cDNAs to replace the first exon, and as such they are expressed under control of the native fly promoter and (intronic) regulatory

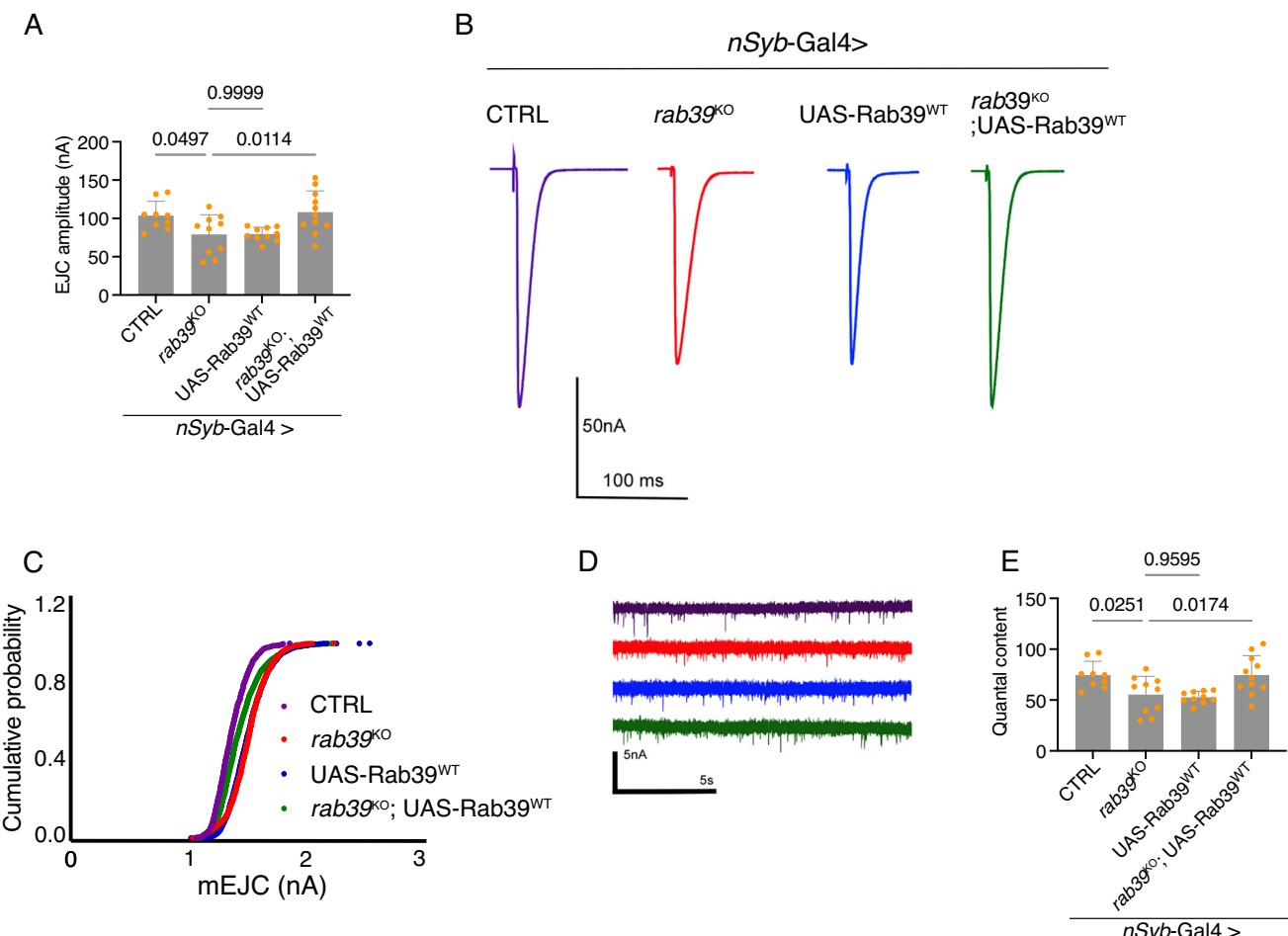

**Figure 4. Disrupted Rab39 expression impairs synaptic transmission at the NMJ.**

(A) Quantification of evoked junctional current (EJC) amplitudes recorded from larval NMJs in 1 mM extracellular Ca$^{2+}$ under 1 Hz stimulation in the following genotypes: control ($w^{1118}w^{+}$, CTRL), $rab39^{KO}$, Rab39 wild-type overexpression ($nSyb$-$Gal4 > UAS$-$Rab39^{WT}$), and $rab39^{KO}$ rescue ($nSyb$-$Gal4 > rab39^{KO}$; $UAS$-$Rab39^{WT}$). (B) Representative EJC traces corresponding to the genotypes shown in (A). (C) Cumulative probability distribution of mEJC amplitudes for the genotypes shown in (A). (D) Raw data of spontaneous activity (miniature EJC (mEJC)). (E) Quantal content calculated as the ratio of EJC to mEJC amplitudes for the genotypes shown in (A). (A, E) Statistical test: one-way ANOVA with Dunnett's post hoc test; orange dots represent individual recordings from 7 larvae (CTRL), 5 larvae ($rab39^{KO}$), 6 larvae ($nSyb$ -$Gal4 > UAS$-$Rab39^{WT}$), and 6 larvae ($nSyb$ -$Gal4 > rab39^{KO}$; $UAS$-$Rab39^{WT}$); error bars: mean ± SD. Source data are available online for this figure.

elements, allowing for physiological spatiotemporal expression of human Rab39B variants (Pech et al, 2024). We subsequently assessed Atg8-mCherry fluorescence at presynaptic terminals. While animals expressing RAB39B$^{WT}$ and controls had a similar (low) amount of Atg8-labeled puncta at presynaptic terminals, this number was significantly higher at synapses of animals expressing RAB39B$^{T168K}$ (Fig. EV3D,E). Hence, while wild-type human RAB39B significantly compensates for the loss of fly Rab39, the pathogenic RAB39B$^{T168K}$ mutant is defective, deregulating synaptic autophagy. Our data also confirm that RAB39B$^{T168K}$ is a loss of RAB39B function mutant (Gao et al, 2020a).

## Rab39 GTPase activity regulates synaptic autophagy

Rab39 is a Rab-GTPase cycling between active GTP-bound and inactive GDP-bound states (Müller and Goody, 2017). To determine whether Rab39's GTPase activity is required for the

Rab39-mediated synaptic autophagy regulation, we expressed either *Drosophila* wild-type (Rab39$^{WT}$), a constitutively active (Rab39$^{Q69L}$), or a dominant-negative mutant (Rab39$^{S23N}$) in $rab39^{KO}$ mutants (Zhang et al, 2007) and assessed synaptic autophagy. Rab39$^{WT}$ and the constitutively active Rab39$^{Q69L}$ rescued the elevated synaptic autophagy in $rab39^{KO}$ to levels comparable to controls (Fig. 5A–D,F). In contrast, overexpressing the GTPase-inactive Rab39$^{S23N}$ did not reduce the elevated synaptic autophagy levels in $rab39^{KO}$ (Fig. 5E,F). Hence, Rab39 GTPase activity is required to inhibit synaptic autophagy.

## Rab39 increases axonal Atg9 vesicle abundance and delivery to synapses

The relationship between EndoA and Rab39 warranted further investigation because the proteins are thought to localize to different neuronal compartments: EndoA strongly localizes to

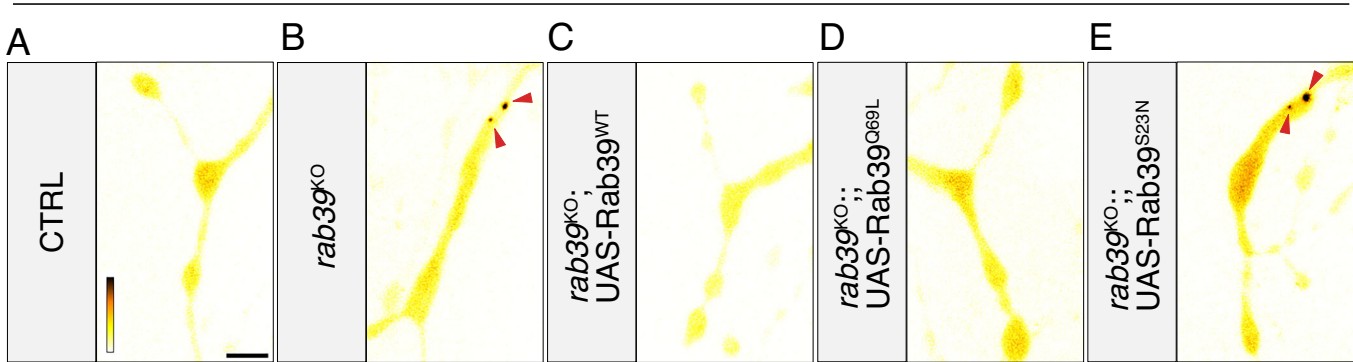

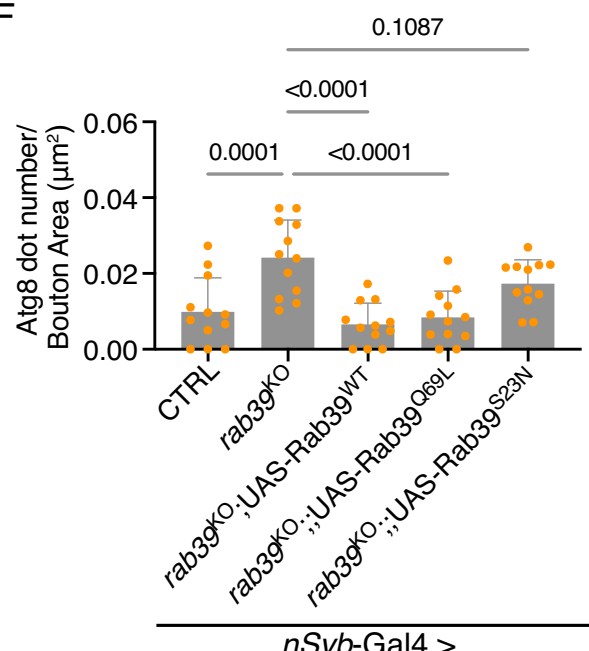

**Figure 5. Rab39 GTPase activity regulates synaptic autophagy.**

(A–E) Live imaging of genomically expressed Atg8-mCherry at NMJ boutons in control ($w^{1118}w^+$, CTRL) and $rab39^{KO}$ (B) animals expressing UAS-$Rab39^{WT}$(C), UAS-$Rab39^{Q69L}$ (D), or UAS-$Rab39^{S23N}$ (E) under the control of $nSyb$-Gal4. Fluorescence intensities are shown using the gray value range indicated in (A) (474–3423). Arrowheads mark Atg8-mCherry-positive puncta. Scale bar: 5 μm. (F) Quantification of the number of Atg8-mCherry puncta per bouton area from the experiment shown in (A–E) $rab39^{KO}$-$rab39^{KO}$; UAS-$Rab39^{WT}$: exact $P = 8.871 \times 10^{-5}$; $rab39^{KO}$-$rab39^{KO}$; UAS-$Rab39^{Q69L}$: exact $P = 5.848339 \times 10^{-5}$. Statistical test: one-way ANOVA with Dunnett's multiple comparison test; $n = 12$; error bars: mean ± SD. Source data are available online for this figure.

presynaptic terminals in flies (Verstreken et al, 2002), while Rab39 is thought to be a soma-associated protein (Chan et al, 2011; Gao et al, 2020c). We verified the intracellular distribution of Rab39 in flies expressing functional eYFP-tagged Rab39 at endogenous levels (Dunst et al, 2015; Miao et al, 2023) and indeed found Rab39 to be strongly concentrated in the neuronal cell body (Fig. 6A,C), without significant labeling at synaptic boutons of NMJs (Fig. 6B,C).

We then wondered how a soma-restricted protein could control autophagy at synapses?

Our genetic suppressor screen (Fig. 1) identified regulators of "cytoskeleton organization", and one of our strongest common suppressor genes was *shot* (Fig. 2D–F). Shot is a microtubule-

binding protein (Applewhite et al, 2010) stabilizing the axonal cytoskeleton, organizing the axon initial segment (AIS) (Bottenberg et al, 2009; Lee and Kolodziej, 2002) and initiating vesicle trafficking in conjunction with the motor protein Unc104/Kif1A (Voelzmann et al, 2016). We reasoned that such a protein could be involved in soma-to-synapse communication to regulate synaptic autophagy. We therefore crossed the mutant shot alleles *shot³*, *shot^BU009* or *shot^R017* to *rab39^KO*. Heterozygous loss of *shot* not only rescued the *rab39^KO*-ERG phenotype but also restored the levels of synaptic autophagy to control levels (Fig. 7A–E,G). This was specific to the loss of *shot* function as re-introducing a functional copy of *shot⁺* in *rab39^KO*; *shot³/+* animals again showed increased synaptic autophagy levels comparable to *rab39^KO* (Fig. 7F,G). We next examined an established

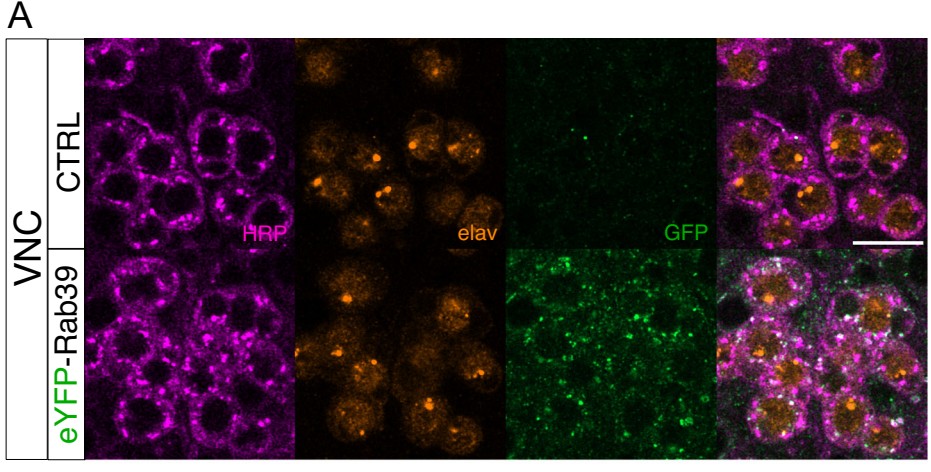

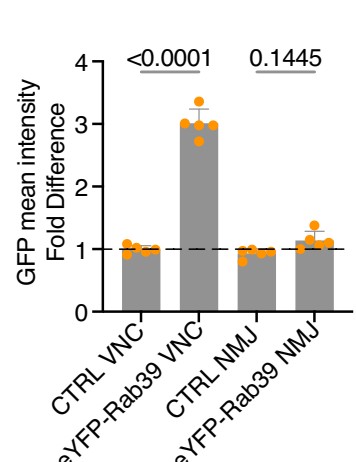

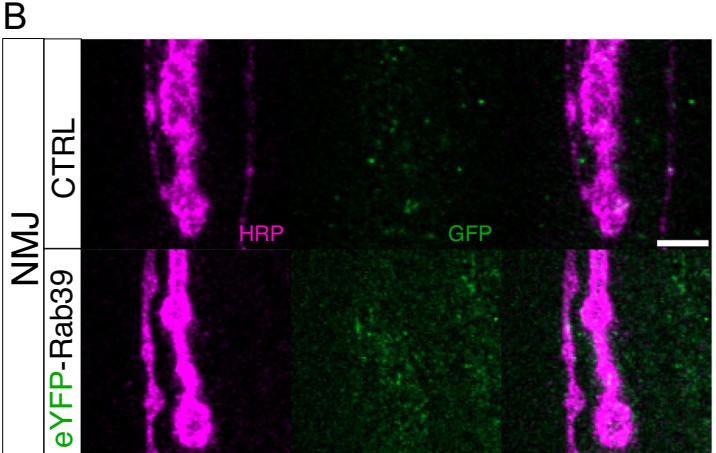

**Figure 6. Rab39 localizes to neuronal somata in the ventral nerve cord.**

(A) Representative Airyscan confocal image of the larval ventral nerve cord (VNC) in control ($w^{1118}$) and eYFP-Rab39 animals. VNCs were stained with anti-HRP (magenta) to label neuronal membranes, anti-Elav (orange) to mark neuronal nuclei, and anti-GFP (green) to visualize endogenously tagged eYFP-Rab39. Scale bar: 10 µm. (B) Representative Airyscan confocal image of a neuromuscular junction (NMJ) bouton stained with anti-GFP (eYFP-Rab39) and anti-HRP (neuronal membrane). Scale bar: 5 µm. (C) Quantification of GFP mean fluorescence intensity in VNC and NMJ regions. Values are normalized to control ($w^{1118}$) levels and shown as fold change. CTRL VNC-eYFP-Rab39: exact $P = 1.436296 \times 10^{-12}$. The dotted line indicates the background signal threshold. Statistical test: one-way ANOVA with Tukey's post hoc test; $n = 5$; error bars: mean ± SD. Source data are available online for this figure.

interaction partner of Shot, Unc104/Kif1A. Heterozygous loss of $unc104$ ($unc104^{P350}/+$) also rescued the increased synaptic autophagy level in $rab39^{KO}$ (Fig. 7H), phenocopying $shot^3$. Hence, our data are consistent with a model where Rab39 regulates synaptic autophagy via an axonal transport mechanism.

Atg9 is a transmembrane protein associated with vesicles and a key component for autophagy initiation (Mari et al, 2010; Rao et al, 2016). Such vesicles need to pass the AIS and be transported to synapses to support synaptic autophagy (Stavoe et al, 2016). Given that Unc104/Kif1A is the principal motor for anterograde transport of Atg9-positive vesicles (Stavoe et al, 2016), we wondered if Rab39 could regulate synaptic autophagy by modulating the delivery of Atg9 vesicles to synaptic terminals (Fig. 7I).

We first asked if the increased levels of Atg8-mCherry puncta in $rab39^{KO}$ mutants depended on Atg9. While heterozygous loss of Atg9 ($atg9^{B5}/+$) in wild-type larvae did not affect synaptic

autophagy, removing a copy of $atg9$ in $rab39^{KO}$ ($rab39^{KO}$; $atg9^{B5}/+$) decreased the elevated synaptic autophagy levels seen in $rab39^{KO}$ to control levels (Fig. 7H). We then characterized Atg9-mCherry transport along motor axons (Fig. 7J–L). While $rab39^{KO}$ animals harbored significantly more Atg9-mCherry-labeled vesicles in their motor axons compared to wild-type larvae (Fig. 7M; Movies EV1 and 2), there was no difference in trafficking speed and their direction of movement (Fig. 7N,O). These data indicate that soma-restricted Rab39 regulates the number of axonal Atg9 vesicles for transport, but not transport (speed, direction) itself.

Given the increased number of axonal Atg9 vesicles in $rab39^{KO}$ mutants, we wondered if more Atg9 is delivered to synapses. We first assessed total synaptic Atg9 levels and quantified immunolabeling intensities at synaptic boutons of endogenously tagged Atg9-mCherry. This revealed no significant difference between $rab39^{KO}$ and control animals (Fig. 8A,B). We next used a FRAP assay to

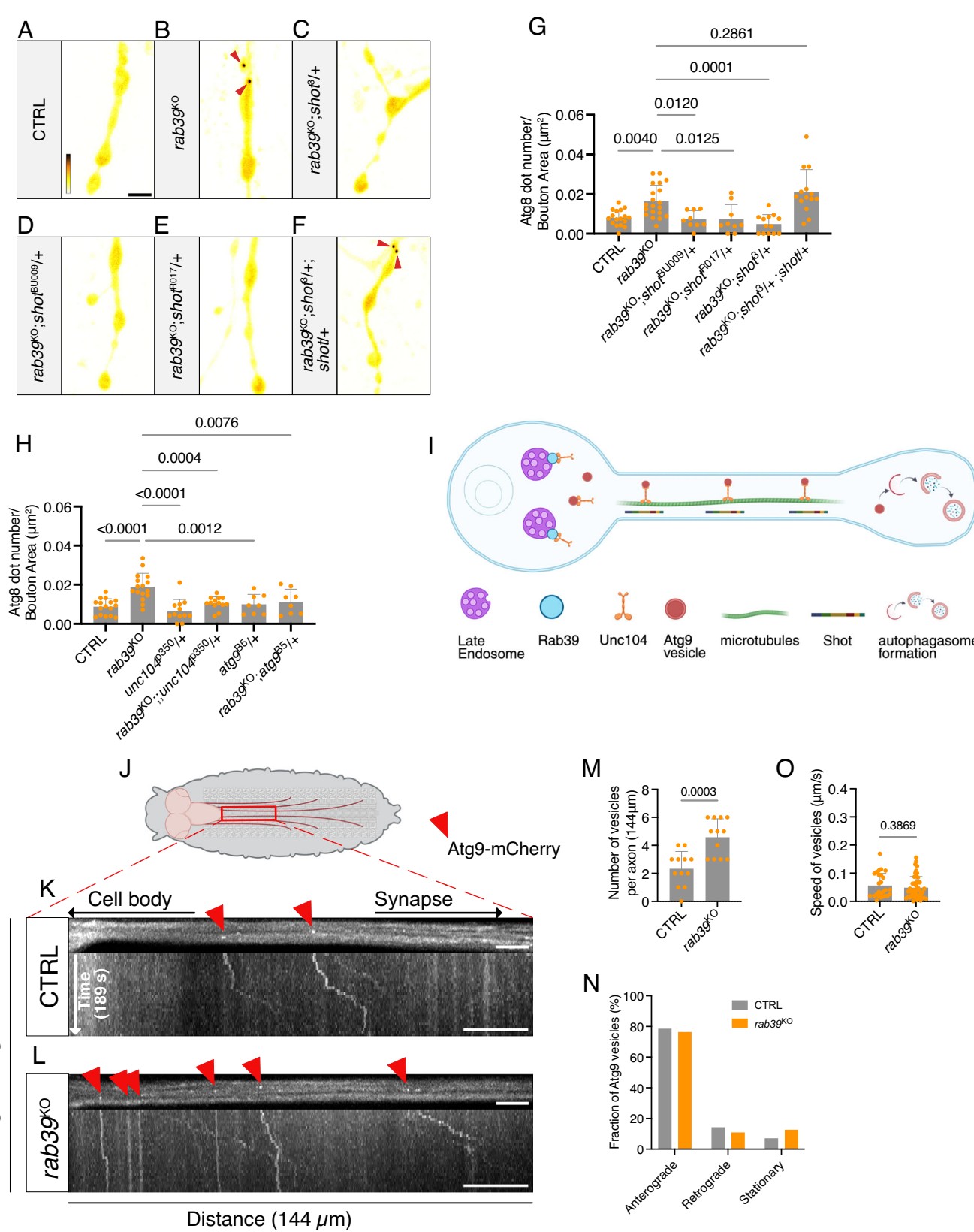

**Figure 7. Rab39 increases axonal Atg9 vesicle abundance and delivery to synapses.**

(A–F) Live imaging of Atg8-mCherry in NMJ boutons of control ($w^{1118}w^+$, CTRL) (A), $rab39^{KO}$ (B), $rab39^{KO}$; $shot^3$/+ (C), $rab39^{KO}$; $shot^{BU009}$/+ (D), $rab39^{KO}$; $shot^{R017}$/+ (E), and $rab39^{KO}$; $shot^{3}$/+; $shot$ + (F). Fluorescence intensities are shown using the gray value range indicated in (A) (365–3325). Arrowheads mark Atg8-mCherry-positive puncta. Scale bar: 5 µm. (G) Quantification of Atg8-mCherry puncta per bouton area from the experiment shown in (A–F). Statistical test: one-way ANOVA with Dunnett's post hoc test; $n = 9$–20; error bars: mean ± SD. (H) Quantification of Atg8-mCherry puncta per bouton area in control and $rab39^{KO}$ animals carrying null alleles of unc104 ($unc104^{p350}$) and atg9 ($atg9^{B5}$). CTRL- $rab39^{KO}$: exact $P = 1.532475 \times 10^{-5}$; $rab39^{KO}$ - $unc104^{p350}$: exact $P = 1.425731 \times 10^{-6}$. Statistical test: one-way ANOVA with Dunnett's post hoc test; $n = 8$–16; error bars: mean ± SD. (I) Schematic model illustrating the proposed role of Rab39 in regulating synaptic autophagy. (J) Schematic of the larval ventral nerve cord and axons in *Drosophila* third-instar larvae, indicating the regions used for Atg9 vesicle movement imaging. (K–L) Live imaging of control ($w^{1118}w^+$, CTRL) (K) and $rab39^{KO}$ (L) animals expressing genomic Atg9-mCherry in the $atg9^{B5}$/+ background. Scale bars: 5 µm (axon images); 10 µm (kymographs, *x* axis). Red arrowheads indicate Atg9-mCherry-positive puncta. (M) Quantification of the number of Atg9-mCherry puncta within a 144 µm segment of axon recorded for 189 s from the experiment in (K–L). Statistical test: unpaired *t* test; $n = 12$; error bars: mean ± SD. (N) Directionality distribution of all Atg9-mCherry puncta in control and $rab39^{KO}$ axons, respectively (from M). (O) Quantification of the speed (µm/s) of moving Atg9-mCherry puncta in axons from (K–L). Statistical test: Mann–Whitney test; $n = 26$–47; error bars: mean ± SD. Statistical test: Statistical test: Fisher's Exact Test with the Freeman–Halton extension; *P* value = 0.7282. Vesicle counts are also provided in Appendix Table S1. Source data are available online for this figure.

measure Atg9 delivery to presynaptic boutons. We photobleached Atg9-GFP expressed in motor neurons (*D42*-Gal4) at boutons proximal to the incoming motor axon and followed fluorescence recovery. This showed a faster recovery rate of Atg9-GFP fluorescence at $rab39^{KO}$ synaptic boutons compared to controls (Fig. 8C,D). Taken together, these data suggest increased Atg9 delivery (more axonal vesicles, faster FRAP) and turnover (by increased autophagy) in $rab39^{KO}$ mutants.

## Rab39 does not affect synaptic vesicle or mitochondrial transport

To determine if Rab39 also regulates the trafficking and delivery of components other than Atg9 vesicles to synapses, we conducted several experiments. First, we imaged synaptic and Atg9-positive vesicles, assessing overlap. We did not observe significant colocalization between Atg9-mCherry and Synaptotagmin-eGFP, labeling synaptic transport vesicles, in axons or neuronal cell bodies (Fig. EV4A–A'). This is consistent with previous findings indicating Atg9 vesicles are molecularly distinct from classical synaptic (transport) vesicles, leaving open the possibility their transport is independently regulated (Binotti et al, 2024).

Next, we assessed axonal transport of synaptic transport vesicles (Synaptotagmin-eGFP) and mitochondria (mitoGFP) in $rab39^{KO}$ and control axons using live imaging (Fig. EV4B–I). Given the high density of axonal Synaptotagmin-eGFP or mitoGFP labeled particles, we photobleached a portion of the axon and quantified the number and velocity of fluorescent particles moving into and through the bleached area. However, we did not find significant differences between $rab39^{KO}$ and control animals (Fig. EV4C–E,G–I), indicating that Rab39 loss does not globally affect axonal transport, but rather exerts cargo-specific effects on (at least) Atg9 vesicles. We do not exclude that Rab39 may also regulate other cargoes that we did not investigate here.

## Rab39 predominantly localizes to compartments of the endolysosomal pathway

Rab GTPases are key regulators of membrane trafficking and compartmentalization along the endolysosomal system (Wandinger-Ness and Zerial, 2014). Rab39a and Rab39b, mammalian orthologs of *rab39*, exhibit distinct intracellular localizations. Rab39a is found on late endosomes and LAMP2-positive vesicles

(Seto et al, 2013). Rab39b localizes to the endoplasmic reticulum/ *cis*-Golgi interface (Tudela et al, 2019). To investigate the subcellular localization of *Drosophila* Rab39 within neuronal somata, we examined its spatial relationship with different organelle markers under basal conditions (Fig. EV5). We thus analyzed the colocalization of Rab39 with markers for the late endosomes (Rab7 (Vitelli et al, 1997)) (Fig. EV5A–C), late endosomes-lysosomes (Lamp1 (Eskelinen et al, 2003)) (Fig. EV5D–F), cis-Golgi (GM130 (Nakamura et al, 1995)) (Fig. EV5G–I) and *trans*-Golgi (Syntaxin-16 (Tang et al, 1998)) (Fig. EV5J–L). Our analysis revealed a high degree of colocalization of Rab39 with Rab7 and Lamp1, with each pair showing ~40% Mander's coefficient (Fig. EV5C,F). In contrast, only a small fraction of Rab39 localized to the *cis*-Golgi and *trans*-Golgi compartments, as indicated by the low colocalization with GM130 (~20%) (Fig. EV5G–I) and Syntaxin-16 (~20%) (Fig. EV5J–L). This suggests that a larger fraction of the Rab39 pool is associated with the endolysosomal system as compared to the Golgi network.

## *shot* rescues dopaminergic neuron synapse loss in $rab39^{KO}$ mutants

Dopaminergic neuron synapse and cell loss are pathological hallmarks of PD that drive locomotor defects (Antony et al, 2013; Aggarwal et al, 2019). Similarly, $rab39^{KO}$ mutant flies exhibit a progressive decline in locomotor behavior and reduced synaptic innervation from protocerebral anterior medial (PAM) dopaminergic neurons (DAN) to the mushroom bodies (Kaempf et al, 2024b; Pech et al, 2024) (Fig. 9A). Since the heterozygous loss of *shot* (Fig. 7A–G) rescues the increased synaptic autophagy phenotype of $rab39^{KO}$ animals, we tested whether this also prevents synapse loss. DAN and their synapses were labeled using anti-tyrosine hydroxylase (TH), and we quantified afferent abundance (synapse density) within the mushroom body area labeled by anti-DLG. In young 5-day-old $rab39^{KO}$ flies and controls, the mushroom body was well-innervated by TH-positive PAM DAN (Fig. 9B–C',J); similarly, $shot^3$/+ heterozygous mutants or $rab39^{KO}$; $shot^3$/+ showed abundant innervation similar to controls (Fig. 9D–E',J). In contrast, 45-day-old $rab39^{KO}$ flies showed significantly less TH+ synaptic structures in the mushroom body area compared to controls (Fig. 9F–G',K) and (Kaempf et al, 2024b). This defect was rescued by introducing one mutant copy of $shot^3$ (Fig. 9H–I',K).

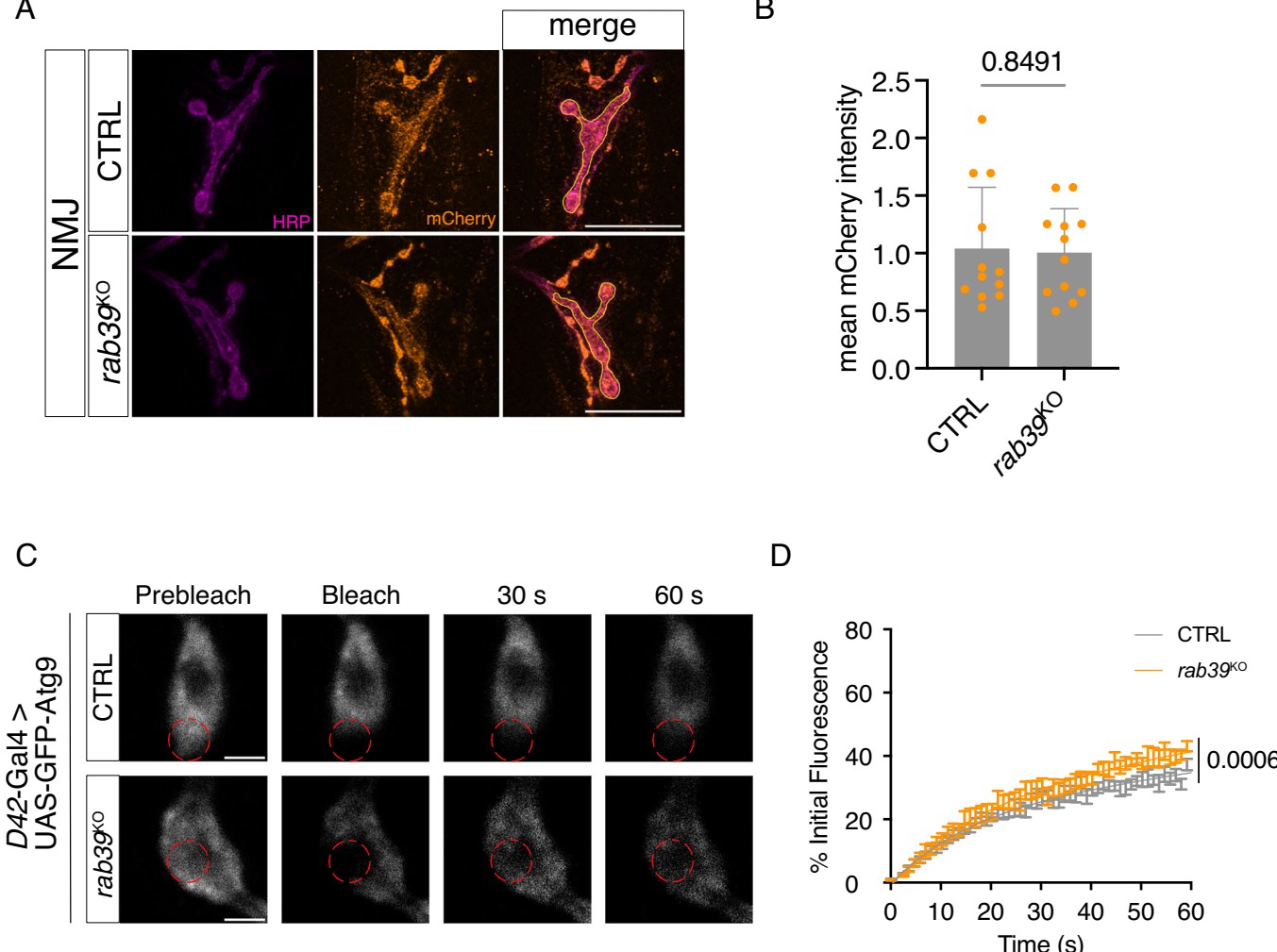

**Figure 8.** *rab39*[KO] increases Atg9 vesicle delivery to presynaptic terminals.

(A) Representative Airyscan confocal images of NMJs from fixed and immunostained third-instar larvae of control ($w^{1118}w^+$, CTRL) and *rab39*[KO] animals expressing genomically tagged Atg9-mCherry in *atg9*[B5] (null) background. Presynaptic boutons were labeled with anti-HRP (magenta), and Atg9-mCherry stained with anti-mCherry (orange). Scale bar: 20 µm. (B) Quantification of mean gray value of Atg9-mCherry fluorescence at NMJs, normalized to control levels. Statistical test: unpaired *t* test; $n = 10$; error bars: mean ± SD. (C) FRAP assay to assess Atg9-GFP mobility within presynaptic boutons. Control and Rab39[KO] larvae expressed UAS-*Atg9-GFP* under the control of the *D42*-Gal4 motor neuron driver. Representative images show fluorescence recovery after photobleaching of a small region (dotted circle) within an NMJ bouton over a 60 s period. Scale bar: 5 µm. (D) Fluorescence recovery curves of Atg9-GFP signal over time in control and Rab39[KO] boutons, fit with a double-exponential model. Statistical test: two-way ANOVA; $n = 10$; error bars: mean ± SD. Source data are available online for this figure.

Hence, the loss of the axonal trafficking factor *shot* not only rescues the increased synaptic autophagy in *rab39*[KO] mutants, but it also prevents the progressive denervation of DAN in the brain of adult *rab39*[KO] flies.

## Discussion

Here, we identify a soma-to-synapse regulatory mechanism by which Rab39 modulates synaptic autophagy.

Specifically, Rab39, which localizes to the neuronal soma, acts as a negative regulator of Atg9 vesicle transport towards synaptic terminals. In *rab39*[KO] animals, we observed increased anterograde trafficking of Atg9-carrying vesicles, increased delivery to synapses, correlating with elevated autophagosome formation at presynaptic terminals and enhanced retrograde transport of Atg8-positive vesicles. This suggests that loss of Rab39 promotes synaptic autophagy by increasing the supply of Atg9 to presynaptic sites. Supporting this model, the increased synaptic autophagy phenotype in *rab39*[KO] larvae is suppressed by lowering Atg9 levels. Moreover, expression of *EndoA*[DA], a dominant-negative mutant that blocks synaptic autophagy, also abolishes the increased synaptic autophagy phenotype of *rab39*[KO] mutants, suggesting that Rab39 and Atg9 converge onto EndoA-dependent synaptic autophagy. While our data support this possibility, and our (unbiased) genetic screen pointed strongly to trafficking and synapse organization pathways, we cannot rule out that Rab39 also has parallel roles in inhibiting synaptic autophagy pathways not uncovered by this work.

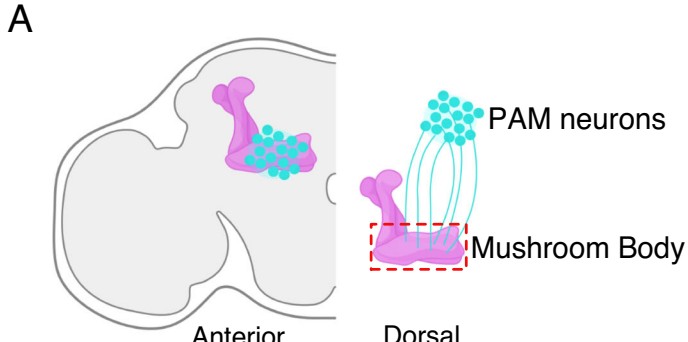

PAM neurons

Mushroom Body

Anterior    Dorsal

5 days old

45 days old

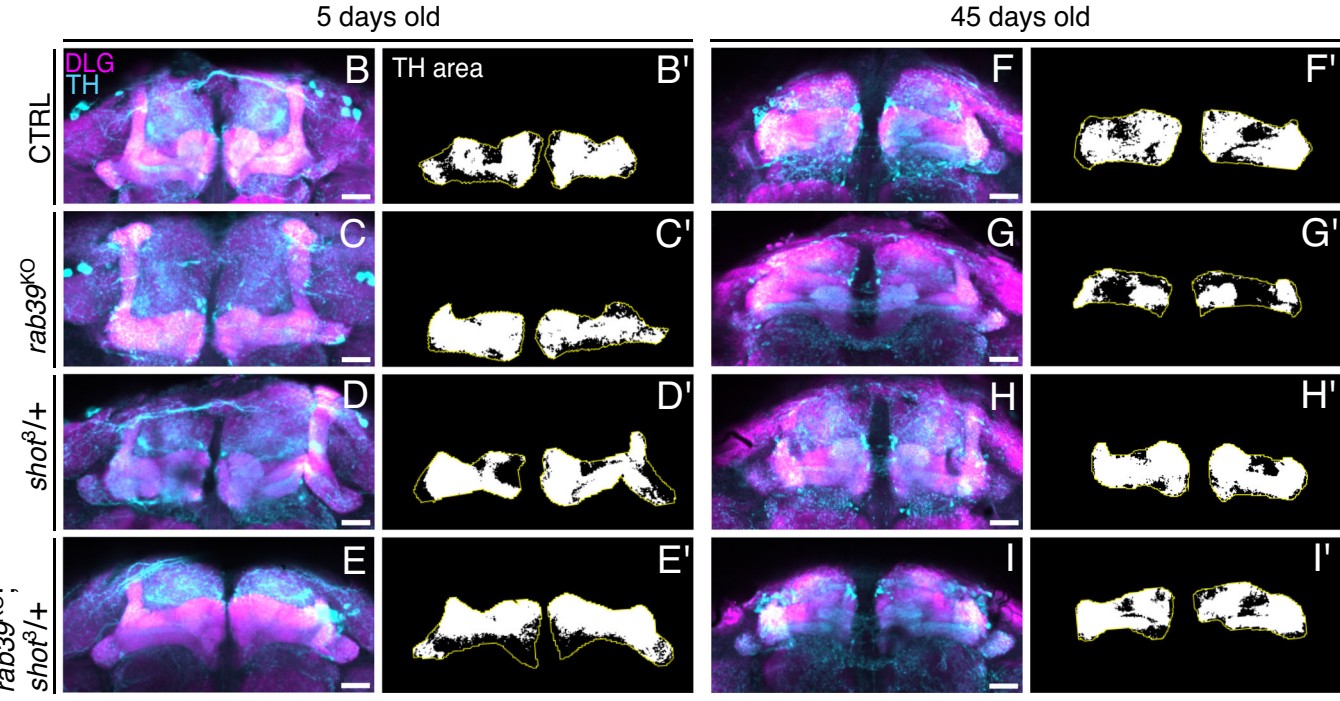

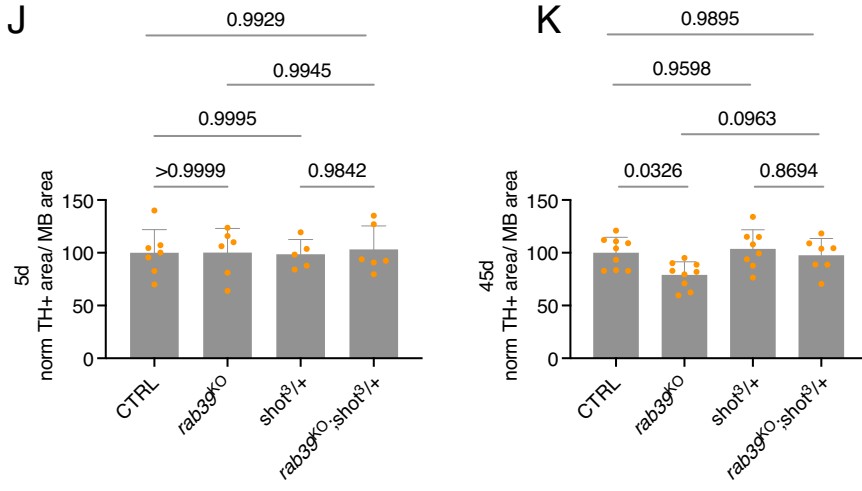

**Figure 9.  Reducing *shot* levels partially rescues the loss of dopaminergic neuron synapse area in *rab39*[KO].**

(A) Schematic overview of dopaminergic protocerebral anterior medial (PAM) neurons and the mushroom body in anterior (left) and dorsal (right) views of the *Drosophila* adult brain. The red dotted area indicates the region within the mushroom body where dopaminergic innervation was analyzed. (B–I') Maximum intensity projections of confocal images of the mushroom body in the anterior brain of 5-day-old (B–E') and 45-day-old (F–I') flies. Tissues were labeled with anti-Tyrosine Hydroxylase (TH, blue) to mark dopaminergic neurons and anti-DLG (magenta) to define synaptic regions. (B'–E', F'–I') TH-immunoreactive area (white) innervating the mushroom body lobes (yellow region of interest, ROI) was derived from the TH signal in (B–E, F–I). The DLG-positive area was used to define the mushroom body area (yellow outline), and the TH-positive area was quantified as "TH[+] area." Scale bar: 20 µm. (J) Quantification of TH-immunoreactive area within the mushroom body lobes of 5-day-old flies from (B–E'). Statistical test: one-way ANOVA with Tukey's multiple comparison test; $n = 5–7$; error bars: mean ± SD. (K) Quantification of TH-immunoreactive area within the mushroom body lobes of 45-day-old flies from (F–I'). Statistical test: one-way ANOVA with Tukey's multiple comparison test; $n = 7–9$; error bars: mean ± SD. Source data are available online for this figure.

While synaptic autophagy is acutely governed by local regulators (e.g., BSN, EndoA, and Synj1) (Okerlund et al, 2017; Vanhauwaert et al, 2017; Soukup et al, 2016; Nikoletopoulou and Tavernarakis, 2018) our results reveal a soma-centered mechanism that, in a more chronic way, modulates the supply of autophagy machinery to the NMJ synapse. Our data indicate that GTP-bound Rab39 restricts synaptic autophagosome biogenesis by inhibiting either the formation or anterograde transport of Atg9-positive vesicles. When Rab39 is inactivated, e.g., through pathological mutations or when in GDP-bound state, this suppression is lifted, resulting in enhanced Atg9 delivery to synaptic terminals and increased presynaptic autophagosome formation.

Building on this model, we propose that Rab39 acts as a regulator of basal, EndoA-dependent synaptic autophagy. However, its potential role in acute autophagy induction, such as during sustained neuronal activity or cell stress, remains to be investigated. Importantly, synaptic and somatic autophagy differ not only in location but also likely in cargo specificity, dynamics, and regulatory mechanisms. This highlights the need for side-by-side studies that clearly tell these different autophagy routes apart, so we do not conflate how they are controlled or what their function is.

As Rab39 localizes predominantly to the neuronal soma, a central question arises: how does it modulate local synaptic autophagy? Our genetic screen provides insight by identifying microtubule-associated transport machinery components as key modulators of the *rab39* increased autophagy phenotype. Specifically, loss-of-function mutations in the microtubule regulators *shot* and *unc104* suppress elevated synaptic autophagy in *rab39* mutants. Shot promotes microtubule stability and axonal growth through interactions with EB1 and could potentially function at the AIS to restrict Atg9 vesicle entry into axons (Alves-Silva et al, 2012; Leterrier et al, 2011; Poliakova et al, 2014). In contrast, the kinesin motor protein Unc104 directly mediates anterograde transport. Shot and Unc104 may act at distinct steps—Shot at the level of cargo gating and Unc104 during cargo movement—within a shared Rab39-dependent axis that ensures selective vesicle delivery to the synapse. More work is now needed to test this model and elucidate the exact mode of action.

Although Unc104/Kif1A is a broadly acting kinesin motor (Hall and Hedgecock, 1991; Kern et al, 2013; Pack-Chung et al, 2007), Atg9 trafficking is selectively enhanced in *rab39*[KO] mutants, while transport of other cargoes like synaptotagmin (a known Kif1A cargo (Okada et al, 1995)) and mitochondria (primarily transported by other kinesins like Kif1B (Nangaku et al, 1994)) remain unaffected. This cargo specificity suggests that Rab39 does not generally impair axonal transport, but rather selectively modulates Kif1A cargo prioritization. There are different possibilities as to

how Rab39-GTP may achieve this. Rab39 could direct Kif1A toward other cargos or alternatively, it could increase the formation of Atg9-positive vesicles in the soma, thus altering cargo availability. Rab39-GTP may also direct Kif1A towards lysosomes for degradation, thereby limiting its availability for Atg9 vesicle transport, but in this scenario, other trafficking events are expected to be affected. When Rab39 is GDP-bound or absent, as in our knockout animals, these constraints are lifted, allowing KiF1A to engage more readily with Atg9 vesicles and enhance their delivery to synaptic terminals. The adapter Prd1 may potentially also contribute to cargo selectivity (Gillingham et al, 2014; Hummel and Hoogenraad, 2021). Consistent with our model, Kif1A cargo specificity is known to be regulated by adapter proteins and post-translational modifications. For example, MADD/Rab3GEP links Kif1A to synaptic vesicles, while Arl8A mediates Kif1A association with dense-core vesicles and lysosomes. Moreover, phosphorylation of Kif1A and motor dimerization further contribute to cargo selection (Hummel and Hoogenraad, 2021). However, how Rab39 intersects with such known mechanisms of motor regulation, such as adapter-mediated cargo selection, remains to be determined. Uncovering how Rab39 influences cargo prioritization and motor allocation could shed light on broader principles of selective vesicle trafficking for autophagy in neurons. Moreover, it remains to be determined whether this regulatory axis is specific to Atg9 vesicles or extends to additional cargo types that we did not test in this work.

At baseline conditions, autophagosomes are formed at synaptic terminals and undergo dynein-dependent retrograde transport to the soma, maturing through fusion with late endosomes and lysosomes en route (Maday et al, 2012; Neisch et al, 2017; Stavoe and Holzbaur, 2019; Hill and Colón-Ramos, 2020). Efficient retrograde clearance is essential as impaired transport causes autophagosomes to accumulate at presynaptic terminals (Neisch et al, 2017; Kargbo-Hill et al, 2019). However, the increased number of synaptic autophagosomes that we observed in rab39 mutants is unlikely due to impaired retrograde trafficking. Live imaging in *rab39*[KO] axons revealed elevated numbers of retrogradely transported autophagosomes, with transport velocity comparable to controls, supporting a model in which autophagosome formation itself is upregulated at presynaptic NMJ terminals. Despite advances in understanding retrograde autophagosome transport (Hollenbeck, 1993; Maday et al, 2012; Kononenko et al, 2017), two fundamental questions remain. First, why is autophagosome biogenesis spatially enriched at synaptic terminals (Maday et al, 2012)? This suggests that essential molecular components for autophagosome initiation are selectively enriched at distal sites, despite the much larger volume of the axon. Second, how is this

spatial bias regulated by the soma? These questions are likely connected: the targeted delivery of key autophagy components, such as Atg9 from the soma to synaptic terminals—e.g., via Unc104 (Stavoe et al, 2016)—may establish spatial restriction. Our data support this model: loss of either Unc104 or Atg9 in *rab39*[KO] mutants suppresses the increased synaptic autophagy phenotype, indicating that Rab39 modulates autophagosome formation by controlling the soma-to-synapse trafficking of required components. This does not exclude that Rab39 also inhibits synaptic autophagosome formation by acting on other pathways, but those did not emerge from our unbiased genetic screen. Together, our findings and published work indicate that the soma coordinates both outbound trafficking of autophagy machinery components and the inbound return of autophagosomes, thereby regulating the full autophagic cycle at synapses.

Despite species-specific nuances in autophagy regulation, such as varying responses to starvation, the core autophagy machinery is highly conserved between flies and mammals (Nezis, 2012; Mukherjee et al, 2016; Maday and Holzbaur, 2014; Demir and Kacew, 2023) and also more specific regulators appear to act across species. Altered synaptic autophagy has been observed in both *Drosophila* and human models of PD (Decet and Verstreken, 2021; Nachman and Verstreken, 2022). For example, the pathogenic *Synj1*[R258Q] mutant and PD-risk variant in *EndoA1/SH3GL2* reduce synaptic autophagy levels both in flies and human induced neurons (Soukup et al, 2016; Vanhauwaert et al, 2017; Bademosi et al, 2023), underscoring the relevance of fly models for studying synaptic autophagy in PD. Interestingly, EndoA and Synj1 are strong binding partners at synapses (Ringstad et al, 1997). However, there are also important differences. For example, flies do not express alpha-synuclein, a protein that critically accumulates in Lewy bodies in most PD cases (Goedert et al, 2017). Likewise, RAB39B mutant patients also show extensive Lewy bodies (Wilson et al, 2014; Gao et al, 2020a) and -interestingly- increased alpha-synuclein levels correlate with decreased EndoA1 levels (Westphal and Chandra, 2013). While we were not able to investigate the link between defective synaptic autophagy and alpha-synuclein accumulation, our work in *Drosophila* does show the functional cooperation between EndoA and Rab39, adding to the list of proteins that all act in a PD-relevant pathway that orchestrates synaptic autophagy (Uytterhoeven et al, 2025).

The impact of dysregulated synaptic autophagy on neuronal health remains incompletely understood. Increased synaptic autophagy drives the turnover of autophagy-cargo proteins including nSyb as we show here (Kallergi et al, 2023; Goldsmith et al, 2022), but at the same time, overall protein levels of autophagy-cargo proteins at synapses remain constant. While we did not measure post-translational modifications and protein oxidation in the mutants, the overall functional consequence of this is unclear. We do find that *rab39*[KO] mutants display defects in neurotransmitter release, and both excessive and insufficient autophagy caused synaptic degeneration in dopaminergic neurons. This suggests the process must be finely tuned to maintain neuronal proteostasis. One compelling hypothesis is that imbalances in autophagy-mediated protein turnover disrupt sensitive signaling networks (Uytterhoeven et al, 2025). Future studies should aim to identify the specific autophagy substrates and regulatory nodes that critically contribute to synaptic dysfunction and promote neurodegeneration. Such mechanistic insights could pave the way to

therapeutic strategies that target these critical substrates and restore the pathways that were deregulated by autophagic balance disturbance in PD.

While our data strongly support a model whereby Rab39 mediates soma-to-synapse control of EndoA-dependent synaptic autophagy, several questions remain. First, the precise synaptic substrates targeted during Rab39-regulated autophagy have yet to be defined. Second, although Atg9 trafficking emerges as a key mediator in this pathway, the molecular mechanisms by which Rab39 regulates Atg9 vesicle biogenesis or selective transport remain unknown. Third, it is conceivable that increased Atg9 delivery to synapses contributes to the increased synaptic autophagy in *rab39* mutants, but it may also not be sufficient, and Rab39 could also inhibit other processes that drive synaptic autophagy. Finally, while our study uses *Drosophila* as a model system, it will be important to extend these findings to human dopaminergic (and other) neurons, particularly those carrying the PD-associated *RAB39B*[T168K] mutation.

# Methods

**Reagents and tools table**

| Reagent/resource | Reference or source | Identifier or catalog number |
|---|---|---|
| **Fly lines used in this study** | See Appendix Table S2 | |
| **Antibodies used in this study** | See Appendix Table S3 | |
| **Chemicals, enzymes and other reagents** | | |
| Ethyl methanesulfonate (EMS) | Sigma-Aldrich | Cat#: M0880 |
| Paraformaldehyde | Sigma-Aldrich | Cat#: 252549 |
| Bouin's fixative | Sigma-Aldrich | Cat#: HT10132 |
| Vectashield Mounting Medium | Vector Laboratories | Cat#: H-1000 |
| RapiClear 1.47 | SUNJin Lab | Cat#: RC147002 |
| ProLong Diamond Antifade Mountant | Thermo Fisher | Cat#: P36965 |
| DAPI | Sigma-Aldrich | Cat#: D9542 |
| Triton X-100 | Sigma-Aldrich | Cat#: X100RS |
| SuperScript III First-Strand Synthesis System | Thermo Fisher Thermo Fisher | Cat#: 18080051 |
| Maxwell RSC microRNA tissue kit | Promega | Cat#: AS1460 |
| **Software** | | |
| ZEN Blue | Carl Zeiss Microscopy GmbH | version 2.3 lite |
| ZEN Black | Carl Zeiss Microscopy GmbH | version 3.7 |
| NIS-Elements | Nikon Instruments Europe B.V. | version 5.42.06 |
| pCLAMP10 | Molecular Devices | version 10.7 |
| IGOR Pro | WaveMetrics | version 6.37 |
| ImageJ | NIH | version 1.14.0/1.54 f |
| GraphPad Prism | GraphPad Software | version 9 |

| Reagent/resource | Reference or source | Identifier or catalog number |
|---|---|---|
| R Studio | RStudio, PBC / Posit | version 2024.09.1 + 394 |
| GIMP | The GIMP Development Team | version 2.10.30 |
| Arivis Vision4D | Arivis AG | version 4.12 |
| Affinity designer | Serif Europe Ltd. | version 1.10.8 |

## Fly husbandry

Drosophila stocks (see Appendix Table S2) were maintained under standard conditions and fed a conventional diet composed of cornmeal, agar, yeast, sucrose, and dextrose. Experimental crosses were raised at 25 °C. Wandering third-instar (L3) larvae of the appropriate genotypes were selected for immunostaining, autophagy assays, tandem Atg8-GFP-mCherry imaging, fluorescent timer experiments, vesicle and mitochondrial transport analysis, and FRAP experiments. For electroretinogram (ERG) recordings, 3-day-old adult flies were either kept under constant light or in complete darkness as control. For aging experiments, newly eclosed adult flies of the indicated genotypes were collected and aged in groups of 10–20 at 25 °C for either 5 or 45 days, with transfers to fresh food vials every 3 days.

## Ethyl methanesulfonate (EMS) mutagenesis

Mutagenesis was performed on isogenized cn bwiso males that were fed 15 nM EMS in a sucrose solution. After recovery from mutagenesis, these males were mated with endoA26 null virgin females. In the F1 generation, cn bw mut*; endoA26 null (mut* indicated the EMS-induced mutation) virgins were collected and crossed with endoASD; endoAΔ4null males to conduct ERG phenotype screening. 4403 lines carrying lethal mutations were balanced and kept, the remaining stocks were discarded.

## Light-induced neurodegeneration and ERGs

All ERG experiments were conducted with male flies in a cn bw background; these flies have white eyes. For ERG recordings, flies were collected 1 day after eclosion and kept in the incubator at 25 °C until day 3. Three-day-old male flies were transferred to an incubator where they were exposed to constant illumination (1300 LUX, 24 h/day) at 25 °C. endoA26 mutant and EMS-generated mutants used for ERG screening were kept under constant light for 3 days, while PD knockout mutants, including rab39KO, lrrkKO and synjKO were maintained under continuous light exposure for 7 days. For the ERG measurements, flies were immobilized on glass microscope slides with double-sided tape. Filamented glass micropipettes were used to generate electrodes using the Laser-pipette-puller P-2000 (Sutter Instrument). Electrodes were filled with a 3 M NaCl solution. The recording electrode was placed lightly onto the ommatidia. The reference electrode was inserted into the thorax. The flies were exposed to darkness for 3 s, followed by 1 s of LED light illumination. This was repeated five times for each fly. Light-evoked signals were amplified by a DC amplifier, and the amplified signal was processed by a data acquisition device (Clampex), connected to a PC running Clampfit software. ERG traces were analyzed in IGOR Pro 6.37 using a custom-made macro (Kaempf et al, 2024).

## Next-generation sequencing and candidate gene selection

Next-generation sequencing of flies was performed at the VIB Nucleomics Core as described in the Illumina NebNEXT DNA Ultra protocol (Gautreau, 2015). The GATK4 (v4.0.9.0) Germline short variant discovery (SNPs + Indels) workflow (Poplin et al, 2017) was adapted to the Drosophila genome and applied to the Illumina paired read data. Paired reads were aligned to the Drosophila reference genome (Dm_BDGP6) (Hoskins et al, 2015). Known variants were obtained from NCBI (Sherry et al, 2001) and FlyVar (F Wang et al, 2015). The final VCF data was annotated and filtered with SnpEff (v4.3) (Cingolani et al, 2012), followed by additional internal filtering logic to prioritize candidate mutations.

As part of this internal logic, we implemented two additional filtering steps to reduce false positives, collectively referred to as the ΔendoA screen. First, we excluded recurrent variants found in multiple mutant lines, which are likely background-specific SNVs introduced after isogenization and unlikely to be causative of lethality. Second, we removed genes that consistently carried an unusually high number of SNVs across sequenced lines, as these are prone to mapping artifacts or non-specific variation. Variants removed by the DendoA screen were excluded from downstream candidate lists.

FlyBase gene ontology terms were used for gene ontology analysis (Jenkins et al, 2022).

### Quantitative RT-PCR

Total RNA was extracted from 10 adult Drosophila heads (5 males and 5 females, 5 days old) per sample using the Maxwell RSC Instrument and Maxwell RSC RNA Kit (Promega), according to the manufacturer's instructions. First-strand cDNA synthesis was performed using the SuperScript III First-Strand Synthesis System (Thermo Fisher Scientific). Quantitative PCR (qPCR) was conducted on a LightCycler 480 system (Roche) using the 480 SYBR Green I Master Mix (Roche). Relative gene expression levels were calculated using the $\Delta\Delta Ct$ method, with Ct values normalized to the reference gene rp49. Expression levels were then expressed as a percentage relative to control samples, which were set to 100%.

Primers used for rab39 quantification:

- Forward (F1_qPCR_Rab39): 5′-GCTCGCTGCTCAAATTCTTC-3′
- Reverse (R1_qPCR_Rab39): 5′-TGTGTGCCGTCCTTCATT-3′

Primers used for rp49 quantification:

- Forward (F1_qPCR_Rp49): 5′-ATCGGTTACGGATCGAACAA-3′
- Reverse (R1_qPCR_Rp49): 5′-GACAATCTCCTTGCGCTTCT-3′.

### Immunohistochemistry

For immunostaining of HRP, BRP, and CSP at the neuromuscular junction (NMJ), third-instar larval fillets were fixed in 3.7%

paraformaldehyde in HL3 (100 mM NaCl, 5 mM KCl, 10 mM NaHCO$_3$, 5 mM HEPES, 30 mM sucrose, 5 mM trehalose, 10 mM MgCl$_2$, pH 7.2) for 20 min. Samples were permeabilized with 0.4% PBX (0.4% Triton X-100 in 1× PBS) four times for 15 min on a shaker, blocked for 1 h in 10% normal goat serum in PBX, and incubated overnight at 4 °C with primary antibodies. After washing with 0.4% PBX, fillets were incubated with secondary antibodies for 90 min at room temperature. Samples were mounted in Vectashield (Vector Laboratories).

For immunostaining of Syntaxin at the NMJ, and for imaging eYFP-Rab39 localization in the ventral nerve cord (VNC) and NMJ, as well as Rab7, Lamp1, Syntaxin-16, GM130, and Atg9-mCherry, larval fillets were fixed in Bouin's fixative (Sigma-Aldrich) for 10 min. Fillets were permeabilized with 0.05% PBX (0.05% Triton X-100 in 1× PBS) four times for 15 min on a shaker, blocked for 1 h in 5% normal goat serum in PBX, and incubated overnight at 4 °C with primary antibodies. After several washes, secondary antibodies were applied overnight at 4 °C. Samples were mounted in RapiClear 1.47 (SUNJin Lab) on high-performance 1.5H coverslips (Marienfeld) for imaging on the Nikon Ti2 N-SIM S microscope, in ProLong Diamond (Invitrogen) for the Zeiss Elyra 7 microscope, and in Vectashield (Vector Laboratories) for the Nikon NiE A1R and Zeiss LSM900 microscopes.

For immunohistochemistry of adult brain samples, brains of 5- and 45-day-old animals were dissected in ice-cold PBS and fixed for 30 min in a freshly prepared 3.7% paraformaldehyde solution containing 1× PBS and 0.2% Triton X-100 (PBX) at room temperature (RT). After three 15 min washes in PBX at RT on a shaker, brains were placed in a blocking solution (PBX with 10% normal goat serum (NGS)) for 1 h at RT. The samples were then incubated with primary antibodies in blocking solution at 4 °C for 1.5–2 days, followed by three 15 min washes in PBX at RT. Secondary antibodies in PBX with 10% NGS were applied overnight at 4 °C. Afterward, brains were washed three times for 15 min in PBT at RT on a shaker and then mounted anterior-side up in RapiClear 1.47 (SUNJin Lab).

The following primary antibodies were used: rabbit anti-HRP (1:2000), Alexa Fluor 647-conjugated goat anti-HRP (1:100), mouse anti-DLG (1:50 or 1:100), mouse anti-Brp (nc82; 1:250), mouse anti-CSP (AB49; 1:500), mouse anti-Syntaxin-1A (8C3; 1:20), mouse anti-Rab7 (1:50), rabbit anti-Lamp1 (1:500), rabbit anti-GM130 (1:500), rabbit anti-Syntaxin-16 (1:500), chicken anti-GFP (1:500), rabbit anti-mCherry (1:500), rat anti-Elav (1:500), and rabbit anti-TH (1:200). Secondary Alexa Fluor-conjugated antibodies (Invitrogen) were used at a dilution of 1:500 for confocal microscopy and 1:250 for super-resolution imaging. These included Alexa Fluor 488 goat anti-rabbit, Alexa Fluor 488 goat anti-chicken, Alexa Fluor 488 goat anti-mouse IgG1, Alexa Fluor 555 goat anti-rabbit, Alexa Fluor 555 goat anti-mouse IgG2a, Alexa Fluor 555 goat anti-rat, Alexa Fluor 647 goat anti-mouse, and Alexa Fluor 647 goat anti-rat. Antibodies used in this study, along with their working concentrations, are listed in Appendix Table S3.

### Confocal live imaging and quantification

For analysis of synaptic Atg8 puncta, third-instar larvae expressing Atg8-mCherry were dissected in fresh HL3 solution (100 mM NaCl, 5 mM KCl, 10 mM NaHCO$_3$, 5 mM HEPES, 30 mM sucrose, 5 mM trehalose, 10 mM MgCl$_2$, pH 7.2), rinsed thoroughly, and mounted for live imaging. Imaging was performed using a Nikon NiE A1R

confocal microscope equipped with a NIR Apo ×60 1.0 W DIC N2 water-dipping objective (NA 1.0). Z-stacks covering the entire NMJ were collected. Atg8-mCherry–positive puncta within synaptic boutons were manually counted in ImageJ after thresholding to define bouton regions. Puncta outside the NMJ were excluded from analysis. Four NMJs were imaged per animal, and the average of four NMJs was used as a single data point.

For synaptic vesicle turnover analysis, third-instar larvae expressing FT::nSyb were dissected in HL3 and imaged under the same conditions. Z-stacks (21 slices, 0.45-µm step size) were collected in blue (405 nm excitation) and red (561 nm excitation) channels. For ratiometric analysis, bouton intensities were quantified in both channels using ImageJ. Ratio images were generated in Fiji by converting both channels to 32-bit, applying a threshold to retain the bouton signal, dividing the red channel by the blue channel, and applying a look-up table. Background regions were excluded. Four NMJs were imaged per animal, and the average of four NMJs was used as a data point.

For autophagosome acidification measurements, third-instar larvae expressing UAS-Atg8-mCherry-GFP were dissected in HL3 and imaged as above. Z-stacks (20 slices, 0.43-µm step size) were collected in green (488 nm excitation) and red (561 nm excitation) channels. For quantification, integrated densities of both channels were measured within bouton regions in ImageJ, and red/green ratios were calculated. Two NMJs were imaged per animal, and the average of two NMJs was used as a single data point.

### Confocal imaging and quantification

For adult brain imaging, Z-stacks were acquired using a Nikon NiE A1R confocal microscope equipped with a Plan Apo VC 20X DIC N2 water immersion objective (NA 0.95). Brains were imaged with a 3 µm Z-step, using identical settings across genotypes. Image analysis was performed in Fiji. The mushroom body (MB) region was identified in five consecutive Z-planes based on anti-DLG staining, and a region of interest (ROI) was defined using a sum projection. Within this ROI, anti-TH fluorescence was thresholded using a fixed threshold consistent with the control, and the total TH-positive area across the five Z-planes was measured and normalized to the MB area (TH+ area/MB area). These values were normalized to the control mean per experiment. For display, maximum projections of five Z-planes and a thresholded middle Z-plane are shown for aged fly brains.

For analysis of Atg8-positive puncta in larval neuronal somata, third-instar larvae expressing Atg8-mCherry were dissected in HL3. The ventral nerve cords were mounted in HL3 between two stacked bookbinder rings on a slide and covered with a coverslip. Imaging was performed using a Nikon NiE A1R confocal microscope equipped with a NIR Apo 60×1.4 Oil DIC N2 objective (NA 1.4). Z-stacks (40 slices, 0.45-µm step size) were acquired. Somata were outlined in Fiji using the Freehand tool, and the total area was measured. Images were thresholded to identify Atg8-positive puncta, which were counted using the Analyze Particles tool. The number of puncta was normalized to soma area.

For quantification of synaptic protein levels, third-instar larvae were fixed and stained with antibodies against BRP, Syntaxin, and CSP. Imaging was performed using a Nikon NiE A1R confocal microscope equipped with a NIR Apo 60 × 1.20 water immersion objective (NA 1.2). Z-stacks (20 slices, 0.45 µm step size) were acquired. Image analysis was conducted in Fiji. Synaptic boutons

were defined using anti-HRP staining, and the mean gray intensity of each synaptic protein was measured. Two NMJs were imaged per animal.

To examine the subcellular localization of Rab39 in neuronal somata and neuromuscular junctions (NMJs), and to quantify Atg9 levels at NMJs, confocal images were acquired using a Zeiss LSM900 microscope equipped with Airyscan2 and a 63× Plan Apochromat oil immersion objective (NA 1.40). The setup was controlled by ZEN Blue software (version 2.3 lite, Carl Zeiss Microscopy GmbH).

Fluorescent signals were detected using secondary antibodies conjugated to Alexa Fluor 488, Alexa Fluor 555, and Alexa Fluor 647, excited with 488 nm, 561 nm, and 647 nm laser lines, respectively. Emission was collected using appropriate filters (e.g., 525/50 for Alexa 488, 595/50 for Alexa 555, and 680/30 for Alexa 647).

Z-stack images (20 slices covering a total depth of 9.5 μm) were acquired and processed using the Airyscan workflow. Image analysis was performed in Fiji. Regions of interest corresponding to the soma or NMJ were delineated using the HRP signal, and the mean gray value was measured for the Alexa Fluor 488 (Rab39) or Alexa Fluor 555 (Atg9) channel. For Rab39 quantification, both soma and NMJ regions were analyzed; for Atg9 intensity measurements, only NMJs were analyzed. Two NMJs were imaged per animal, and the average was used as a single data point.

### Spinning disc imaging and quantification

For the tracking of Atg8 vesicles, live imaging was performed in axons immediately adjacent to type Ib boutons between muscle 12 and 13 in dissected third-instar larvae mounted in fresh HL3 solution. For image acquisition, we used a Nikon NiE upright microscope (Nikon Instruments B.V.) equipped with a Yokogawa CSU-X spinning-disk module coupled to a Prime BSI camera (Teledyne Photometrics) in combination with a Nikon NIR Apo 60× W DIC N2 water-dipping objective (NA 1.0). The setup was controlled by NIS-Elements (version 5.42.06, Nikon Instruments Europe B.V.). mCherry was excited with a 561 nm laser line and collected with a 590/50 nm emission filter. Time-lapse series were acquired with no delay between frames, over a total duration of 3 min, and included 9 Z-stacks with a step size of 1 μm. The images were post-processed in batch using a GA3 protocol to denoise and create an EDF image of each Z-stack. Also, the intensity was equalized over time and the time lapse was aligned. For quantification, Atg8-mCherry-positive puncta were manually traced using the Manual Tracking tool in Fiji/ImageJ and normalized to axon length.

### Atg9 vesicle tracking

Atg9 vesicle trafficking was analyzed in dissected third-instar larvae prepared in HL3, with axons kept intact between the VNC and NMJs, and mounted in glass-bottom MatTek dishes for immediate imaging. For image acquisition, a Zeiss LSM980 Airyscan inverted laser scanning confocal microscope was used in combination with a Zeiss 20× C Epiplan Apochromat Air objective (NA 0.70). The setup was controlled by ZEN Blue (version 2.3, Carl Zeiss Microscopy GmbH). Confocality was set to Airyscan super-resolution mode (confocal pinhole setting not applicable) and the sampling rate was 0.09 μm/pixel. Atg9-mCherry was excited with a 561 nm laser line and collected with a LP568 nm emission filter. For post-processing and image analysis, ZEN Blue (version 2.3 lite, Carl Zeiss Microscopy GmbH) was used. The images were acquired as a time-lapse series over 189 s with 30 time points and 7 Z-stacks per

frame (total depth: 3.66 μm) and processed using the default Airyscan workflow. Atg9-positive vesicles were manually tracked in ImageJ using the "Manual Tracking" tool. Vesicles with a total displacement below 4 μm were classified as stationary.

### Super-resolution microscopy and colocalization analysis

Colocalization analysis was performed using a Zeiss Elyra 7 microscope equipped with Lattice SIM (Carl Zeiss Microscopy GmbH), two PCO.edge 4.2 CLHS sCMOS cameras (PCO AG), and a Zeiss ×60 Plan Apochromat oil immersion objective (NA 1.4). The setup was controlled by ZEN Black (version 3.7, Carl Zeiss Microscopy GmbH). Images were acquired at a Nyquist sampling rate of 0.06 μm × 0.06 μm × 0.11 μm ($X \times Y \times Z$). Alexa Fluor 488 (Rab39) was excited with a 488 nm laser and emission collected at 515 nm. Alexa Fluor 555 (Lamp1, GM130, Syntaxin-16) and Alexa Fluor 647 (Rab7) were excited with 561 nm and 642 nm lasers, respectively, with emission collected above 590 nm for both. Images were reconstructed using the 3D SIM algorithm in ZEN Black, applying a sharpness filter of 9.2 for Rab7 and Lamp1, and using a value of 8 for Syntaxin-16 and GM130, always keeping the original dynamic range. For analysis, all channels were pre-processed in ImageJ by applying background subtraction using the Rolling Ball algorithm (radius = 5 pixels), followed by a Gaussian blur (sigma = 2). Three Z-stacks from the appropriate focal plane for soma localization were selected per sample. Thresholding was used to segment organelles of interest. Two somata were analyzed per animal, and their average was used as a single data point. Colocalization was assessed using the JACoP plugin in ImageJ, with statistical significance evaluated by Costes' randomization test using over 100 iterations.

### FRAP assay of Atg9 mobility

Fluorescence recovery after photobleaching (FRAP) assay was used for the assessment of Atg9 mobility in *Drosophila* larvae as previously described (Seabrooke et al, 2010; Zhou et al, 2017). UAS-Atg9-GFP; D42-Gal4 flies were crossed with wild-type (w1118) or Rab39[KO] flies. At third-instar larval stage, male larvae were selected, dissected, and immediately imaged. Images were taken via Nikon A1R confocal microscope with a Nikon NIR Apo 60× W DIC N2 water-dipping objective (NA 1.0) in HL3 buffer. Images were acquired at 1 image/s, with a resolution 512 × 512 pixels using the 488 nm laser. To reduce variability, only the boutons in muscle 13 were used. In total, 6–10 boutons were used per animal. Only boutons that were bleached 95% or more were used for the analysis. For the FRAP experiment, a single image was taken before photobleaching. Afterward, a region with 0.7 μm diameter within the bouton was photobleached using 8 rounds of 100% laser power of a 488 nm laser. After the photobleaching step, images were taken at 1 s intervals, and fluorescence recovery was measured during the first 60 s after photobleaching. Images were aligned and quantified by Nikon Software 5.42. The data were processed and visualized using R Studio Version 2024.09.1 + 394 and GraphPad Prism.

## Axon cargo tracking

Fluorescence recovery after photobleaching (FRAP) assay combined with Single-Particle Tracking was used for the assessment of speed and number of mitochondria and synaptic vesicles along the *Drosophila* axon. UAS-mitoGFP and UAS-Synaptotagmin-eGFP

constructs were expressed under the control of D42-Gal4, respectively. At the third-instar larval stage, the larvae were dissected in HL3 buffer and imaged using Nikon A1R confocal microscope with a Nikon NIR Apo 60× W DIC N2 water-dipping objective (NA 1.0) in HL3 buffer. For the FRAP experiment, the optimal axon position was determined based on GFP signal. Before photobleaching, two images were taken as a reference. Afterward, most of the axon was bleached by using up to 20 rounds of 100% laser power of 488 nm laser. Afterwards, the area was imaged with 1 s intervals for 5 min. The images were analyzed using Fiji. The region of the axon was cropped, gaussian blur of 1 was added, and finally threshold was selected manually to visualize the particles. Trackmate was used to further analyze the speed, directionality and number of each particle during the fluorescent recovery step. For the selection of the parameters, the output of TrackMate was compared with the raw image. For the speed analysis, only the particles that were in the field of view for a minimum of three frames were selected. For the particle numbers, the spots on the tracks were manually checked and labeled with the multitask tool. The data results were further analyzed using R studio and GraphPad Prism.

## Electrophysiology

Electrophysiological recordings were performed at third-instar *Drosophila melanogaster* NMJs, specifically on muscle 6 of abdominal segments A2–A4. Dissections were carried out in HL3 with 1 mM $CaCl_2$; pH 7.2. Recordings were conducted at room temperature (21–23 °C) using sharp intracellular electrodes with resistances <20 MΩ when filled with 3 M KCl. Cells with input resistance ≥4 MΩ and resting membrane potential more negative than –60 mV were selected for analysis. The membrane potential was held at –70 mV throughout the experiment.

Synaptic currents and membrane potentials were recorded using an Axoclamp 900 A amplifier and digitized via a Digidata 1440 A interface (Molecular Devices). Data acquisition and analysis were performed using pCLAMP 10 (Clampfit, Molecular Devices). EJCs were evoked by stimulating the segmental motor nerve at two times threshold using a suction electrode, delivering 1 Hz pulses. EJCs were low-pass filtered at 1 kHz. For each cell, basal EJC amplitude was calculated by averaging 60 consecutive traces.

Spontaneous miniature EJCs (mEJCs) were recorded in 1 mM extracellular $Ca^{2+}$ for 5 min per cell, with signals filtered at 600 Hz. mEJC amplitudes were automatically detected and quantified using the event detection module in Clampfit.

## Correlative light electron microscopy (CLEM)

To characterize the Atg8-mCherry-positive structures with their ultrastructure, CLEM was performed (Bademosi et al, 2023; Vanhauwaert et al, 2017; Soukup et al, 2016; Decet et al, 2024). First, third-instar *rab39*[KO] mutant larvae endogenously expressing Atg8-mCherry were dissected in cold $Ca^{2+}$-free HL3 (100 mM NaCl, 5 mM KCl, 10 mM $NaHCO_3$, 5 mM Hepes, 30 mM sucrose, 5 mM trehalose, 10 mM $MgCl_2$, pH 7.2) and subsequently fixed for 2 h at 4 °C (0.5% glutaraldehyde, 2% paraformaldehyde in 0.1 M phosphate buffer (PB), pH 7.4). After washing in 0.1 M PB, fillets were incubated with DAPI (Sigma). Next, using a Zeiss LSM 780 equipped with a Mai Tai HP DeepSee laser (Spectra-Physics) at

880 nm with 40% maximal power output, near-infrared branding (NIRB) was used to apply visible branding marks. Before and after branding, Z-stacks of the region of interest (ROI) were acquired with a 25× water immersion lens (NA 0.8). Immediately after NIRB, fillets were post-fixed (4% paraformaldehyde, 2.5% glutaraldehyde in 0.1 PB) overnight at 4 °C and washed with 0.1 M PB and dd$H_2O$ until the dehydration steps. Subsequently, fillets were first osmicated for 1 h (1% $OsO_4$ and 1.5% potassium ferrocyanide) and then incubated in a 0.2% tannic acid for 30 min followed by a second osmication step (1% $OsO_4$ for 30 min) and then incubated for 20 min in 1% thiocarbohydrazide. Again, fillets were osmicated for a third time (1% $OsO_4$ for 30 min) and incubated overnight in 0.5% uranyl acetate. Thereafter, fillets were stained with lead aspartate (Walton's lead aspartate: 20 mM lead nitrate in 30 mM sodium aspartate, pH 5.5) for 30 min at 60 °C. After a final washing step, and a dehydration series (with solutions of increasing ethanol concentration (30%, 50%, 70%, 90% and twice with 100%)), fillets were incubated twice for 10 min with propylene oxide. Finally, fillets were infiltrated with resin agar 100 (Laborimpex), flat-embedded in resin agar 100 and placed at 60 °C for 48 h.

The flat resin-embedded samples were cropped into 1-mm$^2$ pieces with ROI in the middle and sectioned until the first branding marks were reached, and muscle morphology was recognized by correlating with the light microscopy data. Next, ultrathin sections (70 nm) were cut on an ultramicrotome (EM UC7, Leica), collected on 1 × 2 mm slot, copper grids (Ted Pella, Inc.) and imaged using a JEM-1400 transmission electron microscope (Jeol) at 80 keV. NIRB branding marks around the NMJ and DAPI signal were used to correlate the confocal images with the TEM micrographs of the NMJ boutons. Overlay images were generated using ImageJ and GIMP.

### Statistical analysis

All statistical analyses were performed using GraphPad Prism (version 10.0.2) and R (version 2023.09.1 + 494) with appropriate packages. Prior to statistical comparison, datasets were screened for outliers using the ROUT method (Q = 1%) in GraphPad Prism. Outliers were excluded from further analysis. Normality was assessed using the Shapiro–Wilk test or Kolmogorov–Smirnov test, depending on sample size. For normally distributed data, comparisons between two groups were performed using an unpaired two-tailed Student's *t* test, and comparisons among more than two groups were analyzed using a one-way ANOVA with post hoc Tukey or Dunnett test. For data not following a normal distribution, non-parametric tests were used: the Mann–Whitney *U* test for two groups, or the Kruskal–Wallis test for multiple groups, followed by Dunn's multiple comparisons test where applicable. For fractional data such as vesicle directionality (antero-grade, retrograde, stationary), statistical significance was assessed using Fisher's Exact Test with the Freeman–Halton extension where applicable. These tests were performed in R using the package *RVAideMemoire* for multi-category comparisons. Exact p-values are reported for all statistical tests. Data are presented as mean ± standard deviation (SD). Individual data points represent biological replicates.

### Experimental study design

No statistical methods were used to pre-determine sample size. Sample sizes were chosen based on previous experience with similar experiments and are consistent with standards in the field.

For all experiments, each individual *Drosophila* larva or adult was considered one biological replicate (n). In autophagy assays,

fluorescent timer assay, and dual-tag Atg8 assay, 2–4 NMJs per animal were analyzed, and the average value was used as a single data point ($n = 1$). For Rab39 and Atg9 NMJ stainings, the same quantification approach was applied. In localization studies of Rab39 with organelle markers, 2 neuronal somas per larval brain were analyzed and averaged per animal ($n = 1$). In axonal trafficking assays, one axon per animal was imaged and analyzed.

Unless stated otherwise, only male *Drosophila* were included in the analysis, as *rab39* is located on the X chromosome and hemizygous males allow clear interpretation of knockout genotypes. A small number of samples were excluded due to damaged tissues or poor imaging quality, based on criteria applied consistently across experiments.

No blinding was performed during data collection or analysis.

## Use of language assistance tools

ChatGPT (OpenAI) was used to assist with grammar checking, text editing, and formatting during the preparation of this manuscript. This tool was not used for data analysis, interpretation of results, or drawing scientific conclusions.

## Data availability

The whole-genome sequencing data from EMS-mutagenized adult *Drosophila*, generated to identify genetic modifiers of EndoA function, have been submitted to the NCBI Sequence Read Archive (SRA) under submission number SUB15273829 and assigned BioProject accession number PRJNA1257385. The data can be accessed via the following link: https://www.ncbi.nlm.nih.gov/bioproject/?term=PRJNA1257385. Further information and requests for resources and reagents should be directed to the lead contact, Patrik Verstreken (patrik.verstreken@kuleuven.be). Data, code, *Drosophila* models, and reagents are available upon request.

The source data of this paper are collected in the following database record: biostudies:S-SCDT-10_1038-S44318-025-00536-8.

## Peer review information

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

## Acknowledgements

We thank the Cell and Tissue Imaging Cluster at KU Leuven, supported by Hercules AKUL/11/37 and FWO grant G.0929.15 to Pieter Vanden Berghe, for technical support. We are also grateful to Dr. Gábor Juhász for generously providing fly stocks, and to all members of the Verstreken lab for insightful discussions throughout this work. AK (11E2223N), GO (1165925N), DV (151095), and EMP (11O3225N) were supported by PhD fellowships from FWO Vlaanderen, and EN (1282123N) by a postdoctoral fellowship from FWO Vlaanderen. This research was further supported by the European Research Council (ERC Advanced Grant ERC-2022-AdG 101054310, Synaptic resilience in Tau-induced neurodegeneration), the Chan Zuckerberg Initiative, a Methusalem grant from the Flemish Government, and FWO Vlaanderen project grants (The Role of Parkinsonism Genes in Synaptic Autophagy, G031324N; Synaptic Autophagy: A Common Culprit in Parkinson's Disease, G0B8119N). Additional support was provided by Hibernating Synapses—Opening the Future, and the KU Leuven Parkinson Fonds (Disentangling Complex Genetic Interactions in Idiopathic Parkinson's Disease). PV is an alumnus of the FENS-Kavli Network of Excellence. All scientific illustrations were created using BioRender.com under an academic license.

## Author contributions

**Ayse Kilic**: Conceptualization; Resources; Data curation; Software; Formal analysis; Funding acquisition; Investigation; Visualization; Methodology; Writing—original draft; Project administration; Writing—review and editing. **Gokhan Ozturan**: Data curation; Formal analysis; Funding acquisition; Investigation; Visualization; Methodology; Writing—review and editing. **Dirk Vandekerkhove**: Data curation; Formal analysis; Funding acquisition; Investigation; Methodology. **Sabine Kuenen**: Resources; Data curation; Formal analysis; Visualization; Writing—original draft. **Jef Swerts**: Data curation. **Esther Muñoz Pedrazo**: Funding acquisition; Validation. **Carles Calatayud Aristoy**: Validation. **Abril Escamilla Ayala**: Data curation; Formal analysis; Investigation; Visualization; Methodology. **Nikky Corthout**: Formal analysis. **Pablo Hernandez Varas**: Data curation; Formal analysis; Methodology. **Stephane Plaisance**: Data curation; Software; Formal analysis. **Valerie Uytterhoeven**: Conceptualization; Data curation; Formal analysis; Supervision; Investigation; Visualization; Methodology; Writing—original draft; Project administration; Writing—review and editing. **Eliana Nachman**: Conceptualization; Data curation; Supervision; Funding acquisition; Validation; Investigation; Methodology; Writing—original draft; Project administration; Writing—review and editing. **Patrik Verstreken**: Conceptualization; Resources; Supervision; Funding acquisition; Methodology; Writing—original draft; Project administration; Writing—review and editing.

Source data underlying figure panels in this paper may have individual authorship assigned. Where available, figure panel/source data authorship is listed in the following database record: biostudies:S-SCDT-10_1038-S44318-025-00536-8.

## Disclosure and competing interests statement

PV is the scientific founder of Jay Therapeutics. The remaining authors declare no competing interests.

# Expanded View Figures

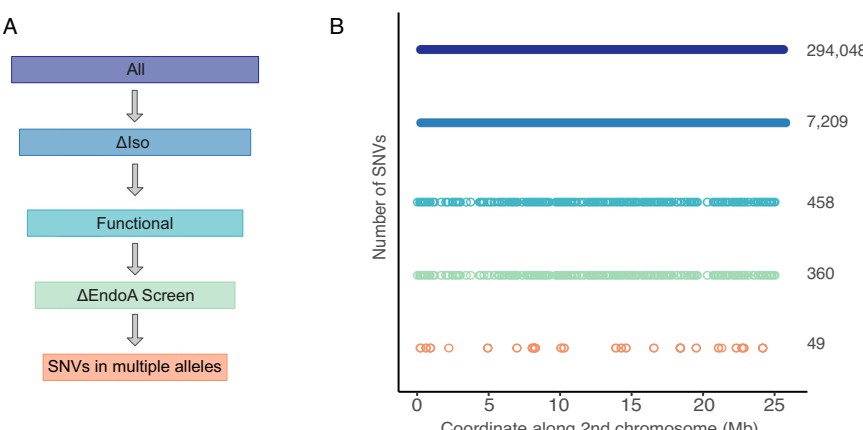

**Figure EV1.   Filtering process to identify candidate mutations in heterozygous EMS lines.**

(A) Flowchart summarizing the filtering pipeline used to identify candidate mutations in heterozygous EMS-induced mutants (*cn bw*). All single nucleotide variants (SNVs) identified (ALL) were first filtered against variants present in the isogenized second chromosome (ΔIso). Next, only SNVs affecting coding regions or splice sites were retained (functional). Finally, recurrent background variants detected in multiple sequenced genomes were excluded (ΔEndoA Screen). (B) Overview of the number of SNVs on the second chromosome (25 Mb interval) at each filtering step described in (A).

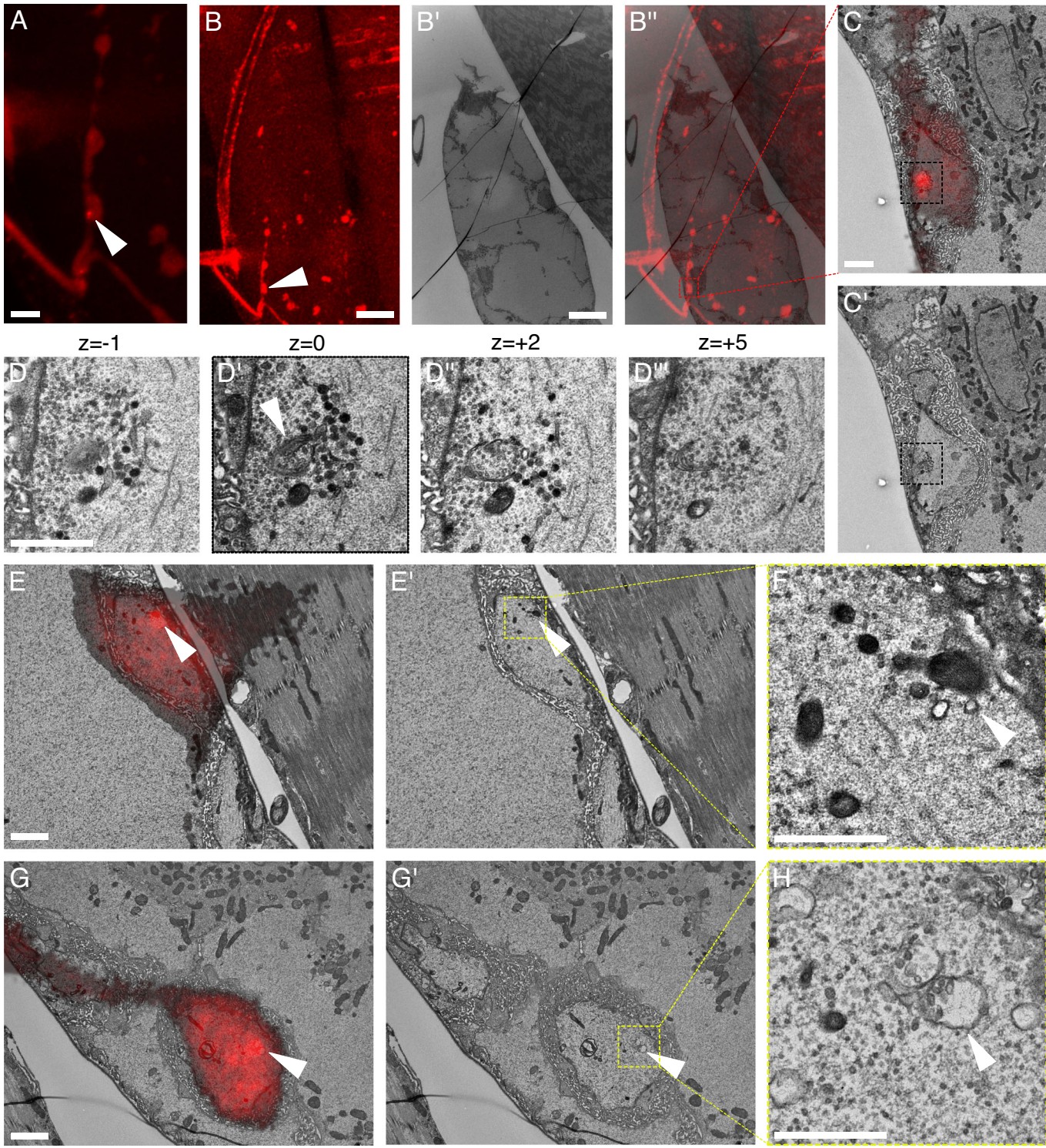

◀   **Figure EV2.   CLEM of boutons of *rab39*^KO animals expressing Atg8-mCherry.**

(A) Maximum intensity projection of confocal slices of an example NMJ1 displaying an Atg8-mCherry-positive structure (arrowhead). Scale bar: 5 µm. (B) Zoomed-out maximum projection of the same NMJ1 after branding shown in (A). Arrowhead indicates the bouton corresponds to the bouton in (A). Scale bar: 20 µm. (B′) Electron micrograph of the same region as in (B). Scale bar: 20 µm. (B″) Overlay of confocal image in (B) with the electron micrograph in (B′). (C, C′) Zoomed views of the bouton containing the Atg8-mCherry structure shown as overlay in the red square (B″) and EM alone (C′). The black square highlights the structure corresponding to the mCherry signal. Scale bar: 2 µm. (D–D‴) Single TEM slices showing the putative autophagosomal structure (arrowhead) shown in black square in (C-C′), visible in multiple consecutive sections. Scale bar: 1 µm. (E, E′) Overlay of a fluorescence image section from NMJ2 with the corresponding EM image. Arrowheads indicate the structure corresponding to the mCherry signal. Scale bar: 2 µm. (F) Zoom of the structures highlighted in yellow square in (E′). Scale bar: 1 µm. Arrowheads indicate the putative autophagosomal structure. (G, G′) Overlay of a fluorescence image section from another NMJ3 with its corresponding EM image. Arrowheads indicate the structure corresponding to the mCherry signal. Scale bar: 2 µm. (H) Zoom of the structures highlighted in yellow square in (G′). Scale bar: 1 µm. Arrowheads indicate the putative autophagosomal structure.

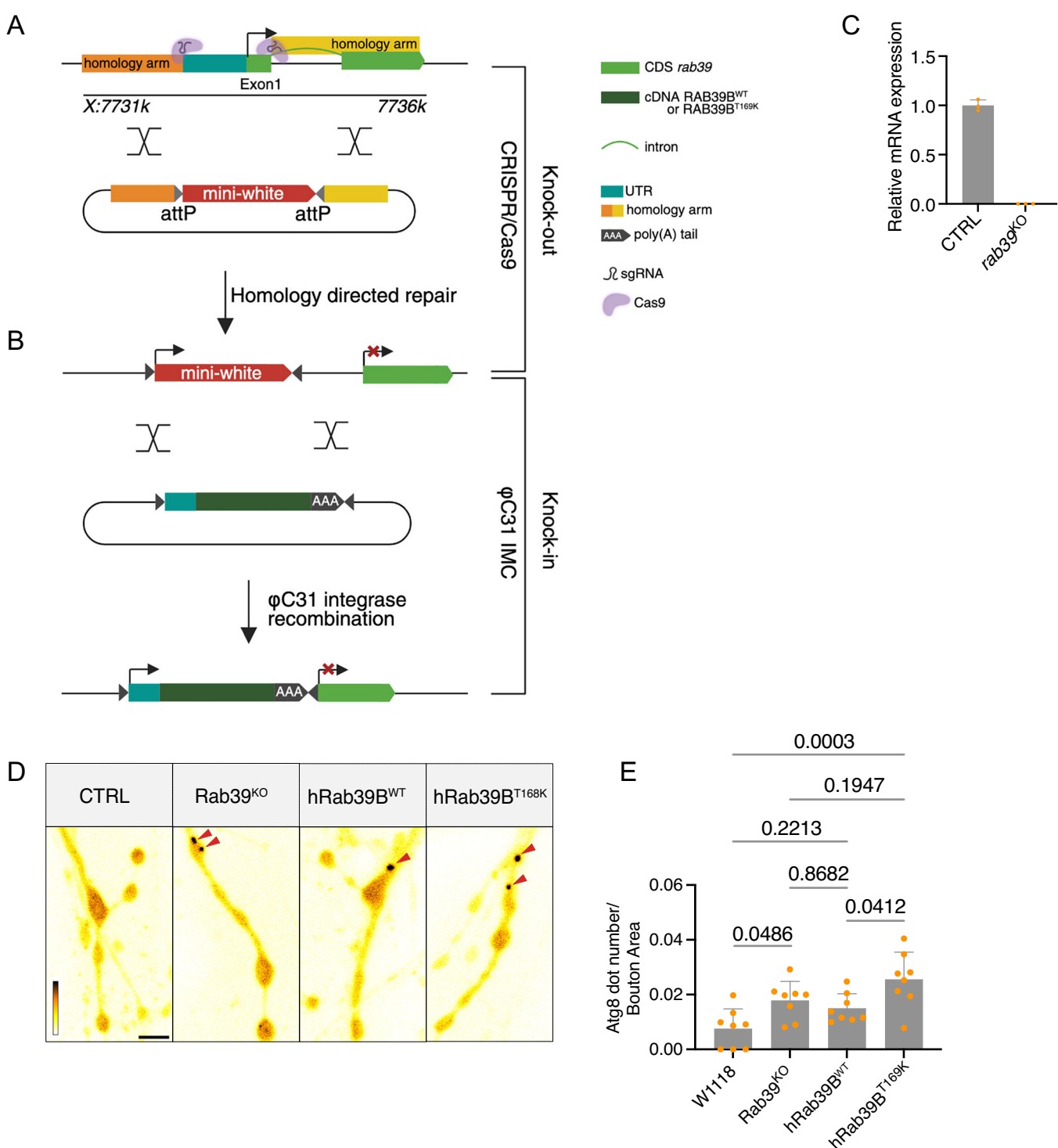

**Figure EV3. Pathogenic *RAB39B^{T168K}* mutant mimics *rab39*^{KO} and increases synaptic autophagy.**

(A, B) Schematic of the *rab39* knockout (A) and knock-in (B) strategy. The first exon of *Drosophila rab39* was replaced with an attP-flanked mini-white cassette via CRISPR/Cas9-mediated homologous recombination, generating a null allele (*rab39*^{KO}). The chromosomal insertion site is indicated. This mini-white cassette was then replaced by the human wild-type or pathogenic mutant cDNA (including a stop codon) to create "human knock in alleles". (C) *rab39* mRNA expression levels measured by RT-qPCR. Expression is shown relative to endogenous *Drosophila rab39* transcript levels (*w^{1118}w^+*, CTRL). (D) Live imaging of genomically expressed Atg8-mCherry in NMJ boutons of control (*w^{1118}w^+*, CTRL), *rab39*^{KO}, *RAB39B^{WT}*, and *RAB39B^{T168K}* animals. Fluorescence intensities are shown using the gray value range indicated in ((D), CTRL) (323–3045). Red arrowheads mark Atg8-mCherry-positive puncta. Scale bar: 5 μm. (F) Quantification of Atg8-mCherry puncta per bouton area from the experiment shown in (E). Statistical test: two-way ANOVA with Tukey's multiple comparison test; *n* = 8; error bars: mean ± SD.

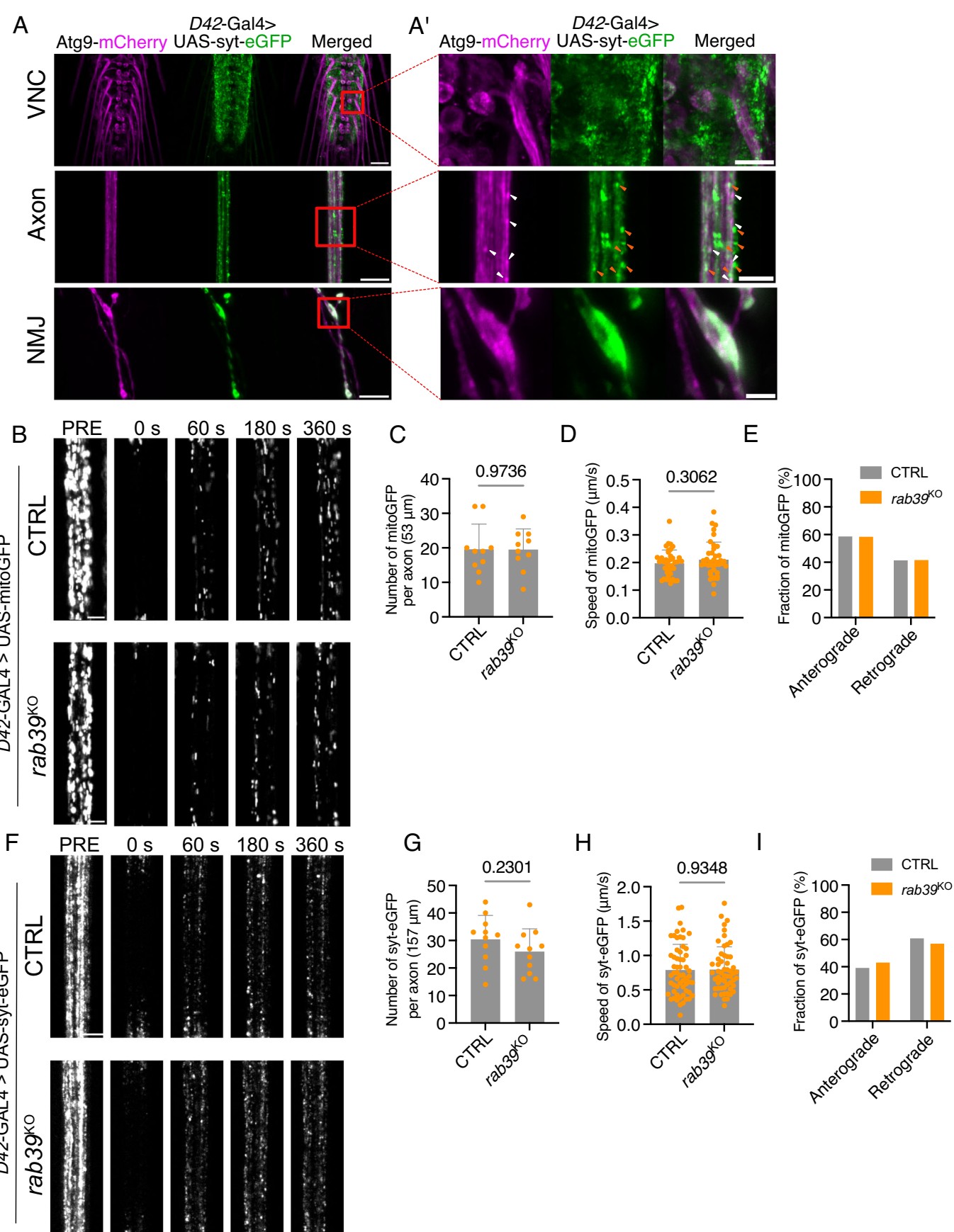

◀ **Figure EV4. Rab39 does not affect synaptic vesicle or mitochondrial transport.**

(A–A′) Representative live confocal images showing Atg9-mCherry and Syt-eGFP in the ventral nerve cord (VNC), axons, and neuromuscular junctions (NMJs) of *Drosophila* third-instar $w^{1118}$ larvae. Red squares in (A) indicate regions enlarged in (A′). White arrowheads mark Atg9-mCherry vesicles; orange arrowheads indicate Syt-eGFP vesicles. Scale bars: 100 μm (A) 40 μm (A′) (VNC); 40 μm (A) 20 μm (A′) (axon); 40 μm (A), 10 μm (A′) (NMJ). (B, F) Representative pre- and post-bleach images (t = 0 and t = 360 s) of axons expressing *D42*-Gal4 > UAS-mitoGFP (B) or UAS-Syt-eGFP (F) in control and $rab39^{KO}$ larvae. Scale bars: 5 μm (B), 20 μm (F). (C, G) Quantification of mitoGFP- and Syt-eGFP-positive particles per defined axon length (53 μm and 157 μm, respectively). Statistical test: unpaired *t* test; $n = 10$ (C), $n = 11$ (G); error bars: mean ± SD. (D, H) Quantification of vesicle speed for mitoGFP ($n = 39$ traces/group from 10 animals) and Syt-eGFP ($n = 59$ [CTRL] and 55 [$rab39^{KO}$] traces from 11 animals). Statistical test: unpaired *t* test; error bars: mean ± SD. (E, I) Direction of vesicle movement (anterograde vs. retrograde) for mitoGFP (E) and Syt-eGFP (I) in control and $rab39^{KO}$ axons. Statistical test: Fisher's Exact Test; (E) *P* value = 1; (I) *P* value = 0.3273 exact counts in Appendix Table S1.

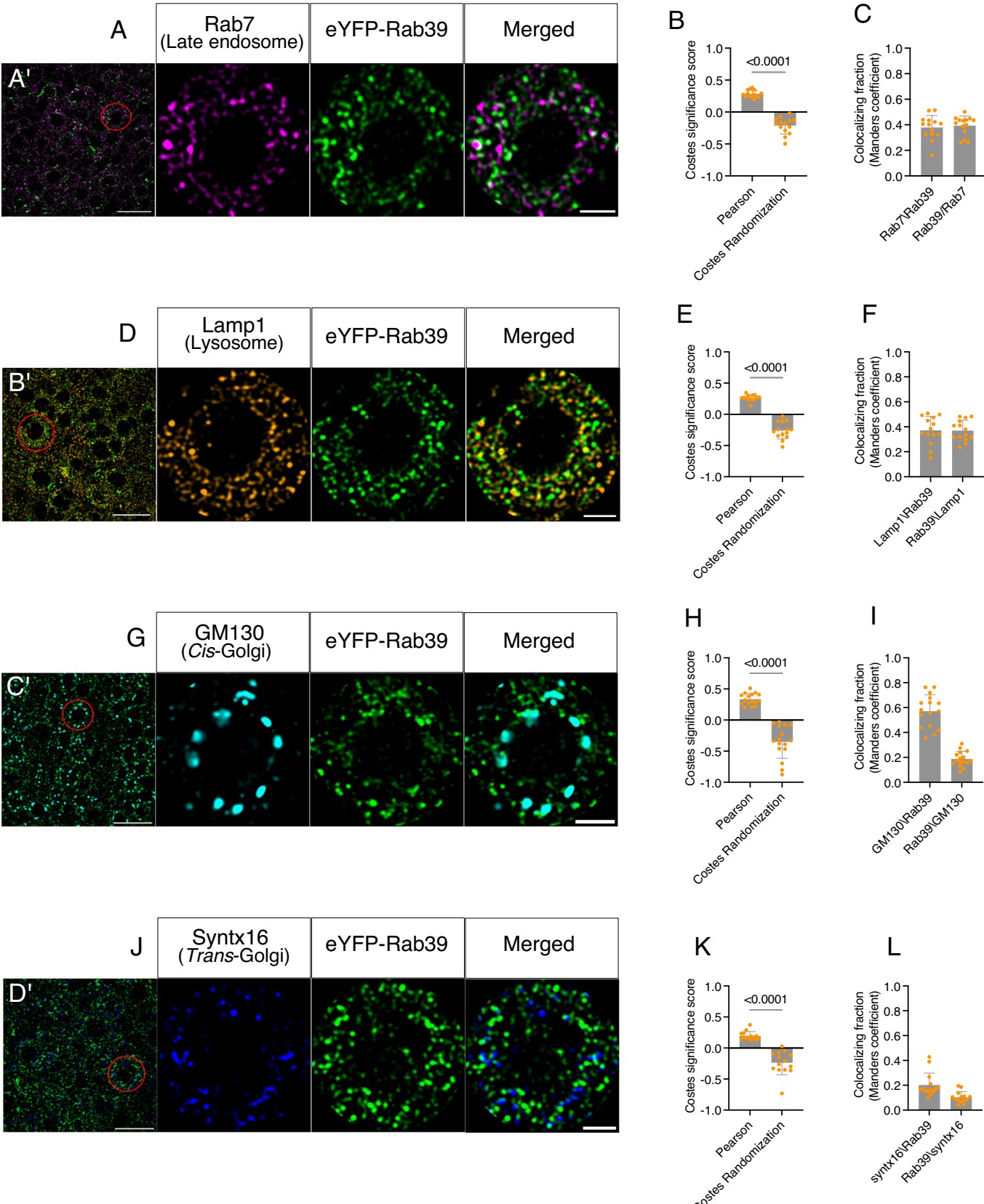

◀ **Figure EV5. Rab39 localizes to organelles involved in Golgi–endolysosomal trafficking.**

(**A–A′, D–D′, G–G′, J–J′**) Representative Elyra super-resolution images of third-instar larval VNCs showing neuronal cell bodies stained with anti-GFP (eYFP-Rab39, green) and markers for various organelles. Overviews of the VNC are shown in (**A′, D′, G′, J′**). Scale bar: 20 μm; zoomed-in cell bodies in (**A, D, G, J**) are in red circular ROIs in ((**A′**), **D′, G′, J′**) respectively, Scale bar: 3 μm. Co-stains include anti-Rab7 (late endosome/lysosomes, magenta; (**A**)), anti-Lamp1 (late endosomes/lysosomes, orange; (**D**)), anti-GM130 (cis-Golgi, cyan; (**G**)), and anti-Syntaxin-16 (trans-Golgi, blue; (**J**)). (**B, E, H, K**) Pearson's correlation coefficients for Rab39 and each marker, as well as with Costes' randomization threshold (≥100 iterations). Exact $P$ (**B**) $= 2.913 \times 10^{-8}$; exact $P$ (**E**) $= 3.618 \times 10^{-9}$; exact $P$ (**H**) $= 2.419 \times 10^{-7}$; exact $P$ (**K**) $= 8.892 \times 10^{-6}$. Statistical test: unpaired $t$ test; $n = 14$ (Rab7, Lamp1, Syntaxin-16), $n = 16$ (GM130); error bars: mean ± SD. (**C, F, I, L**) Manders' overlap coefficients indicating the fraction of Rab39 overlapping with each marker and vice versa.

