## [Peer Review File · The EMBO Journal]

Soma-localized Rab39 inhibits synaptic autophagy by controlling trafficking of Atg9 vesicles

Ayse Kilic, Gokhan Ozturan, Dirk Vandekerckhove, Sabine Kuenen, Jef Swerts, Esther Munoz Pedraza, Carles Calatayud, Abril Escamilla Ayala, Nikky Corthout, Pablo Hernandez Varas, Stephane Plaisance, Valerie Uytterhoeven, Eliana Nachman, and Patrik Verstreken

Corresponding author(s): Patrik Verstreken (patrik.verstreken@kuleuven.be) , Eliana Nachman (eliana.nachman@kuleuven.be), Valerie Uytterhoeven (valerie.uytterhoeven@kuleuven.be)

Review Timeline:

Submission Date:	10th Dec 24
Editorial Decision:	21st Jan 25
Revision Received:	30th May 25
Editorial Decision:	30th Jun 25
Revision Received:	6th Jul 25
Accepted:	25th Jul 25

Editor: Ioannis Papaioannou

Transaction Report:

Dear Patrik,

Thank you again for submitting your manuscript EMBOJ-2024-119885 for consideration by The EMBO Journal, and for your patience during peer review. Your manuscript has now been seen by three experts in the field, and we have received the full set of their comments, which are included below.

As you will see, all three referees indicate interest in the topic and your findings, and their assessment is overall in agreement with our editorial decision to consider this manuscript for publication in The EMBO Journal. However, they also identify limitations in the study and the manuscript, and they raise a number of concerns, which together indicate that the conclusions of the study are not fully supported by the available data, and certain aspects should be further clarified and strengthened by performing additional experimental work. Furthermore, one of the raised concerns that must be addressed refers to the need for better contextualization of the study and its conclusions by incorporating relevant progress from other labs and model systems.

Given the referees' comments and recommendations, and our understanding that the specific points made by the referees are largely addressable, I would like to invite you to submit a thoroughly revised version of the manuscript along with a detailed point-by-point response addressing all referees' comments. I should add that it is The EMBO Journal policy to allow only a single round of major revision, and acceptance of your manuscript will therefore depend on the completeness of your responses in this revised version. Please let me know if you would like to discuss with me further any particular points from the referees' reports, or if you have any questions or other comments.

We generally allow three months as standard revision time (April 20, 2025). As a matter of policy, competing manuscripts published during this period will not negatively impact our assessment of the conceptual advance presented by your study. However, we request that you contact us as soon as possible upon publication of any related work, to discuss how to proceed. Should you foresee a problem in meeting this three-month deadline, please let us know in advance and we may be able to grant an extension.

Thank you for the opportunity to consider your work for publication in The EMBO Journal. I look forward to your revision.

Best regards,

Ioannis

Instructions for preparing your revised manuscript

1. When you are ready to submit the revision, please upload:

- A Word file of the manuscript text (including legends of main Figures, EV Figures and Tables). Please make sure that changes are highlighted (or "tracked") to be clearly visible.

- Individual production-quality figure files (one file per figure). When assembling your figures, please refer to our figure preparation guidelines in order to ensure proper formatting and readability in print as well as on screen:

If the data shown in a figure are obtained from n {less than or equal to} 2, please use scatter plots showing the individual data points.

- i. the name of the statistical test used to generate error bars and P values
- ii. the number (n) of independent experiments (please specify technical or biological replicates) underlying each data point (discussion of statistical methodology can be reported in the Materials and Methods section, but figure legends should contain a basic description of n , P , and the test applied)
- iii. the nature of the bars and error bars (s.d., s.e.m.).

- A point-by-point response to the referees' comments, with a detailed description of the changes made (as a word file). All referees' concerns must be fully addressed and their suggestions taken on board. When preparing your letter of response to the referees' comments, please bear in mind that this will form part of the Review Process File and will therefore be available online to the community. Please note that you have the possibility to opt out of the transparent process at any stage prior to publication by letting the editorial office know (contact@embojournal.org); if you do opt out, the Review Process File link will point to the following statement: "No Review Process File is available with this article, as the authors have chosen not to make the review process public in this case.". For more details on our Transparent Editorial Process, please visit our website: <https://www.embopress.org/page/journal/14602075/authorguide#transparentprocess>

- Expanded View (EV) files (replacing Supplementary Information) that are collapsible/expandable online. A maximum of 5 EV Figures can be typeset. EV Figures should be cited as "Figure EV1, Figure EV2" etc. in the text, and their respective legends should be included in the manuscript file after the legends of regular figures. See detailed instructions regarding Expanded View files here: <https://www.embopress.org/page/journal/14602075/authorguide#expandedview>

- For the figures that you do NOT wish to display as Expanded View figures, they should be bundled together with their legends in a single PDF file called "Appendix", which should start with a short Table of Contents (including page numbers). Appendix figures should be referred to in the main text as: "Appendix Figure S1, Appendix Figure S2" etc. Please see detailed instructions here: <https://www.embopress.org/page/journal/14602075/authorguide#expandedview>

- A complete author checklist, which you can download from our author guidelines (<https://www.embopress.org/page/journal/14602075/authorguide>). Please note that the checklist will also be part of the Review Process File.

2. Please note that no statistics should be calculated and shown in Figures if $n=2$. Please also note that each p value should be reported as an exact value.

3. Before submitting your revision, primary datasets (and computer code, where appropriate) produced in this study need to be deposited in appropriate public databases (see <https://www.embopress.org/page/journal/14602075/authorguide#dataavailability>).

In particular, we kindly request you to deposit all next-generation sequencing data produced in this study in appropriate databases. If new code was generated for data analysis, it should also be made publicly available. The accession numbers, database, and the specific URLs (links) should be listed in a formal "Data availability" section (placed after Methods), following the example below:

"The RNA-seq datasets produced in this study are available in the following database:
Gene Expression Omnibus GSE46843 (<https://www.ncbi.nlm.nih.gov/geo/query/acc.cgi?acc=GSE46843>)"

*** All links should resolve to a page where the data can be accessed. ***

*** Please remember to provide in the Data availability section of your revised manuscript reviewer passwords if the datasets are not yet public. ***

*** The Data Availability Section is restricted to new primary data that are part of this study. In case you have no data that require deposition in a public database, please state so instead of referring to the database: "Our study includes no data deposited in public repositories." under the heading "Data availability". ***

4. Please check that the title and the abstract of the manuscript are brief, yet explicit, even to non-specialists. The length of the title should not exceed 100 characters, and the abstract should be a single paragraph not exceeding 175 words.

5. Please also note our reference format: <https://www.embopress.org/page/journal/14602075/authorguide#referencesformat>.

7. Please remember: digital image enhancement is acceptable practice, as long as it accurately represents the original data and conforms to community standards. If a figure has been subjected to significant electronic manipulation, this must be noted in the figure legend or in the "Materials and Methods" section. The editors reserve the right to request original versions of figures and the original images that were used to assemble the figure.

8. Our journal encourages inclusion of data citations in the reference list to directly cite datasets that were obtained from public databases. Data citations in the article text are distinct from normal bibliographical citations and should directly link to the database records from which the data can be accessed. In the main text, data citations are formatted as follows: "Data ref:

Smith et al, 2001" or "Data ref: NCBI Sequence Read Archive PRJNA342805, 2017". In the Reference list, data citations must be labeled with "[DATASET]". A data reference must provide the database name, accession number/identifiers, and a resolvable link to the landing page from which the data can be accessed at the end of the reference. Further instructions are available at: <https://www.embopress.org/page/journal/14602075/authorguide#referencesformat>.

9. We request authors to consider both actual and perceived competing interests. Please review our policy (<https://www.embopress.org/page/journal/14602075/authorguide#conflictsofinterest>) and update your competing interests statement if necessary. Please name this section 'Disclosure and competing interests statement' and place it after the Acknowledgements section.

10. Please note that all corresponding authors are required to provide an ORCID ID upon submission of a revised manuscript (<https://orcid.org/>). Please find instructions on how to link your ORCID ID to your account in our manuscript tracking system in our Author guidelines (<https://www.embopress.org/page/journal/14602075/authorguide#authorshipguidelines>).

11. We use CRediT to specify the contributions of each author in the journal submission system. CRediT replaces the author contribution section, which should be removed from the manuscript. Please use the free text box to provide more detailed descriptions. See also guide to authors: <https://www.embopress.org/page/journal/14602075/authorguide#authorshipguidelines>.

13. We would also welcome the submission of cover suggestions or motifs to be used by our Graphics Illustrator in designing a cover.

14. Please use the link below to submit your revision:
<https://emboj.msubmit.net/cgi-bin/main.plex>

Referee #1:

Autophagy is crucial for maintaining neuronal health by clearing and recycling dysfunctional components, processes that are essential for synaptic integrity and neuronal communication. While synapses are often located far from the neuronal cell body and are typically thought to regulate autophagy independently, the authors present data suggesting a soma-centered mechanism involving Rab39. The loss of Rab39-a gene whose human homolog is linked to Parkinson's disease (PD)-appears to increase synaptic autophagy, leading to faster protein turnover and neurodegeneration.

Through a genetic modifier screen, the authors identify cytoskeletal and axonal organizing proteins as suppressors of excessive synaptic autophagy. Their findings indicate that Rab39 regulates the trafficking of Atg9-positive vesicles from the soma to synapses via cytoskeletal and axonal transport proteins, including Shortstop and Unc104/KIF1a. Under starvation conditions, Rab39 partially shifts its localization from endosomes to lysosomes in the soma, potentially controlling the availability of Atg9 vesicles for synaptic trafficking. Additionally, PD-associated mutations in Rab39B are shown to result in increased synaptic autophagy, similar to effects observed with LRRK2 mutations or phosphomimetic EndoA mutants.

These results support the notion that Rab39-mediated soma-centric regulation plays a role in balancing synaptic autophagy, with significant implications for PD pathophysiology and neuronal health.

This is per se an intriguing manuscript that should be considered for publication in the EMBO Journal, particularly due to the strength of the screening and genetic interaction studies. However, I have several criticisms that I would like the authors to address, at least in parts experimentally:

1. Claim of an exclusively somatic mechanism: Establishing negative evidence-such as the absence of Rab39 function in the presynaptic terminal-is inherently challenging. The authors mention that they used a line expressing YFP-tagged Rab39 at endogenous levels and found Rab39 exclusively in the neuronal cell body, without significant labeling at synaptic boutons of neuromuscular junctions (NMJs). However, the provided images are not entirely convincing. I recommend quantifying the Rab39 localization, possibly using individual confocal sections, and offering more detailed visuals of the Rab39-positive structures in the cell bodies. By the way: is the endogenously EYFP-Rab39 functional? What do they know about the Rab39 positive spots in the motoneuron somata? Anything co-localizing?
2. Interpretation of Atg9-positive vesicle transport: The authors observe a higher number of Atg9-positive vesicles in the axons of rab39 mutants, but the transport speed of these vesicles decreases. What is the overall effect of this observation? They should measure flux rates-specifically, the number of Atg9-positive vesicles crossing a certain axonal point normalized over time. In my opinion, they have not conclusively demonstrated an increased flux of Atg9-positive vesicles down the axons in rab39 mutants.

There also appears to be a trend of more Atg9-positive vesicles moving retrogradely in the rab39 mutant. Providing imaging trajectories of the Atg9-positive vesicles would strengthen their findings.

3. Specificity of the effect on Atg9-positive vesicles: How specific is the observed effect on Atg9-positive vesicles in rab39 mutants? Do other vesicular species to be transported in axons exhibit similar changes?

4. Their readout of autophagy (function) nearly exclusively relies on counting ATG8 spots at the NMJ terminals (apart from the ERGs). For example: are there any physiological changes to be expected as a result of ATG9/Rab39 manipulation at the NMJ terminals?

Referee #2:

Summary

In this study, Kilic et al. proposed a possible role for Rab39 in the neuronal autophagic process using a *Drosophila* model system. Based on findings from mutant fly phenotypes, the authors claimed that Rab39, whose human homolog is mutated in Parkinson's disease (PD) and is exclusively expressed in the somatic region, regulates synaptic autophagy by controlling the anterograde transport of Atg9 vesicles. As the authors noted, presynaptic autophagy and the functional roles of Atg9 vesicles are very interesting topics in neuroscience. However, the evidence presented by the authors is somewhat fragmented and not well connected. To solidify this concept, the manuscript needs to address key questions that remain unresolved.

Major concerns

1. The authors claimed that Rab39, which localizes to the somatic region, regulates the autophagic process at synapses by inhibiting the anterograde transport of Atg9 vesicles from the soma to nerve terminals. This concept is quite interesting. However, given the significant distance between the neuronal soma and synapses, it remains unclear how this transport process can effectively support presynaptic autophagy in a timely manner. Additionally, the findings presented in the manuscript are not well connected. For instance, there is no direct evidence demonstrating how the Rab39 knockout (KO)-mediated increase in Atg9 vesicle transport disrupts synaptic autophagy. What is the physiological significance of the increased Atg9 vesicle transport to nerve terminals? Without data directly linking this process to synaptic autophagy, it is difficult to conclude that Rab39-regulated anterograde trafficking of Atg9 vesicles is crucial for presynaptic autophagy and neuronal health.

2. Although the authors showed that synaptic protein levels remain unchanged in both control and Rab39-KO conditions, it is important to demonstrate that the anterograde transport of synaptic vesicle proteins is also unaffected in Rab39-KO axons. Ideally, measuring both Atg9 and synaptic vesicle proteins simultaneously within the same axon would provide more comprehensive and convincing evidence.

3. The data (Figure 5) show that the anterograde transport of Atg9 vesicles is significantly increased in Rab39-KO neurons. However, what about the total levels of Atg9 at synapses? Are they also increased?

Minor concerns

1. For readers unfamiliar with the LIND technique, it would be helpful to provide more details in the text or include a schematic representation in the figure.

2. The high-magnification EM images in Figure 3 should be displayed at a larger size.

3. The changes shown in Figures 3J and 3K are very subtle. At the very least, the authors should acknowledge this in the main text.

4. Please label the fluorescent molecule (Atg9-mCherry) in Figures 5J-L to help readers understand the figure without having to refer to the figure legend.

5. In Figure 5O, the graph should display the raw data, as is done in other figures.

6. Is the sentence in lines 285-286 accurately conveying what the authors intend to say?

7. There are a few typos, such as the period in line 71 and the missing parenthesis in line 290.

Referee #3:

The manuscript by Kilic and colleagues describes a large-scale genetic modifier screen for suppressors of synaptic autophagy. The authors identify Rab39 and shot as genes of interest, and then examine the effects of Rab39 and shot mutants on autophagy. Trafficking data suggest that both Rab39 and shot participate in the trafficking of Atg9 vesicles to synapses. Based on localization data, the authors propose that Rab39 functions to regulate this pathway in the soma, from either endosomes or lysosomes, while Shot is more likely to function either at the AIS or along the axon, although a function in the soma is not ruled out. The work concludes with a model in which Rab39-mediated trafficking in the soma orchestrates a cross-compartmental mechanism to regulate autophagy abundance or flux at synapses.

This work addresses a pathway implicated in the regulation of neuronal homeostasis, and reports some new observations that are likely to be of interest to the field. However, significant revisions are required prior to publication. The logic flow is unclear in

many sections, some of the data shown are less than convincing, many conclusions are not strongly supported by the data provided, and key aspects of the authors' model remain unproven. The work is also a bit myopic, building on too narrow a base by not fully incorporating progress from work from other labs on autophagy in *Drosophila*, *C. elegans*, and other model systems. These points are all addressable, and substantial revisions would make this work a much stronger contribution to the literature.

Specific points to address:

1. The title states that there is Rab39-regulated anterograde trafficking of Atg9, but this is not directly assessed in the work; rather, it is a conclusion by inference. The title should more accurately reflect what is actually being measured.
2. Autophagy is a highly conserved pathway in general, and neuronal autophagy also appears to be conserved across fly, worm, and mouse models. However, both the introduction and the analysis presented in this work fail to take advantage of the extensive literature documenting mechanisms of autophagosome formation and maturation, previous work on the synaptic proteins that are degraded by synaptic autophagy, and previous work on the role of Kif1A in Atg9 transport. For example, synaptic autophagosomes have been shown to mature during trafficking in fly models (Neisch et al., 2017), but this possibility is not considered here although it likely affects some of their conclusions. While some aspects of synaptic autophagy may be distinct between fly and vertebrate models, such as the effects of starvation on neuronal autophagy or the effects of mutant LRRK2 on autophagosome formation, it would be helpful to describe these differences more clearly rather than omitting them from the discussion of the results in the current study, especially given their interest in using *Drosophila* as a model for human disease mechanisms.
3. The authors report an interesting genetic screen to identify components regulating synaptic autophagy, which relies on exposing flies to constant light. I'm curious - what are the effects of this assay on circadian rhythms, as there has been some thought that circadian rhythms regulate autophagy?
4. Line 153 states that Atg8 marks the formation and maturation of autophagosomes - this is not exactly true. More importantly, the signals that the authors are quantitating are very difficult to see in the images provided here. Given this issue, and the large variability in signal (see Figure 5G), it would be good to include larger galleries of the primary images in the supplement, with the points being quantitated clearly labeled.
5. The authors report more puncta in Rab39KO neurons, although this is somewhat difficult to appreciate in the images shown in Figure 3. This could be due to faster or more frequent formation, or slower trafficking away from the synapse, or slower turnover. The authors report a small increase in the ratio of magenta over green puncta, which is consistent with a small change in either decreased trafficking or maturation or both.
6. In Figure 3H, the authors use CLEM to verify that the puncta they are tracking are indeed autophagosomes. These images are not particularly convincing. The first panel appears to be unaligned with the subsequent panels, and the zoom-in images are not strikingly clear. Cleaner and/or more examples would strengthen this point.
7. Similarly, it is really hard to appreciate the indicated changes in the primary data shown in Figure 3L. Again, more and better examples would strengthen this point.
8. The authors report seeing no changes in the absolute levels of BRP, CSP, and synaptotagmin1A, but have these proteins been reported to be turned over by synaptic autophagy? Querying proteins identified in proteomic screens might be helpful here.
9. The conclusion that human Rab39B can compensate for the loss of Rab39 would not be strictly correct unless the first exon encodes the entire protein. Can the authors verify this, or moderate their conclusion?
10. The conclusion that EndoA acts downstream of Rab39 seems weak based on the experiment reported in Figure 3N-P. What if EndoA initiates autophagosome formation and Rab39 regulates subsequent steps in assembly or maturation? Given the multiple possibilities, this seems like too strong a conclusion.
11. The title of Figure 5 is that Rab39 regulates presynaptic autophagy by the transport of Atg9 vesicles. But the authors don't actually measure the trafficking of Atg9 vesicles, so it is unclear how they can make this strong conclusion. I agree that it is a likely possibility, but one that requires more direct proof.
12. The experiments in Figure 6 are confusing to interpret, and lack necessary controls. Is Rab7 a specific marker for endosomes and Lamp1 a specific marker for lysosomes in the fly, unlike in other organisms? If so, it should be straightforward to show co-staining with distinct and nonoverlapping distributions. This would better support the interpretation that there is a change in Rab39 localization under the two different conditions, but additional data from an orthogonal assay (biochemistry or something similar) would be required to solidify this conclusion.
13. Several of the conclusions drawn in the Discussion section seem too definitive, given the limited data provided in support. For example, have the authors really demonstrated a pivotal role for Rab39 in organizing vesicular trafficking within the neuronal soma towards lysosomes? Their model that Rab39 regulates the loading of Atg9-positive vesicles is plausible, but not demonstrated. And as they note in their further discussion of this model, many elements remain speculative. For example, they cannot explain why Rab39 mutations do not affect all Kif1a trafficking, and they cannot clarify whether Rab39 and shot are working in the soma or the AIS.

In sum, a more conservatively written paper that includes only rigorously supported conclusions would be a stronger contribution to the literature than the manuscript in its current form.

Referee #1:

Autophagy is crucial for maintaining neuronal health by clearing and recycling dysfunctional components, processes that are essential for synaptic integrity and neuronal communication. While synapses are often located far from the neuronal cell body and are typically thought to regulate autophagy independently, the authors present data suggesting a soma-centered mechanism involving Rab39. The loss of Rab39 -a gene whose human homolog is linked to Parkinson's disease (PD)-appears to increase synaptic autophagy, leading to faster protein turnover and neurodegeneration.

Through a genetic modifier screen, the authors identify cytoskeletal and axonal organizing proteins as suppressors of excessive synaptic autophagy. Their findings indicate that Rab39 regulates the trafficking of Atg9-positive vesicles from the soma to synapses via cytoskeletal and axonal transport proteins, including Shortstop and Unc104/KIF1a. Under starvation conditions, Rab39 partially shifts its localization from endosomes to lysosomes in the soma, potentially controlling the availability of Atg9 vesicles for synaptic trafficking. Additionally, PD-associated mutations in Rab39B are shown to result in increased synaptic autophagy, similar to effects observed with LRRK2 mutations or phosphomimetic EndoA mutants.

These results support the notion that Rab39-mediated soma-centric regulation plays a role in balancing synaptic autophagy, with significant implications for PD pathophysiology and neuronal health.

This is per se an intriguing manuscript that should be considered for publication in the EMBO Journal, particularly due to the strength of the screening and genetic interaction studies. However, I have several criticisms that I would like the authors to address, at least in parts experimentally:

1. Claim of an exclusively somatic mechanism: Establishing negative evidence-such as the absence of Rab39 function in the presynaptic terminal-is inherently challenging. The authors mention that they used a line expressing YFP-tagged Rab39 at endogenous levels and found Rab39 exclusively in the neuronal cell body, without significant labeling at synaptic boutons of neuromuscular junctions (NMJs). However, the provided images are not entirely convincing. I recommend quantifying the Rab39 localization, possibly using individual confocal sections, and offering more detailed visuals of the Rab39-positive structures in the cell bodies. By the way: is the endogenously EYFP-Rab39 functional? What do they know about the Rab39 positive spots in the motoneuron somata? Anything co-localizing?

The eYFP-Rab39 has been used before and shown to be functional. We included these references to our main text (Dunst et al, 2015; Miao et al, 2023).

To further address the reviewer's question, we also conducted additional imaging along their recommendation and quantification of Rab39 localization in both neuronal cell bodies and synapses using immunostaining and Airyscan microscopy. We compared GFP signal intensity in w^{1118} (wild type negative control) and eYFP-Rab39 animals using anti-GFP immunostaining, co-labeled with anti-elav and anti-HRP to mark nuclei in neuronal cell bodies and neuronal membranes, respectively, and anti-HRP to label presynaptic compartments at neuromuscular junctions.

In eYFP-Rab39 animals, we observed strong GFP labeling in neuronal cell bodies but no significant signal at synapses, consistent with soma-restricted Rab39 localization. No signal was detected in w^{1118} animals. This experiment is now described in the revised manuscript, and corresponding quantification and representative images are included in Figure 6 (see also Review Response Figure 1).

Review Response Figure 1: Rab39 localizes to neuronal somata in the ventral nerve cord

(A) Representative Airyscan confocal image of the larval ventral nerve cord (VNC) in control (CTRL, w^{1118}) and eYFP-Rab39 animals. VNCs were stained with anti-HRP (magenta) to label neuronal membranes, anti-Elav (orange) to mark neuronal nuclei, and anti-GFP (green) to visualize endogenously tagged eYFP-Rab39. Scale bar: 10 μ m. (B) Representative Airyscan confocal image of a neuromuscular junction (NMJ) boutons labeled with anti-GFP and anti-HRP. Scale bar: 5 μ m. (C) Quantification of GFP mean fluorescence intensity in cell bodies in VNC and NMJ regions. Values are normalized to control (w^{1118}) levels within each analyzed region and shown as fold change. The dotted line indicates the background signal threshold. Statistical test: one-way ANOVA with Tukey's post hoc test; $n = 5$; error bars: mean \pm SD.

To investigate the subcellular localization of *Drosophila* Rab39 within neuronal somata, we examined its spatial relationship with different organelle markers under basal conditions. We thus

analysed the colocalization of Rab39 with markers for the cis-Golgi (GM130 (Nakamura et al, 1995)), trans-Golgi (Syntaxin-16 (Tang et al, 1998)), late endosome-lysosomes (Rab7 (Vitelli et al, 1997)), and (Lamp1 (Eskelinen et al, 2003)). Only a small fraction of Rab39 localized to the cis-Golgi and trans-Golgi compartments, as indicated by the low colocalization with GM130 (~%20) and Syntaxin-16 (~%20). In contrast, our analysis revealed a higher degree of co-localization of Rab39 with Rab7 and Lamp1, with each pair showing approximately 40% Mander's coefficient. This suggests that a larger fraction of the Rab39 pool is associated with the endo-lysosomal pathway compared to the Golgi network. These new data are now included in the revised manuscript in Figure EV5 (see also Review Response Figure 2).

Review Response Figure 2: Rab39 localizes to organelles involved in Golgi–endo-lysosomal trafficking

(A–A', D–D', G–G', J–J') Representative Elyra super-resolution images of third instar larval VNCs showing neuronal cell bodies stained with anti-GFP (eYFP-Rab39, green) and markers for various organelles. Overviews of the VNC are shown in A', D', G', and J', Scale bar: 20 μm ; zoomed-in cell bodies in A, D, G, and J, Scale bar: 3 μm . Co-stains include anti-Rab7 (late endosome/lysosomes, magenta; A), anti-Lamp1 (late endosomes/lysosomes, orange; D), anti-GM130 (cis-Golgi, cyan; G), and anti-Syntaxin 16 (trans-Golgi, blue; J). (B, E, H, K) Pearson's correlation coefficients for Rab39 and each marker, with Costes' randomization threshold (≥ 100 iterations). Statistical test: unpaired t-test; $n = 14$ (Rab7, Lamp1, Syntaxin 16), $n = 16$ (GM130); error bars: mean \pm SD. (C, F, I, L) Manders' overlap coefficients indicating the fraction of Rab39 overlapping with each marker and vice versa.

References

- Dunst S, Kazimiers T, von Zadow F, Jambor H, Sagner A, Brankatschk B, Mahmoud A, Spann S, Tomancak P, Eaton S, et al (2015) Endogenously Tagged Rab Proteins: A Resource to Study Membrane Trafficking in *Drosophila*. *Developmental Cell* 33: 351–365
- Eskelinen E-L, Tanaka Y & Saftig P (2003) At the acidic edge: emerging functions for lysosomal membrane proteins. *Trends Cell Biol* 13: 137–145
- Miao H, Millage M, Rollins KR & Blankenship JT (2023) A Rab39-Klp98A-Rab35 endocytic recycling pathway is essential for rapid Golgi-dependent furrow ingression. *Development* 150: dev201547
- Nakamura N, Rabouille C, Watson R, Nilsson T, Hui N, Slusarewicz P, Kreis TE & Warren G (1995) Characterization of a cis-Golgi matrix protein, GM130. *The Journal of cell biology* 131: 1715–1726
- Tang BL, Low DYH, Lee SS, Tan AEH & Hong W (1998) Molecular Cloning and Localization of Human Syntaxin 16, a Member of the Syntaxin Family of SNARE Proteins. *Biochemical and Biophysical Research Communications* 242: 673–679
- Vitelli R, Santillo M, Lattero D, Chiariello M, Bifulco M, Bruni CB & Bucci C (1997) Role of the Small GTPase RAB7 in the Late Endocytic Pathway *. *Journal of Biological Chemistry* 272: 4391–4397

2. Interpretation of Atg9-positive vesicle transport: The authors observe a higher number of Atg9-positive vesicles in the axons of rab39 mutants, but the transport speed of these vesicles decreases. What is the overall effect of this observation? They should measure flux rates—specifically, the number of Atg9-positive vesicles crossing a certain axonal point normalized over time. In my opinion, they have not conclusively demonstrated an increased flux of Atg9-positive vesicles down the axons in rab39 mutants. There also appears to be a trend of more Atg9-positive vesicles moving retrogradely in the rab39 mutant. Providing imaging trajectories of the Atg9-positive vesicles would strengthen their findings.

While a slight reduction in Atg9 vesicle speed was observed in rab39^{KO} axons, this was not statistically significant and likely reflects limitations of the initial imaging setup we used; we have since performed higher-resolution imaging under optimized conditions to more accurately capture vesicle dynamics, and we now present these data in the reworked manuscript. Importantly, the new data confirm the overall trend observed previously, but now providing improved resolution

without altering the original interpretation.

We performed new experiments using a Zeiss LSM 980 Airyscan system, which provided superior temporal resolution (6.33 seconds per frame) and finer Z-stack resolution (7 slices spanning 3.66 μm total; $\sim 0.52 \mu\text{m}$ intervals). This allowed for more accurate 3D tracking of vesicle movement and minimized the risk of missing fast-moving vesicles due to undersampling. In these improved recordings, the average vesicle speed was consistent between control and *rab39^{KO}* animals, confirming that the previously observed (non-significant) speed difference was likely due to technical limitations of our setup rather than a genuine alteration in transport dynamics (Figure 7J-O, see also Review Response Figure 3).

Regarding vesicle flux, we quantified the number of Atg9-positive vesicles within a defined 144 μm segment of the axon, imaged over 180 seconds. While we appreciate the suggestion to measure vesicles crossing a fixed point, in practice this was technically challenging due to the relatively low frequency of vesicle passages within a small defined region and the variability in movement onset during the imaging window. We chose instead to determine vesicle abundance and behavior by analyzing the total number of vesicles visible within a consistent axonal stretch and time frame across genotypes, which provides a robust comparative measure. Furthermore, we examined vesicle directionality and found that the proportions of anterograde, retrograde, and stationary vesicles were not significantly different between genotypes.

To further address the reviewer's request, we have now also included kymographs and representative time-lapse videos with trajectory overlays (Video 1-2). These visualizations provide direct evidence of vesicle motility patterns and confirm that Atg9-positive vesicle transport dynamics remains largely unaffected in *rab39^{KO}* axons, despite the observed increase in axonal vesicle number.

Review Response Figure 3: *rab39^{KO}* increases the abundance of Atg9 vesicles in axons

(A) Schematic of the larval ventral nerve cord and axons in *Drosophila* third instar larvae, the red rectangle is indicating the regions used for Atg9 vesicle movement imaging. (A–B) Live imaging of control (CTRL: $w^{1118} w^1$) (B) and *rab39*^{KO} (C) animals expressing genomic Atg9-mCherry in the *atg9*^{B5} null background. Scale bars: 5 μm (axon images); 10 μm (kymographs, x-axis). Red arrowheads indicate Atg9-mCherry-positive puncta. (D) Quantification of the number of Atg9-mCherry puncta within a 144 μm segment of axon recorded for 189 s, from the experiment in (B–C). Statistical test: unpaired t-test; $n = 12$; error bars: mean \pm SD. (E) Quantification of the speed ($\mu\text{m}/\text{s}$) of moving Atg9-mCherry puncta in axons from (B–C). Statistical test: Mann–Whitney test; $n = 26–47$; error bars: mean \pm SD. (F) Direction of movement distribution of Atg9-mCherry puncta in control and *rab39*^{KO} axons (from B–C). Statistical test: Fisher’s Exact Test with the Freeman–Halton extension; p-value = 0.7282. Vesicle counts are also provided in Appendix Table 2.

3. Specificity of the effect on Atg9-positive vesicles: How specific is the observed effect on Atg9-positive vesicles in *rab39* mutants? Do other vesicular species to be transported in axons exhibit similar changes?

This serves as an excellent control to assess the specificity of the Atg9 trafficking defect in rab39 mutants. To determine whether this effect is unique to Atg9, we also examined the transport of other axonal cargos.

We focused on two additional markers that also require kinesin-mediated transport: mito-GFP (Pilling et al, 2006) and Syt-GFP (Barkus et al, 2008). To analyze mitochondrial and synaptotagmin trafficking, we expressed UAS-mitoGFP and UAS-Syt-eGFP in motor neurons and performed photobleaching of a defined region of the axon (a technique amply used before) (Nakata et al, 1998; Pilling et al, 2006). This method differs from our approach for Atg9, where individual vesicles are easily trackable due to their sparse distribution. In contrast, the high density of mitoGFP and Syt-GFP organelles preclude single-particle tracking. We therefore instead quantified the number of fluorescent structures crossing into the bleached area and their average speed.

*We found no significant differences between *rab39*^{KO} and control animals in terms of the number of organelles (either synaptic vesicles or mitochondria) trafficking, either in anterograde or retrograde direction. Also the average velocities were not different between *rab39*^{KO} and control (Review Response Figure 4).*

These findings indicate that Rab39 loss does not globally disrupt axonal trafficking, and instead the data support our conclusion that Rab39 has a role in cargo-specific regulation of Atg9-positive vesicle dynamics. This experiment is now described in the revised manuscript, and corresponding quantification and representative images are included in Figure EV4.

Review Response Figure 4: *rab39^{KO}* does not affect synaptic vesicle or mitochondria transport (A, E) Representative pre- and post-bleach images ($t = 0$ and $t = 360$ s) of axons expressing *D42-Gal4 > UAS-mitoGFP* (A) or *UAS-Syt-eGFP* (E) in control and *rab39^{KO}* larvae. Scale bars: 5 μm (A), 20 μm (E). (B, F) Quantification of mitoGFP- and Syt-eGFP-positive particles per defined axon length (53 μm and 157 μm , respectively). Statistical test: unpaired t-test; $n = 10$ (B), $n = 11$ (F); error bars: mean \pm SD. (C, G) Quantification of vesicle speed for mitoGFP ($n = 39$ traces/group from 10 animals) and Syt-eGFP ($n = 59$ [CTRL] and 55 [*rab39^{KO}*] traces from 11 animals). Statistical test: unpaired t-test; error bars: mean \pm SD. (D, H) Direction of vesicle movement (anterograde vs. retrograde) for mitoGFP (D) and Syt-eGFP (H) in control and *rab39^{KO}* axons. Statistical test: Fisher's Exact Test; (D) p-value = 1; (H) p-value = 0.3273 exact counts in Appendix Table 2.

References

Barkus RV, Klyachko O, Horiuchi D, Dickson BJ & Saxton WM (2008) Identification of an Axonal Kinesin-3 Motor for Fast Anterograde Vesicle Transport that Facilitates Retrograde Transport of Neuropeptides. *MboC* 19: 274–283

Nakata T, Terada S & Hirokawa N (1998) Visualization of the Dynamics of Synaptic Vesicle and Plasma Membrane Proteins in Living Axons. *The Journal of Cell Biology* 140: 659–674

Pilling AD, Horiuchi D, Lively CM & Saxton WM (2006) Kinesin-1 and Dynein Are the Primary Motors for Fast Transport of Mitochondria in *Drosophila* Motor Axons. *MboC* 17: 2057–2068

4. Their readout of autophagy (function) nearly exclusively relies on counting ATG8 spots at the

NMJ terminals (apart from the ERGs). For example: are there any physiological changes to be expected as a result of ATG9/Rab39 manipulation at the NMJ terminals?

To further explore the functional consequences of Rab39 manipulation at the NMJ, we recorded miniature and evoked excitatory junctional currents (EJCs) to assess synaptic transmission (Reviewer Response Figure 5). Our data show that *rab39^{KO}* mutants exhibit a significant reduction in EJC amplitude, and this is rescued when we express wild type Rab39 in *rab39^{KO}* mutants. This indicates that the loss of Rab39 ultimately impairs synaptic transmission. Interestingly, overexpression of wild-type Rab39 in a control (wild-type) background also caused a reduction in EJC amplitude, to levels comparable to those observed in *rab39^{KO}* animals. Together, these results underscore the importance of precise Rab39 regulation for proper synaptic function as both loss and overexpression of Rab39 impair neurotransmission. We have now included these functional data in the revised manuscript and added the corresponding quantifications in Figure 4, along with a discussion of this dose sensitivity in the revised text.

Review Response Figure 5: Disrupted Rab39 expression impairs synaptic transmission at the NMJ

(A) Quantification of evoked junctional current (EJC) amplitudes recorded from larval NMJs in 1 mM extracellular Ca²⁺ and 1 Hz stimulation in the following genotypes: control (*w¹¹¹⁸ w⁺*), *rab39^{KO}*, Rab39 wild-type overexpression (*nSyb-Gal4 > UAS-Rab39^{WT}*), and Rab39 rescue (*nSyb-Gal4 > rab39^{KO}; UAS-Rab39^{WT}*). (B) Representative EJC traces corresponding to the genotypes shown in (A). (C) Cumulative probability distribution of mEJC amplitudes for the genotypes shown in (A). (D) Raw data of spontaneous activity (miniature EJC (mEJC)). (E) Quantal content calculated as the ratio of EJC over mEJC amplitudes for the same genotypes shown in (A). (A, E) Statistical test: one-way ANOVA with Dunnett's post hoc test; orange dots represent individual recordings from 7 larvae (control), 5 larvae

(*rab39^{KO}*), 6 larvae (*nSyb-Gal4 > UAS-Rab39^{W1}*), and 6 larvae (*nSyb-Gal4 > rab39^{KO}; UAS-Rab39^{W1}*); error bars: mean \pm SD.

Referee #2:

Summary

In this study, Kilic et al. proposed a possible role for Rab39 in the neuronal autophagic process using a *Drosophila* model system. Based on findings from mutant fly phenotypes, the authors claimed that Rab39, whose human homolog is mutated in Parkinson's disease (PD) and is exclusively expressed in the somatic region, regulates synaptic autophagy by controlling the anterograde transport of Atg9 vesicles. As the authors noted, presynaptic autophagy and the functional roles of Atg9 vesicles are very interesting topics in neuroscience. However, the evidence presented by the authors is somewhat fragmented and not well connected. To solidify this concept, the manuscript needs to address key questions that remain unresolved.

Major concerns

1. The authors claimed that Rab39, which localizes to the somatic region, regulates the autophagic process at synapses by inhibiting the anterograde transport of Atg9 vesicles from the soma to nerve terminals. This concept is quite interesting. However, given the significant distance between the neuronal soma and synapses, it remains unclear how this transport process can effectively support presynaptic autophagy in a timely manner. Additionally, the findings presented in the manuscript are not well connected. For instance, there is no direct evidence demonstrating how the Rab39 knockout (KO)-mediated increase in Atg9 vesicle transport disrupts synaptic autophagy. What is the physiological significance of the increased Atg9 vesicle transport to nerve terminals? Without data directly linking this process to synaptic autophagy, it is difficult to conclude that Rab39-regulated anterograde trafficking of Atg9 vesicles is crucial for presynaptic autophagy and neuronal health.

*We appreciate the reviewer's comment and have further elaborated on this with textual clarification and an important new experiment. We started by conducting a synaptic FRAP assay, assessing the speed of Atg9 delivery to presynaptic terminals in control and *rab39* mutants. We photo-bleached Atg9-GFP specifically at presynaptic boutons and measured presynaptic fluorescence recovery. We find a modest but highly significant faster Atg9-GFP fluorescence recovery in *rab39* mutants compared to controls. These data indicate that Rab39 inhibits Atg9 vesicle delivery to synapses. We have now included these data in the revised manuscript in Figure 8 (See also Review Response Figure 6).*

We also conducted an electrophysiological analysis and find that *rab39^{KO}* NMJs show decreased excitatory junctional currents (EJC) when stimulated at low frequency (Reviewer Response Figure 5 above) while miniature EJCs are not affected. These data indicate that the loss of Rab39 causes functional synaptic defects.

Finally, we further elaborate on our model. We show in the manuscript that the loss of Rab39 causes increased autophagosome formation at synapses and this is associated with an increased number of Atg9-decorated vesicles trafficking in axons (Reviewer Response Figure 3) and faster delivery of Atg9 to presynaptic terminals (Reviewer Response Figure 6). Furthermore, our genetic interaction studies show that heterozygous loss of *atg9* rescues the increased autophagy, indicating Atg9 is limiting for this phenotype. Finally, we show that heterozygous loss of *shot* or *Kif1a/unc104* also rescue the increased presynaptic autophagy in Rab39 knockouts. These data made us propose a model where Rab39-controlled delivery of Atg9 to synapses via axonal transport is limiting for high levels of autophagy at presynaptic terminals. However, this pathway would not acutely regulate autophagosome formation given the long travel distances and relatively slow response time (we made this now clear in our discussion). We cannot exclude that Rab39 loss also initiates other pathways that would independently drive autophagosome formation, and the mere overexpression of Atg9 in wildtype animals is not sufficient to increase synaptic autophagy. Nonetheless, our work is in line with, and further extend studies in *C. elegans* neurons: Atg9 is initially transported from the soma to presynaptic terminals by the motor protein Unc104, which enables long-range, kinesin-mediated transport along axons (Stavoe et al, 2016); our work now shows the important role of Rab39, a protein mutated in Parkinson's disease and located in the soma, to negatively regulate this process. Once Atg9 is at the synapse, it is not static but is actively recycled through endocytic pathways and tightly coupled to the synaptic vesicle cycle, enabling rapid and activity-dependent mobilization for local and controlled autophagosome formation (Yang et al, 2022).

Review Response Figure 6: rab39KO increases Atg9 vesicle delivery to presynaptic terminals
(A) FRAP assay to assess Atg9-GFP mobility within presynaptic boutons. Live imaging of control and *rab39^{KO}* larvae expressing UAS-Atg9-GFP under the control of the *D42-Gal4* motor neuron driver. Representative images show fluorescence recovery after photobleaching of a small region (dotted circle) within an NMJ bouton over a 60-second period. Scale bar: 5 μ m. (B) Fluorescence recovery curves (quantification) of Atg9-GFP signal over time in control and *rab39^{KO}* boutons, fit with a double-exponential model. Statistical test: two-way ANOVA; *n* = 10; error bars: mean \pm SD.

References

Stavoe AKH, Hill SE, Hall DH & Colón-Ramos DA (2016) KIF1A/UNC-104 Transports ATG-9 to Regulate Neurodevelopment and Autophagy at Synapses. *Developmental Cell* 38: 171–185

Yang S, Park D, Manning L, Hill SE, Cao M, Xuan Z, Gonzalez I, Dong Y, Clark B, Shao L, et al (2022) Presynaptic autophagy is coupled to the synaptic vesicle cycle via ATG-9. *Neuron* 110: 824-840.e10

2. Although the authors showed that synaptic protein levels remain unchanged in both control and Rab39-KO conditions, it is important to demonstrate that the anterograde transport of synaptic vesicle proteins is also unaffected in Rab39-KO axons. Ideally, measuring both Atg9 and synaptic vesicle proteins simultaneously within the same axon would provide more comprehensive and convincing evidence.

To address the reviewer's question, we co-expressed endogenously tagged Atg9-mCherry with UAS-Synaptotagmin-GFP, a synaptic vesicle associated protein. We observed almost no colocalization of these two markers in neuronal somata or in axons. Our results are in line with (Binotti et al, 2024) and indicate that Atg9 vesicles in the soma and axons are molecularly distinct from classical transport vesicles with SV proteins (Review response figure 7). We included these data in the revised manuscript in Figure EV4.

To then further ask if the anterograde transport of synaptic vesicle proteins is affected by rab39^{KO} we live imaged UAS-Synaptotagmin-GFP (Syt-GFP) vesicle movement in motor neuron axons. We found no significant difference in the number or speed of synaptotagmin-positive vesicles between control and rab39^{KO} animals. Similarly, rab39^{KO} also did not affect the transport of mitochondria (mito-GFP) (Review Response Figure 4). These results indicate that synaptic vesicle protein trafficking (and mitochondria) remain unaffected in the absence of Rab39, while we do observe an increased number of Atg9 vesicles in axons and faster delivery to synapses (Review Response Figure 6). These data are all included in the revised manuscript.

Review Response Figure 7: Atg9 and synaptic vesicles (Syt) show distinct localization patterns (A–A') Representative live confocal images showing Atg9-mCherry and Syt-eGFP in the ventral nerve cord (VNC), axons, and neuromuscular junctions (NMJs) of *Drosophila* third instar w^{1118} larvae. Red squares in (A) indicate regions enlarged in (A'). White arrowheads mark Atg9-mCherry vesicles; orange arrowheads indicate Syt-eGFP vesicles. Scale bars: 100 μm (A) 40 μm (A') (VNC); 40 μm (A) 20 μm (A') (axon); 40 μm (A), 10 μm (A') (NMJ).

Reference

Binotti B, Ninov M, Cepeda AP, Ganzella M, Matti U, Riedel D, Urlaub H, Sambandan S & Jahn R (2024) ATG9 resides on a unique population of small vesicles in presynaptic nerve terminals. *Autophagy* 20: 883–901

3. The data (Figure 5) show that the anterograde transport of Atg9 vesicles is significantly increased in Rab39-KO neurons. However, what about the total levels of Atg9 at synapses? Are they also increased?

To assess whether total Atg9 levels at synapses are altered in $rab39^{KO}$ neurons, we used endogenously expressed Atg9-3xmCherry and performed immunostainings on larval preparations. Quantification of Atg9 intensity at synaptic boutons revealed no significant difference between $rab39^{KO}$ and control animals (Review response figure 8), indicating that despite enhanced anterograde trafficking and delivery to synapses (FRAP experiments), steady state synaptic Atg9 levels remain similar between control and $rab39^{KO}$. This is consistent with our observation of increased synaptic autophagy in $rab39^{KO}$ mutants in that the increased delivery of Atg9 is balanced by enhanced turnover or utilization of Atg9 at the synapse. These observations align with our finding of increased retrograde transport of Atg8-positive autophagosomes in $rab39^{KO}$ neurons (please see Review response Figure 10 below), further supporting a model of enhanced autophagic flux at synapses and subsequent retrograde transport of autophagosomes. We have now included these data in the revised manuscript in Figure 8.

Review Response Figure 8: Atg9 levels are similar at control and *rab39*^{KO} NMJs

(A) Representative Airyscan confocal images of NMJs in control (*w*¹¹¹⁸ *w*⁺, CTRL) and *rab39*^{KO} animals expressing genomically tagged Atg9-mCherry in *atg9*^{B5} (null) background. Presynaptic boutons were labelled with anti-HRP (magenta), and anti-mCherry (orange). The third image in each row shows the merged channels with a yellow outline indicating the NMJ area. Scale bar: 20 μ m. (B) Quantification of mean fluorescence intensity of Atg9-mCherry at NMJs, normalized to control levels. Statistical test: unpaired t-test; $n = 10$; error bars: mean \pm SD.

Minor concerns

1. For readers unfamiliar with the LIND technique, it would be helpful to provide more details in the text or include a schematic representation in the figure.

We have revised the manuscript to provide a more detailed explanation of the LIND technique in the main text.

2. The high-magnification EM images in Figure 3 should be displayed at a larger size.

The high-magnification EM images have now been enlarged and moved to Figure EV2 to improve clarity and visibility (also Review Response Figure 11).

3. The changes shown in Figures 3J and 3K are very subtle. At the very least, the authors should acknowledge this in the main text.

We have now reprocessed the same images to improve contrast, making the observed changes more visible. These updated versions are presented in the revised Figure 3L.

4. Please label the fluorescent molecule (Atg9-mCherry) in Figures 5J-L to help readers understand the figure without having to refer to the figure legend.

We have now added the Atg9-mCherry label directly in revised Figure 7J to improve clarity and ensure the figure is self-explanatory.

5. In Figure 5O, the graph should display the raw data, as is done in other figures.

Figure 5O represents a single percentage value calculated from the total number of vesicles

pooled from 12 different animals per genotype. As such, individual data points cannot be shown in this case. However, we now provide the exact counts and percentages used to generate the graph in Appendix Table 2.

6. Is the sentence in lines 285-286 accurately conveying what the authors intend to say?

We thank the reviewer for drawing our attention to this. There was indeed a typo in the sentence, the word “decrease” was used instead of “increase.” This has now been corrected in the revised manuscript to accurately reflect our intended meaning.

7. There are a few typos, such as the period in line 71 and the missing parenthesis in line 290.

The typos, including the extra period in line 71 and the missing parenthesis in line 290, have now been corrected in the revised manuscript.

Referee #3:

The manuscript by Kilic and colleagues describes a large-scale genetic modifier screen for suppressors of synaptic autophagy. The authors identify Rab39 and shot as genes of interest, and then examine the effects of Rab39 and shot mutants on autophagy. Trafficking data suggest that both Rab39 and shot participate in the trafficking of Atg9 vesicles to synapses. Based on localization data, the authors propose that Rab39 functions to regulate this pathway in the soma, from either endosomes or lysosomes, while Shot is more likely to function either at the AIS or along the axon, although a function in the soma is not ruled out. The work concludes with a model in which Rab39-mediated trafficking in the soma orchestrates a cross-compartmental mechanism to regulate autophagy abundance or flux at synapses.

This work addresses a pathway implicated in the regulation of neuronal homeostasis, and reports some new observations that are likely to be of interest to the field. However, significant revisions are required prior to publication. The logic flow is unclear in many sections, some of the data shown are less than convincing, many conclusions are not strongly supported by the data provided, and key aspects of the authors' model remain unproven. The work is also a bit myopic, building on too narrow a base by not fully incorporating progress from work from other labs on autophagy in *Drosophila*, *C. elegans*, and other model systems. These points are all addressable, and substantial revisions would make this work a much stronger contribution to the literature.

Specific points to address:

1. The title states that there is Rab39-regulated anterograde trafficking of Atg9, but this is not

directly assessed in the work; rather, it is a conclusion by inference. The title should more accurately reflect what is actually being measured.

We have revised the title to more accurately reflect our experimental findings:

Revised title: Soma-restricted Rab39 inhibits Atg9-dependent autophagy at presynaptic terminals and prevents dopaminergic synapse degeneration.

This title is factual and keeps our observation that Rab39 is restricted to the neuron soma (we also included new data) while the main phenotype we investigate is increased autophagy at synapses. It also captures the link with Atg9 vesicle delivery to presynaptic terminals (also based on new FRAP data) and is consistent with our results that show genetic interactions between rab39 and shot, unc104 & atg9.

2. Autophagy is a highly conserved pathway in general, and neuronal autophagy also appears to be conserved across fly, worm, and mouse models. However, both the introduction and the analysis presented in this work fail to take advantage of the extensive literature documenting mechanisms of autophagosome formation and maturation, previous work on the synaptic proteins that are degraded by synaptic autophagy, and previous work on the role of Kif1A in Atg9 transport. For example, synaptic autophagosomes have been shown to mature during trafficking in fly models (Neisch et al., 2017), but this possibility is not considered here although it likely affects some of their conclusions. While some aspects of synaptic autophagy may be distinct between fly and vertebrate models, such as the effects of starvation on neuronal autophagy or the effects of mutant LRRK2 on autophagosome formation, it would be helpful to describe these differences more clearly rather than omitting them from the discussion of the results in the current study, especially given their interest in using *Drosophila* as a model for human disease mechanisms.

In the revised manuscript, we have expanded the Introduction and Discussion to incorporate the following aspects:

*1. We now include prior studies including (Neisch et al, 2017) that demonstrate that autophagosomes mature during their retrograde transport in *Drosophila*. This is particularly relevant, as in new data we show increased retrograde trafficking of Atg8-positive vesicles in axons of rab39 mutants (see below point 5, and Reviewer Response Figure 10). Together with the modest but significant increase of Atg8-mCherry/Atg8-GFP ratio the data suggest that in rab39 mutants more autophagosome form, are acidified and are retrogradely transported to the soma.*

*2. We now discuss additional earlier work showing that Kif1A (and its *Drosophila* ortholog Unc104) was shown to play a critical role in anterograde transport of Atg9 vesicles (Stavoe et al, 2016). Our own finding, that heterozygous loss of unc-104 or shot suppress synaptic autophagy in rab39^{KO} are now discussed in this broader context, and we highlight that our data build upon and extend*

these findings by identifying the soma-restricted regulator Rab39.

3. We present a model that is based on our data and left the more speculative aspects out. We also now present this model in the broader context of existing literature.

4. We have added a discussion on differences in observations related to synaptic autophagy between different systems, cells/cell types and species, and the relatively milder effects of starvation on neuronal autophagy in specific cell culture systems in vitro as compared to NMJs in flies.

5. We also add that Drosophila is a powerful in vivo system to dissect mechanistic regulators of conserved autophagy pathways, keeping in mind there are different types of regulation and autophagy usage between species and conditions.

References

Neisch AL, Neufeld TP & Hays TS (2017) A STRIPAK complex mediates axonal transport of autophagosomes and dense core vesicles through PP2A regulation. Journal of Cell Biology 216: 441–461

Stavoe AKH, Hill SE, Hall DH & Colón-Ramos DA (2016) KIF1A/UNC-104 Transports ATG-9 to Regulate Neurodevelopment and Autophagy at Synapses. Developmental Cell 38: 171–185

3. The authors report an interesting genetic screen to identify components regulating synaptic autophagy, which relies on exposing flies to constant light. I'm curious – what are the effects of this assay on circadian rhythms, as there has been some thought that circadian rhythms regulate autophagy?

It is indeed established that autophagy is subject to circadian regulation in both mammals and Drosophila (Szypulski et al, 2024; Ma et al, 2012; Pastore et al, 2019). While our study was not designed to directly assess circadian regulation of autophagy, we took steps to ensure that our genetic screen was not confounded by circadian effects:

-Our screen was based on the light-induced neurodegeneration (LIND) assay, a sensitized background in which flies carrying the phosphomimetic endoA^{S75D} construct exhibit increased synaptic autophagy and are prone to light-dependent neurodegeneration, as previously demonstrated (Soukup et al, 2016; Bademosi et al, 2023). This results in a robust and quantifiable reduction in ERG depolarization defect, allowing us to identify genetic suppressors of the phenotype.

-To standardize sensitivity to light and minimize variability, we performed the screen in the cn bw mutant background, which causes the flies to have white eyes and renders them more susceptible to LIND (Escobedo et al, 2022; Tearle, 1991). Crucially, both control and experimental genotypes in the screen shared the same genetic background and were exposed to identical constant light conditions. This uniform design ensures that circadian effects or light-induced effects are

consistent across all genotypes. We have now clarified these points in the revised manuscript.

References

Bademosi AT, Decet M, Kuenen S, Calatayud C, Swerts J, Gallego SF, Schoovaerts N, Karamanou S, Louros N, Martin E, et al (2023) EndophilinA-dependent coupling between activity-induced calcium influx and synaptic autophagy is disrupted by a Parkinson-risk mutation. *Neuron* 111: 1402-1422.e13

Escobedo SE, Stanhope SC, Dong Z & Weake VM (2022) Aging and Light Stress Result in Overlapping and Unique Gene Expression Changes in Photoreceptors. *Genes* 13: 264

Ma D, Li S, Molusky MM & Lin JD (2012) Circadian autophagy rhythm: a link between clock and metabolism? *Trends in Endocrinology & Metabolism* 23: 319–325

Pastore N, Vainshtein A, Herz NJ, Huynh T, Brunetti L, Klisch TJ, Mutarelli M, Annunziata P, Kinouchi K, Brunetti-Pierri N, et al (2019) Nutrient-sensitive transcription factors TFEB and TFE3 couple autophagy and metabolism to the peripheral clock. *The EMBO Journal* 38: e101347

Soukup SF, Kuenen S, Vanhauwaert R, Manetsberger J, Hernández-Díaz S, Swerts J, Schoovaerts N, Vilain S, Gounko NV, Vints K, et al (2016) A LRRK2-Dependent EndophilinA Phosphoswitch Is Critical for Macroautophagy at Presynaptic Terminals. *Neuron* 92: 829–844

Szypulski K, Tyszka A, Pyza E & Damulewicz M (2024) Autophagy as a new player in the regulation of clock neurons physiology of *Drosophila melanogaster*. *Sci Rep* 14: 6085

Tearle R (1991) Tissue specific effects of ommochrome pathway mutations in *Drosophila melanogaster*. *Genetics Research* 57: 257–266

4. Line 153 states that Atg8 marks the formation and maturation of autophagosomes – this is not exactly true. More importantly, the signals that the authors are quantitating are very difficult to see in the images provided here. Given this issue, and the large variability in signal (see Figure 5G), it would be good to include larger galleries of the primary images in the supplement, with the points being quantitated clearly labeled.

-We have revised the text in line 153 to more accurately reflect the identity of the Atg8 foci: “Atg8 is a cytosolic protein that becomes conjugated to autophagosomal membranes upon initiation of autophagy and thus serves as a marker for autophagosome formation and the presence of autophagosomes.”

-We no longer refer to Atg8 as marking “maturation” of autophagosomes, as that function is more accurately associated with downstream effectors such as fusion with lysosomes and acidification.

-We also acknowledge the reviewer’s concern regarding the Atg8 signal we show in our images. To address this, we have enhanced image contrast and brightness uniformly across images for clarity, while preserving raw data integrity see in Review Response Figure 9.

Review Response Figure 9: Representative examples of Atg8-positive autophagosomes at NMJs across genotypes shown in 'old' Figure 5G, now Figure 7G in the revised manuscript.

Five representative NMJs per genotype are shown to illustrate Atg8-mCherry positive puncta observed and quantified in Figure 5G. Synaptic boutons were thresholded and outlined in red using the Wand Tool in Fiji to quantify the synaptic bouton area. Atg8-mCherry positive puncta were indicated with red arrowheads. Grayscale levels were adjusted to discern the NMJs and Atg8-mCherry signals. For visualisation, an inverted Orange Hot lookup table was applied across all images. Scale bars: 5 μ m.

5. The authors report more puncta in Rab39KO neurons, although this is somewhat difficult to appreciate in the images shown in Figure 3. This could be due to faster or more frequent formation, or slower trafficking away from the synapse, or slower turnover. The authors report a small increase in the ratio of magenta over green puncta, which is consistent with a small change in either decreased trafficking or maturation or both.

To investigate whether the increased number of Atg8-positive puncta at synapses in rab39^{KO} animals is the consequence of impaired retrograde trafficking, we examined the transport of autophagosomes from synaptic terminals (Review response figure 10). We performed live imaging of Atg8-positive vesicles in axons immediately adjacent to type Ib boutons and quantified vesicle motility. The total number of Atg8-positive vesicles undergoing retrograde movement in axons was elevated in rab39^{KO} animals compared to controls in this assay (Reviewer Response Figure 10). Vesicle speed was not significantly different between genotypes. Together with our observation of the modest, but significant increase in the Atg8-mCherry/Atg8-GFP ratio (GFP is quenched in acidic organelles) in rab39^{KO}, the data suggest that in rab39^{KO} more autophagosomes form, they acidify and are transported in a retrograde fashion to the soma. These data are now also included in the new manuscript in Figure 3.

Review Response Figure 10: Synaptic autophagosomes in *rab39^{KO}* neurons are retrogradely transported to the soma

(A) Representative images of Atg8-mCherry puncta in motor axons from third instar control (*w¹¹¹⁸ w⁺*) and *rab39^{KO}* larvae (top). Arrowheads indicate Atg8-mCherry-positive puncta and corresponding kymographs (bottom) from time-lapse imaging (180 s) of the same axons. Diagonal lines represent motile vesicles; vertical lines represent stationary vesicles. Scale bars: 5 µm (axon images), 10 µm (kymographs, x-axis). (B) Quantification of Atg8-mCherry puncta per axon length from the experiment in (A). Statistical test: Welch's t-test; *n* = 14–15; error bars: mean ± SD. (C) Fractional distribution of vesicle direction (retrograde, stationary) pooled across axons from control and *rab39^{KO}* animals, respectively. Statistical test: Fisher's Exact Test; *p*-value = 0.1755. Vesicle counts are listed in Appendix Table 2. (D) Quantification of the speed (µm/s) of Atg8-mCherry-positive vesicles from (A). Statistical test: Mann-Whitney test; *n* = 3–10; error bars: mean ± SD.

6. In Figure 3H, the authors use CLEM to verify that the puncta they are tracking are indeed autophagosomes. These images are not particularly convincing. The first panel appears to be unaligned with the subsequent panels, and the zoom-in images are not strikingly clear. Cleaner

and/or more examples would strengthen this point.

We have now improved the alignment of the panels and adjusted the presentation for clarity. The updated and better-aligned CLEM images are now included in Figure EV2 to support the identification of autophagosomes more convincingly.

Review Response Figure 11: CLEM of boutons of rab39KO animals expressing Atg8-mCherry

(A) Maximum intensity projection of confocal slices of an example NMJ1 displaying an Atg8-mCherry-positive structure (arrowhead). Scale bar: 5 μm . (B) Zoomed-out maximum projection of the same NMJ1 after branding shown in (A). Arrowhead indicates the bouton corresponds to the bouton in (A). Scale bar: 20 μm . (B') Electron micrograph of the same region as in (A). Scale bar: 10 μm . (B'') Overlay of confocal image in (B) with the electron micrograph in (B'). (C, C') Zoomed views of the bouton containing the Atg8-mCherry structure shown as overlay (C) and EM alone (C'). The red square highlights the structure corresponding to the mCherry signal. Scale bar: 2 μm . (D–D''') Single TEM slices showing the putative autophagosomal structure (arrowhead), visible in multiple consecutive sections. Scale bar: 1 μm . (E, E') Overlay of a fluorescence image section from NMJ2 with the corresponding EM image. Arrowheads indicate the structure corresponding to the mCherry signal. Scale bar: 2 μm . (F) Zoom of the structures highlighted in (E). Arrowheads indicate the putative autophagosomal structure. (G, G') Overlay of a fluorescence image section from another NMJ3 with its corresponding EM image. Arrowheads indicate the structure corresponding to the mCherry

signal. Scale bar: 2 μ m. (H) Zoom of the structures shown in (G). Arrowheads indicate the putative autophagosomal structure.

7. Similarly, it is really hard to appreciate the indicated changes in the primary data shown in Figure 3L. Again, more and better examples would strengthen this point.

To improve clarity, we have now focused specifically on the bouton area and displayed the red and blue fluorescent channels separately as well as the ratio between them. This adjustment makes the indicated changes more discernible in the revised Figure 3N, see also Review Response figure 12.

Review Response Figure 12: Enhanced visualization of Fluorescent Timer signal in NMJ boutons

(A) Live imaging of control and *rab39^{KO}* animals expressing the photoconvertible FT::nSyb construct under the *nSyb-Gal4* driver in NMJ boutons. To improve clarity, red and blue fluorescent channels are shown separately, along with their ratio, specifically focused on the bouton area. Scale bar: 5 μ m.

8. The authors report seeing no changes in the absolute levels of BRP, CSP, and syntaxin1A, but have these proteins been reported to be turned over by synaptic autophagy? Querying proteins identified in proteomic screens might be helpful here.

*To address whether BRP, CSP, and Syntaxin1A are substrates of synaptic autophagy, we examined published proteomic datasets profiling autophagic vesicles in neuronal systems. Homologs of BRP (ERC2) and Syntaxin1A (STX1A) have indeed been detected in LC3-positive autophagic vesicles isolated from mammalian brain tissue (Goldsmith et al, 2022; Kallergi et al, 2023). Moreover, both ERC2 and STX1A accumulate in cultured neurons upon pharmacological inhibition of autophagosome formation and flux, suggesting that they can be turned over via autophagy under physiological conditions (Kallergi et al, 2023). In contrast, homologs of CSP (DNAJC5/5B) have not been identified in these datasets, indicating that CSP may not be a canonical autophagy substrate. Despite synaptic autophagy being upregulated in *rab39^{KO}* animals, we do not observe significant changes in the steady-state level of BRP and Syntaxin1A. A similar situation we observed with *Atg9*, where immunostaining did not reveal changes in abundance (See Review Response Figure 8, Figure 8 in revised manuscript), while FRAP experiments of *Atg9* at synapses show increased delivery of *Atg9* positive vesicles in *rab39^{KO}* mutants (See Review Response Figure 6, Figure 8 in*

revised manuscript). Furthermore, our fluorescent timer analysis of neuronal Synaptobrevin (nSyb) indicates a shift toward younger proteins in *rab39^{KO}* animals (Figure 3N-O in revised manuscript), consistent with enhanced protein turnover. These findings suggest that steady-state protein levels may not reliably indicate altered autophagic flux at these NMJ synapses. Instead, turnover dynamics are more evidently captured by assays such as FRAP and fluorescent timers. We have clarified these points in the revised Discussion section.

References

Goldsmith J, Ordureau A, Harper JW & Holzbaur ELF (2022) Brain-derived autophagosome profiling reveals the engulfment of nucleoid-enriched mitochondrial fragments by basal autophagy in neurons. *Neuron* 110: 967-976.e8

Kallergi E, Siva Sankar D, Matera A, Kolaxi A, Paolicelli RC, Dengjel J & Nikolettou V (2023) Profiling of purified autophagic vesicle degradome in the maturing and aging brain. *Neuron* 111: 2329-2347.e7

9. The conclusion that human Rab39B can compensate for the loss of Rab39 would not be strictly correct unless the first exon encodes the entire protein. Can the authors verify this, or moderate their conclusion?

In our Rab39 humanization strategy, we generated knock-in lines by replacing the first exon of the endogenous Drosophila rab39 gene with either the entire wild-type human RAB39B cDNA or the pathogenic RAB39B^{T168K} human cDNA variant, both including the stop codon. By knocking these cDNAs into the first exon, these cDNAs are expressed under control of the endogenous fly rab39 promoter and regulatory elements (all introns are still present, but the rest of the fly gene is not expressed). A schematic of this knock-in design is included in the revised manuscript in Figure EV3 (see also Review Response Figure 13) to clarify the genetic strategy.

Importantly, deletion of the first exon of rab39 is sufficient to create a full loss-of-function allele, as it contains the translation start site and N-terminal coding region. We confirmed this via qPCR, showing complete loss of Rab39 transcript in the knockout line (Figure EV3, see also Review Response Figure 13).

Functionally, we observed that the wild-type human RAB39B cDNA knock in into the rab39^{KO} line partially rescues the increased autophagosome formation, while the RAB39B^{T168K} knock in variant fully phenocopies the rab39^{KO} phenotype. This is consistent with previous studies showing that the T168K mutation leads to reduced RAB39B protein stability and functional inactivation, effectively mimicking a knockout (Gao et al, 2020; Wilson et al, 2014).

Together, these results support the interpretation that human RAB39B can functionally substitute for the fly gene in the regulation of synaptic autophagy. To reflect this more accurately, we have revised the manuscript to state:

“Wild-type human RAB39B compensates for the loss of fly rab39 in regulating synaptic autophagy,

while the *RAB39B*^{T168K} variant mimics the knockout phenotype, consistent with its reported loss-of-function effect.”

Review Response Figure 13: Genetic strategy to create *Rab39* knock out and pathogenic knock in alleles (A-B) Schematic of the *rab39* knock-out (A) and knock-in (B) strategy. The first exon of *Drosophila rab39* was replaced with an attP-flanked mini-white cassette via CRISPR/Cas9-mediated homologous recombination, generating a null allele (*rab39*^{KO}). The chromosomal insertion site is indicated. This mini-white cassette was then replaced by the human wild type or pathogenic mutant cDNA (including a stop codon) to create “human knock in alleles”. (C) *rab39* mRNA expression levels measured by RT-PCR. Expression is shown relative to endogenous *Drosophila rab39* transcript levels.

References

Gao Y, Martínez-Cerdeño V, Hogan KJ, McLean CA & Lockhart PJ (2020) Clinical and Neuropathological Features Associated With Loss of RAB39B. *Mov Disord* 35: 687–693

Wilson GR, Sim JCH, McLean C, Giannandrea M, Galea CA, Riseley JR, Stephenson SEM, Fitzpatrick E, Haas SA, Pope K, et al (2014) Mutations in RAB39B cause X-linked intellectual disability and early-onset Parkinson disease with α -synuclein pathology. *American Journal of Human Genetics* 95: 729–735

10. The conclusion that EndoA acts downstream of Rab39 seems weak based on the experiment reported in Figure 3N-P. What if EndoA initiates autophagosome formation and Rab39 regulates subsequent steps in assembly or maturation? Given the multiple possibilities, this seems like too strong a conclusion.

Our model is based on the respective localizations of Rab39 and EndoA: Rab39 acts in the soma to negatively regulate the abundance of Atg9 vesicles trafficked to the synapse, whereas EndoA

functions locally at the synapse to facilitate autophagosome formation. Given that Atg9 vesicles are known to serve as a membrane source for autophagosome biogenesis (Yamamoto et al, 2012), and that Atg9 has been reported to act as a lipid scramblase involved in autophagosome membrane expansion (Matoba et al, 2020), we propose that increased Atg9 vesicle abundance in rab39^{KO} synapses might provide excess membrane substrate, enabling (EndoA-dependent) autophagosome formation.

This interpretation is supported by our observation that expressing an autophagy defective (but endocytosis supportive) EndoA mutant in the rab39^{KO} background suppresses the elevated synaptic autophagy phenotype (Figure 3P-R), consistent with the idea that EndoA facilitates the conversion of the increased Atg9 delivery to synapses into autophagosomes.

However, we do acknowledge that the specific molecular relationship between Rab39 and EndoA is not directly demonstrated, and that alternative possibilities such as sequential or parallel contributions to different stages of autophagy are also possible.

To reflect this, we have revised the wording in the manuscript to remove the definitive “upstream/downstream” designation and instead state that:

“This indicates that the loss-of-function phenotype of endoA^{D265A} overrides the effect of Rab39 deletion. In genetic terms, this suggests that endoA^{D265A} is epistatic to rab39.”

We hope the reviewer agrees this revised interpretation more accurately aligns with the current evidence.

References

Matoba K, Kotani T, Tsutsumi A, Tsuji T, Mori T, Noshiro D, Sugita Y, Nomura N, Iwata S, Ohsumi Y, et al (2020) Atg9 is a lipid scramblase that mediates autophagosomal membrane expansion. Nat Struct Mol Biol 27: 1185–1193

Yamamoto H, Kakuta S, Watanabe TM, Kitamura A, Sekito T, Kondo-Kakuta C, Ichikawa R, Kinjo M & Ohsumi Y (2012) Atg9 vesicles are an important membrane source during early steps of autophagosome formation. Journal of Cell Biology 198: 219–233

11. The title of Figure 5 is that Rab39 regulates presynaptic autophagy by the transport of Atg9 vesicles. But the authors don't actually measure the trafficking of Atg9 vesicles, so it is unclear how they can make this strong conclusion. I agree that it is a likely possibility, but one that requires more direct proof.

We agree and to address this further we assessed Atg9 vesicle dynamics in axons by quantifying the total number of Atg9-positive vesicles within a defined 144 μm axonal segment over a 180-second time window. We evaluated vesicle abundance, motility, and directionality within a consistent imaging volume and timeframe across genotypes. This approach allowed us to compare overall trafficking behavior between control and rab39^{KO} animals. Our analysis revealed

an increased number of Atg9-positive vesicles in rab39^{KO} axons, but no significant change in the proportion of anterograde, retrograde, or stationary vesicles, nor in vesicle speed. These data suggest that Atg9 vesicle trafficking itself is not impaired in rab39^{KO} animals, despite the elevated vesicle abundance in axons. To make this clearer we have now also included kymographs and representative time-lapse videos with trajectory overlays in the revised manuscript (Figure 7J-O, Video 1-2; Please see Review Response Figure 3, above).

*Second, we also conducted FRAP experiments to directly assess the delivery of Atg9 vesicles at pre-synaptic terminals (Please see Review Response Figure 6, above). We bleached the Atg9 in presynaptic boutons and quantified the speed of recovery as a measure of Atg9 delivery. These experiments indicate faster fluorescence recovery in rab39^{KO} mutants, consistent with more Atg9 being delivered to synapses. These new data were also included in the reworked paper (Figure 8). Finally, also considering the reviewer's comment, we have revised the title of Figure 5 (Revised manuscript Figure 7) to more accurately reflect the data presented: **"Rab39 increases axonal Atg9 vesicle abundance and delivery to synapses."***

12. The experiments in Figure 6 are confusing to interpret, and lack necessary controls. Is Rab7 a specific marker for endosomes and Lamp1 a specific marker for lysosomes in the fly, unlike in other organisms? If so, it should be straightforward to show co-staining with distinct and nonoverlapping distributions. This would better support the interpretation that there is a change in Rab39 localization under the two different conditions, but additional data from an orthogonal assay (biochemistry or something similar) would be required to solidify this conclusion.

We thank the reviewer for this important point and acknowledge that Lamp1 can also label late endosomal compartments, especially given the dynamic nature of the endo-lysosomal system (Cheng et al, 2018). To address the reviewer's remark, we first assessed localization of Rab7 and Lamp1 in motor neuron cell bodies under our specific experimental conditions. This revealed largely separate, though partially overlapping structures (see Review Response Figure 14), consistent with late endosomes and lysosomes to be dynamic and transiently interacting organelles (Bright et al, 2016; Luzio et al, 2000). Pearson correlation analysis indicated non-random colocalization, and Manders' fraction analysis showed that approximately 40% of Lamp1-positive compartments overlapped with Rab7 and vice versa. Hence, as noted by the reviewer, Rab7 and Lamp1 label the biological continuum between late endosome- and lysosomal compartments, respectively. We have clarified this point in the revised manuscript and included these data as Appendix Figure S3.

While we appreciate the suggestion to include biochemical data, such experiments are particularly challenging in Drosophila larvae (or specifically in motor neurons). The small tissue volume and

cellular heterogeneity of the organism make it difficult to isolate pure fractions to assess late endosomal and lysosomal compartments. Moreover, the dynamic and transient nature of endosome-to-lysosome trafficking complicates detection by bulk biochemical methods.

Review Response Figure 14: Rab7 and Lamp1 localize to partially overlapping compartments (A–A') Representative images of neuronal cell bodies in the ventral nerve cord (VNC) of *Drosophila* third instar larvae acquired using Elyra super-resolution microscopy. (A') Overview of the VNC; the highlighted cell body is shown at higher magnification in (A), stained with anti-Rab7 (late endosomes, magenta) and anti-Lamp1 (lysosomes, orange). Scale bar in (A'): 20 μ m, Scale bar in (A): 3 μ m (B) Quantification of Pearson's correlation coefficients for Rab7 and Lamp1 colocalization, compared to Costes' randomization threshold (over 100 iterations). Statistical test: unpaired t-test; $n = 14$; error bars: mean \pm SD. (C) Manders' colocalization coefficients showing the fraction of Rab7 overlapping with Lamp1 (Rab7/Lamp1) and Lamp1 overlapping with Rab7 (Lamp1/Rab7).

References

Bright NA, Davis LJ & Luzio JP (2016) Endolysosomes Are the Principal Intracellular Sites of Acid Hydrolase Activity. *Curr Biol* 26: 2233–2245

Cheng X-T, Xie ,Yu-Xiang, Zhou ,Bing, Huang ,Ning, Farfel-Becker ,Tamar & and Sheng Z-H (2018) Revisiting LAMP1 as a marker for degradative autophagy-lysosomal organelles in the nervous system. *Autophagy* 14: 1472–1474

Luzio JP, Rous BA, Bright NA, Pryor PR, Mullock BM & Piper RC (2000) Lysosome-endosome fusion and lysosome biogenesis. *J Cell Sci* 113 (Pt 9): 1515–1524

13. Several of the conclusions drawn in the Discussion section seem too definitive, given the limited data provided in support. For example, have the authors really demonstrated a pivotal role for Rab39 in organizing vesicular trafficking within the neuronal soma towards lysosomes? Their model that Rab39 regulates the loading of Atg9-positive vesicles is plausible, but not demonstrated. And as they note in their further discussion of this model, many elements remain speculative. For example, they cannot explain why Rab39 mutations do not affect all Kif1a trafficking, and they cannot clarify whether Rab39 and shot are working in the soma or the AIS.

In sum, a more conservatively written paper that includes only rigorously supported conclusions would be a stronger contribution to the literature than the manuscript in its current form.

In response to this (well-received) comment, we have carefully revised the Discussion section throughout to ensure that our conclusions are more cautiously and accurately framed. Rather than presenting Rab39's role as definitive or "pivotal," we now emphasize that:

- Our data support a regulatory role for soma-localized Rab39 in limiting Atg9 vesicle availability at synapses;*
- The functional relation between Rab39, Shot, and Unc-104 is suggestive of a shared pathway;*
- And several aspects of the model such as site of action (soma vs. AIS) and selectivity or cargo-loading remain to be tested in the future.*

We believe these revisions result in a more measured and focused interpretation of our data, and we appreciate the reviewer's guidance in strengthening the manuscript's clarity and impact.

Dear Patrik,

Thank you for submitting your revised manuscript (EMBOJ-2024-119885R) to The EMBO Journal for our consideration, and for your patience during peer review. Your manuscript has now been seen by the original referees #1 and #3, who had previously assessed the initial version of your manuscript, and we have received their comments (included below).

I am very pleased to say that both referees are very satisfied with the revision, explain that the initially raised criticisms and concerns have been thoroughly and successfully addressed, point out that the manuscript has been significantly strengthened with the addition of new and convincing data, mention that the text has also been appropriately revised to provide balanced contextualization and discussion of the findings, and recommend publication of the manuscript. In light of this very supportive input, I am glad to inform you that your manuscript has been in principle accepted for publication in The EMBO Journal. Congratulations on an excellent work and a very successful revision.

There are only two requests for minor improvements from ref. #2, which we kindly ask you to address in a final version of your manuscript, and in a brief point-by-point response to these comments.

From the editorial side, there are also a few changes and corrections we need you to make in the final version of the manuscript before we can proceed with its publication:

- Please note that the funding information provided in the Acknowledgements section of the manuscript should be identical to that specified in our manuscript tracking system (eJP); currently, the following information is missing from eJP: PhD fellowships from FWO Vlaanderen (1165925N; 151095; 11O3225N); postdoctoral fellowship from FWO Vlaanderen (1282123N); ERC, the Chan Zuckerberg Initiative, a Methusalem grant from the Flemish government, and FWO Vlaanderen.
- Please provide the specific URL for the deposited genome sequencing data in the Data availability section of your manuscript, and make sure that the data will be publicly available at the time of publication.
- The author contributions statement should be removed from the manuscript file. Instead, we use CRediT to specify the contributions of each author in the journal submission system. Please feel free to use the free text box to provide more detailed descriptions during submission. See also our guide to authors for more information: <https://www.embopress.org/page/journal/14602075/authorguide#authorshipguidelines>.
- We noticed that callouts for Figure EV1 and Appendix Table S3 are missing.
- Please remember that callouts should be listed sequentially.
- Please make sure to name in the last column of your Author Checklist the section(s) of the manuscript where the information can be found for each positive answer you have provided in this checklist (a few are currently missing).
- Please make sure that your Reagents and Tools Table is uploaded in .docx rather than PDF format.
- Source file name, title, legend and manuscript callout all need to be updated to "Table EV1" instead of Expanded View Table 1; the legend should be placed above the table in the Excel file.
- Please rename the movie files (and the corresponding callouts) to "Movie EV1" and "Movie EV2"; for each one, the legend should be zipped together with the movie file.
- During our routine data checks, our data editors have raised the following queries regarding figures, data, and legends. Please make sure that all requests below are completely addressed in the final version of your manuscript (please highlight all changes in the revised manuscript):
 1. Legends for Figures S4, S5 are absent, this needs to be rectified.
 2. Please note that the exact p values must be provided in the legends of Figures 1B, D; 3G, M; 5F, 6C, 7H, EV5 B, E; S1 F, S2 A'-C'; S3B.

Please also note that as part of the EMBO publications' Transparent Editorial Process, The EMBO Journal publishes online a Peer Review File along with each accepted manuscript. This File will be published in conjunction with your paper and will include the referee reports, your point-by-point response and all pertinent correspondence relating to the manuscript. You can opt out of this by letting the editorial office know (contact@embojournal.org). If you do opt out, the Peer Review File link will point to the following statement: "No Peer Review File is available with this article, as the authors have chosen not to make the review process public in this case."

We look forward to seeing a final version of your manuscript as soon as possible. Please let us know if you have any questions and use this link to submit your revision: <https://emboj.msubmit.net/cgi-bin/main.plex>.

Best regards,

Ioannis

Referee #1:

The authors have responded very thoroughly to the concerns raised in the initial round of review. In particular, I appreciate the comprehensive set of new experiments that address the questions of Rab39 localization, Atg9 vesicle dynamics, and the specificity of cargo transport. The improved imaging, quantification, and functional assays (including FRAP and electrophysiology) all strengthen the mechanistic conclusions.

I was especially pleased to see the effort invested in clarifying Rab39's subcellular localization and in providing additional evidence for its role as a soma-restricted regulator of synaptic autophagy. The new imaging data convincingly demonstrate the absence of Rab39 at synapses, and the quantification of Atg9 vesicle abundance and dynamics across multiple parameters offers a well-rounded view of this trafficking process. Moreover, the demonstration that other cargo types (mitochondria, synaptic vesicle markers) are unaffected reinforces the specificity of Rab39's role in Atg9 regulation.

The manuscript has also benefited from significant textual revisions that enhance clarity and conceptual framing. The authors have taken care to revise overstatements, contextualize their findings in relation to existing literature (including work in *C. elegans* and *Drosophila*), and refine their model accordingly. This results in a much stronger, more balanced contribution to the field.

In light of these substantial and constructive revisions, I am happy to support publication of this manuscript in the EMBO Journal.

Referee #3:

The manuscript by Kilic et al., is significantly improved. The authors made substantial experimental efforts that addressed most of the concerns raised regarding the initial submission. Further, they have expanded their acknowledgment of previous works that are directly relevant to their findings.

However, two of our initial critiques still stand:

1. In regard to the updated Figure 9. The data addressing the relationship/overlap between Rab39 and either Rab7 or Lamp1 are unconvincing. The example images suggesting changed distributions are not compelling, and the quantitative analyses of these data do not readily align with the primary data. We suggest removing this data from the manuscript entirely - it does not add to the strength of the work.
2. Several of the included CLEM images are more convincing, notably panel C-D and G-H. However, some specifics are still unclear. Is panel B' the zoomed in region of A as stated in the figure legend or the same as panel B as indicated by the overlay? Do these zoomed out panels correspond to any of the subsequent areas in the figure? If so please indicate. Panel H should have a scale bar.

Overall, the manuscript is both significant and rigorous and should be published following minor revisions.

Referee #1:

The authors have responded very thoroughly to the concerns raised in the initial round of review. In particular, I appreciate the comprehensive set of new experiments that address the questions of Rab39 localization, Atg9 vesicle dynamics, and the specificity of cargo transport. The improved imaging, quantification, and functional assays (including FRAP and electrophysiology) all strengthen the mechanistic conclusions.

I was especially pleased to see the effort invested in clarifying Rab39's subcellular localization and in providing additional evidence for its role as a soma-restricted regulator of synaptic autophagy. The new imaging data convincingly demonstrate the absence of Rab39 at synapses, and the quantification of Atg9 vesicle abundance and dynamics across multiple parameters offers a well-rounded view of this trafficking process. Moreover, the demonstration that other cargo types (mitochondria, synaptic vesicle markers) are unaffected reinforces the specificity of Rab39's role in Atg9 regulation.

The manuscript has also benefited from significant textual revisions that enhance clarity and conceptual framing. The authors have taken care to revise overstatements, contextualize their findings in relation to existing literature (including work in *C. elegans* and *Drosophila*), and refine their model accordingly. This results in a much stronger, more balanced contribution to the field.

In light of these substantial and constructive revisions, I am happy to support publication of this manuscript in the EMBO Journal.

We sincerely thank the reviewer for their thoughtful and constructive feedback.

Referee #3:

The manuscript by Kilic et al., is significantly improved. The authors made substantial experimental efforts that addressed most of the concerns raised regarding the initial submission. Further, they have expanded their acknowledgment of previous works that are directly relevant to their findings.

However, two of our initial critiques still stand:

1. In regard to the updated Figure 9. The data addressing the relationship/overlap between Rab39 and either Rab7 or Lamp1 are unconvincing. The example images suggesting changed distributions are not compelling, and the quantitative analyses of these data do not readily align with the primary data. We suggest removing this data from the manuscript entirely - it does not add to the strength of the work.
2. Several of the included CLEM images are more convincing, notably panel C-D and G-H. However, some specifics are still unclear. Is panel B' the zoomed in region of A as stated in the figure legend or the same as panel B as indicated by the overlay? Do these zoomed out panels correspond to any of the subsequent areas in the figure? If so please indicate. Panel H should have a scale bar.

Overall, the manuscript is both significant and rigorous and should be published following minor revisions.

We sincerely thank the reviewer for their thoughtful evaluation and constructive feedback.

In response to the remaining points:

1. *As suggested, we have removed the data presented in the previous Figure 9. To maintain clarity and focus, we now only retain the general organelle localization analysis of Rab39.*
2. *We have carefully revised the CLEM figure and its legend to clarify the correspondence between panels. Specifically, we corrected the description of panel B' and clearly indicated which zoomed-*

out panels correspond to subsequent zoomed-in areas. Additionally, we added a scale bar to panel H as requested.

Dear Patrik,

Congratulations on an excellent manuscript! I am very pleased to inform you that it has been accepted for publication in The EMBO Journal. Thank you for comprehensively addressing the initially raised referees' concerns and all editorial requests for changes and corrections.

If you have any questions, please do not hesitate to contact the Editorial Office. Thank you for your contribution to The EMBO Journal. Working with you has been a pleasure!

Best regards,

Ioannis
